# Glial cells undergo rapid changes following acute chemogenetic manipulation of cortical layer 5 projection neurons
Auguste Vadisiute [1,2] ✉, Elise Meijer [1], Rajeevan Narayanan Therpurakal[1,3], Marissa Mueller [1], Florina Szabó[1], Fernando Messore [1], Alfonsas Jursenas[4], Oliver Bredemeyer [2], Lukas B. Krone[1,5,6], Ed Mann [1], Vladyslav Vyazovskiy [1,7,8], Anna Hoerder-Suabedissen [1,7,8] & Zoltán Molnár [1,2] ✉

Bidirectional communication between neurons and glial cells is crucial to establishing and maintaining normal brain function. Some of these interactions are activity-dependent, yet it remains largely unexplored how acute changes in neuronal activity affect glial-to-neuron and neuron-to-glial dynamics. Here, we use excitatory and inhibitory designer receptors exclusively activated by designer drugs (DREADD) to study the effects of acute chemogenetic manipulations of a subpopulation of layer 5 cortical projection and dentate gyrus neurons in adult (Rbp4$^{Cre}$) mouse brains. We show that acute chemogenetic neuronal activation reduces synaptic density, and increases microglia and astrocyte reactivity, but does not affect parvalbumin (PV+) neurons, only perineuronal nets (PNN). Conversely, acute silencing increases synaptic density and decreases glial reactivity. We show fast glial response upon clozapine-N-oxide (CNO) administration in cortical and subcortical regions. Together, our work provides evidence of fast, activity-dependent, bidirectional interactions between neurons and glial cells.

Communication between neurons and glia is necessary for normal brain development and diverse adult brain functions such as homeostasis, neuronal network connectivity, and plasticity[1–3]. These interactions are relevant to injury protection, and infection prevention[2,4]. Disruptions to neuronal and glial functions can lead to various neuropathological states including epilepsy, stroke, and a range of neuropsychiatric disorders[1,5–8]. Some forms of glial communication with neurons are activity-dependent[9,10], yet it remains unclear how changes in neuronal activity affect glial cells, interactions with other neurons, and the timeframe of glial responses.

Microglia are the tissue-resident macrophages of the central nervous system (CNS)[11], and knowledge surrounding their diverse functions has increased rapidly in recent years. While it is suspected that neuron-microglia interactions may be activity-dependent[11–14] like synapse elimination and neuronal circuit remodelling in the developing brain[11,13–16], intercellular communications underlying systems-level neuron-microglia

plasticity remain unexplored. In contrast, astrocytes play a crucial role in homeostasis, synapse formation, and synaptic plasticity[17,18]. Astrocyte-neuron communication is an input-specific process tightly controlled by synaptic activity[9,19]. Following various pathological conditions and injuries, astrocytes become reactive - a process known as astrogliosis. Astrogliosis is considered to be beneficial for the CNS[20]; reactive astrocytes provide protection by restricting inflammation and the spread of infection[20,21]. However, excessive astrocyte reactivity may lead to a chronic inflammatory state, impaired synaptic function, and pathological network connectivity[5,6,22].

To better understand activity-dependent neuron-glial interactions in the mature brain, we validated and used acute chemogenetic methods to modify the activity of a subset of cortical and dentate gyrus (DG) neurons. This involved acute chemogenetic manipulation via designer receptors exclusively activated by designer drugs (DREADDs), which are activated upon clozapine-N-oxide (CNO) actuator administration[3,23]. hM3Dq

[1]Department of Physiology, Anatomy and Genetics, Sherrington Building, University of Oxford, Parks Road, Oxford, OX1 3PT, United Kingdom. [2]St John's College, University of Oxford, St Giles', Oxford, United Kingdom. [3]Department of Neurology, Düsseldorf University Hospital, Düsseldorf, Germany. [4]Baltic Institute of Advanced Technology, Vilnius, Lithuania. [5]University Hospital of Psychiatry and Psychotherapy, University of Bern, Bern, Switzerland. [6]Centre for Experimental Neurology, University of Bern, Bern, Switzerland. [7]Kavli Institute for Nanoscience Discovery, Sleep and Circadian Neuroscience Institute, University of Oxford, Oxford, United Kingdom. [8]Sleep and Circadian Neuroscience Institute, University of Oxford, Oxford, United Kingdom. ✉e-mail: auguste.vadisiute@dpag.ox.ac.uk; zoltan.molnar@dpag.ox.ac.uk

increases neuron excitability upon CNO binding[24]. This occurs due to hM3Dq activation of phospholipase C, which in turn activates the protein kinase C pathway via diacylglycerol, and intracellular $Ca^{2+}$ via inositol-1,4,5-triphosphate[25]. hM4Di activation by CNO inhibits neuronal activity by increasing Gβ/γ-protein regulated inwardly rectifying potassium channel function. This induces hyperpolarization[24] and inhibits synaptic vesicle release, thereby silencing neurons[26]. hM4Di is understood to strongly inhibit neurotransmitter release within minutes, with effects lasting for several hours[26].

In this study, we genetically expressed either hM3Dq or hM4Di in Cre-dependent manner in a subset of cortical and hippocampal neurons by crossing them with the Rbp4[Cre] mouse line[27–29]. We used DREADDs to acutely activate or inhibit Rbp4[Cre] neurons, in contrast to our previous studies which involved chronically silencing the same set of neurons[28,30–32]. We validated changes in neuron activity after chemogenetic manipulations with in vitro patch clamp recordings and immediate early gene expression. 90 min after activating Rbp4[Cre] neurons, synaptic density decreased near the cell bodies of visual cortex layer 5 (V1 L5) Rbp4[Cre] neurons and in their superior colliculus (SC) projection region. Conversely, when inhibiting Rbp4[Cre] neurons, synaptic density increased within V1 L5 and CA1 regions. In Rbp4[Cre]-hM3Dq, microglial cells retracted their processes and switched to a more reactive morphological state. Astrocytes also became more reactive in CA3 regions. In Rbp4[Cre]-hM4Di, only microglial reactivity decreased. Interestingly, in Rbp4[Cre] brains, both microglia and astrocytes rapidly showed increased reactivity in cortical and subcortical regions upon CNO application alone. We then explored the consequences of activating and silencing Rbp4[Cre] neurons on parvalbumin (PV+) interneurons. In Rbp4[Cre]-hM3Dq, PV+ neuron density was not altered, yet the density of cells surrounded by perineuronal nets (PNN) decreased in L2-3 and 5. Moreover, fewer PV+ cells were surrounded by perineuronal nets in cortical L6a of retrosplenial area (RSP), but we observed increased PV+ co-localisation with perineuronal nets in L4-5 of the primary somatosensory cortex (S1) after CNO application. Collectively, these findings provide insights into rapid and widespread activity-dependent interactions between neuronal networks and glial cells.

## Results

### Excitatory and inhibitory DREADDs validation in the specific combinations used

Adult excitatory (hM3Dq) and inhibitory (hM4Di) DREADD mice were crossed with Rbp4[Cre] to express the receptors in a specific subset of neurons (Fig. 1a) and the resulting mice were used for all experiments in this study. To validate effects elicited by acute manipulations following CNO application, we recorded in vitro responses and activity of L5 neurons in the primary motor cortex (M1) with increasing current administration (Fig. 1b). We exposed neurons to repeating step-current protocols before and after 200 nM CNO application. We evaluated membrane properties at -100pA for both Rbp4[Cre]-hM3Dq and Rbp4[Cre]-hM4Di before and after CNO exposure, and measured effects on evoked neuron activity. Acute CNO application in Rbp4[Cre]-hM3Dq increased the number of evoked events at low currents; however, these quickly plateaued to resemble baseline activity at higher current steps (Fig. 1c, d). Both the reduction in activation threshold and the loss of activity at higher currents can be attributed to tonic membrane depolarization. When analysed, the resting membrane potential (mV) significantly reduced upon CNO application (Fig. 1d). This effect was maintained throughout the duration of CNO application (Fig. 1d). In contrast, in slices expressing the inhibitory DREADD, CNO application reduced activity. Analysis of spike frequency responses to increasing current injection showed no statistically significant delay in the onset of evoked responses (Fig. 1e, f). We found no significant alteration in membrane potential either immediately after CNO application or at later time points, which would have explained the reduction in evoked activity (Fig. 1f). Instead, neuronal membrane resistance increases following CNO administration, both immediately and over time.

To validate effects of CNO concentration on both excitatory and inhibitory DREADDs, we injected mice with different doses of CNO

(excitatory: 0.05 mg/kg, 0.1 mg/kg, 10 mg/kg; inhibitory: 1 mg/kg, 5 mg/kg) or with saline as a control (Fig. 1g, Supplementary Table 3). After systemic CNO application, DREADD-induced behavioural effects are expected to begin within 15 min[32] with plasma CNO concentration peaking within 90 min[33]. We closely monitored animals following CNO administration through video recordings of their behaviour (Supplementary Table 4). In Rbp4[Cre]-hM3Dq, qualitative behavioural effects of CNO administration seemed to peak after about 1 h following intraperitoneal (IP) injection. Low doses appeared to reduce locomotion and induce a sedation-like state. Higher doses caused non-convulsive seizures, all of which gradually wore off. In contrast, Rbp4[Cre]-hM4Di mice demonstrated no overt adverse effects following CNO administration. All mice were perfused 90 min after saline or CNO injection.

Next, we tested whether the alteration in cellular activity, as measured by cFOS immunoreactivity, is CNO dose dependent. We measured cFos+ cell density in saline- and CNO-injected mice in brain regions containing Cre+ cell bodies (M1, primary visual cortex (V1) L5, Fig. 1h–j) and at the projection sites of Cre+ cell populations (SC, and CA1–CA3 regions (Fig. 2a–d)). We observed an increase in cFos+ neuron density, which is consistent with an expected increase in cellular activity following CNO administration (Fig. 1i). In contrast, we observed decreased cFos+ density in Rbp4Cre-hM4Di (Fig. 1j, k).

Specifically, in Rbp4[Cre]-hM3Dq, we observed increased cFos+ density after CNO application (CNO effect: $F_{(3,36)} = 208.6$, $p < 0.0001$, Fig. 2a). This increase was dose dependent in the region of Cre+ cell bodies (Fig. 2a) and cFos+ cell density increased across all cortical layers in V1 (Supplementary Fig. 1a, not quantified). In SC, cFos+ cell density increased significantly only with 0.1 mg/kg CNO (Fig. 2a). In CA1, cFos+ density significantly increased with 0.1 mg/kg and 10 mg/kg CNO, and in the CA3 region containing terminals of DG mossy fibres, we observed significantly increased cFos+ density with 0.05 mg/kg and 10 mg/kg CNO (Fig. 2a). This suggests that administering CNO to Rbp4[Cre]-hM3Dq increases neuronal activity both directly in Cre+ neurons, as well as in some axonal projection target regions. In contrast, administering CNO to Rbp4[Cre]-hM4Di mice decreased cFos+ cell density (CNO effect: $F_{(2,3)} = 168.4$, $p = 0.0008$, Fig. 2c). In V1 L5 this decrease was dose dependent (Fig. 2c), although all layers are likely to have been affected through circuit level interactions (Supplementary Fig. 1b, not quantified). Dose-dependent decreases were also observed in SC, with no changes in CA1–CA3 regions (Fig. 2c, d). This is consistent with prior work[33] which showed that neuronal activity was inhibited in the Rbp4[Cre] population of cortical L5 neurons and in SC target regions.

We performed additional experiments to evaluate whether the background strain or genetic modification itself impacts cortical neurons' activity levels without CNO administration. Mice from the cortical L5 driver line (Rbp4[Cre]), the hM4Di receptor line, and the hM3Dq line (Supplementary Fig. 1c, d) were injected only with saline and activity was assessed using cFos. We observed higher cFos+ cell density in the hM4Di line compared to Rbp4[Cre] injected with saline in CA3 (Supplementary Fig. 1e), and in hM3Dq compared to Rbp4[Cre]-hM3Dq in SC and CA3 (Supplementary Table 4 and Fig. 1e). No differences in cFos+ cell density were observed between Rbp4[Cre]-hM3Dq and Rbp4[Cre]-hM4Di (Supplementary Fig. 1f), suggesting constitutive Cre- and CNO-independent DREADD receptor expression and activity as recently reported (https://www.jax.org/strain/026219#). We injected the Rbp4[Cre] with saline and 10 mg/kg CNO (Supplementary Fig. 2a) and immunostained for cFos+. No changes in cFos+ neuron density were observed in V1 and CA3 (Supplementary Fig. 2b, c), confirming that CNO alone does not influence neuronal activity.

### Chemogenetic manipulations of Rbp4[Cre] neurons affect excitatory synapses

To examine effects of acute chemogenetic manipulations, we measured synaptic density via immunohistochemical co-localisation of presynaptic (vGlut1) and postsynaptic (PSD95) markers, which are specific for excitatory synapses. In Rbp4[Cre]-hM3Dq, synaptic density decreased after CNO application (CNO effect: $F_{(3,28)} = 36.57$, $p < 0.0001$), being significantly

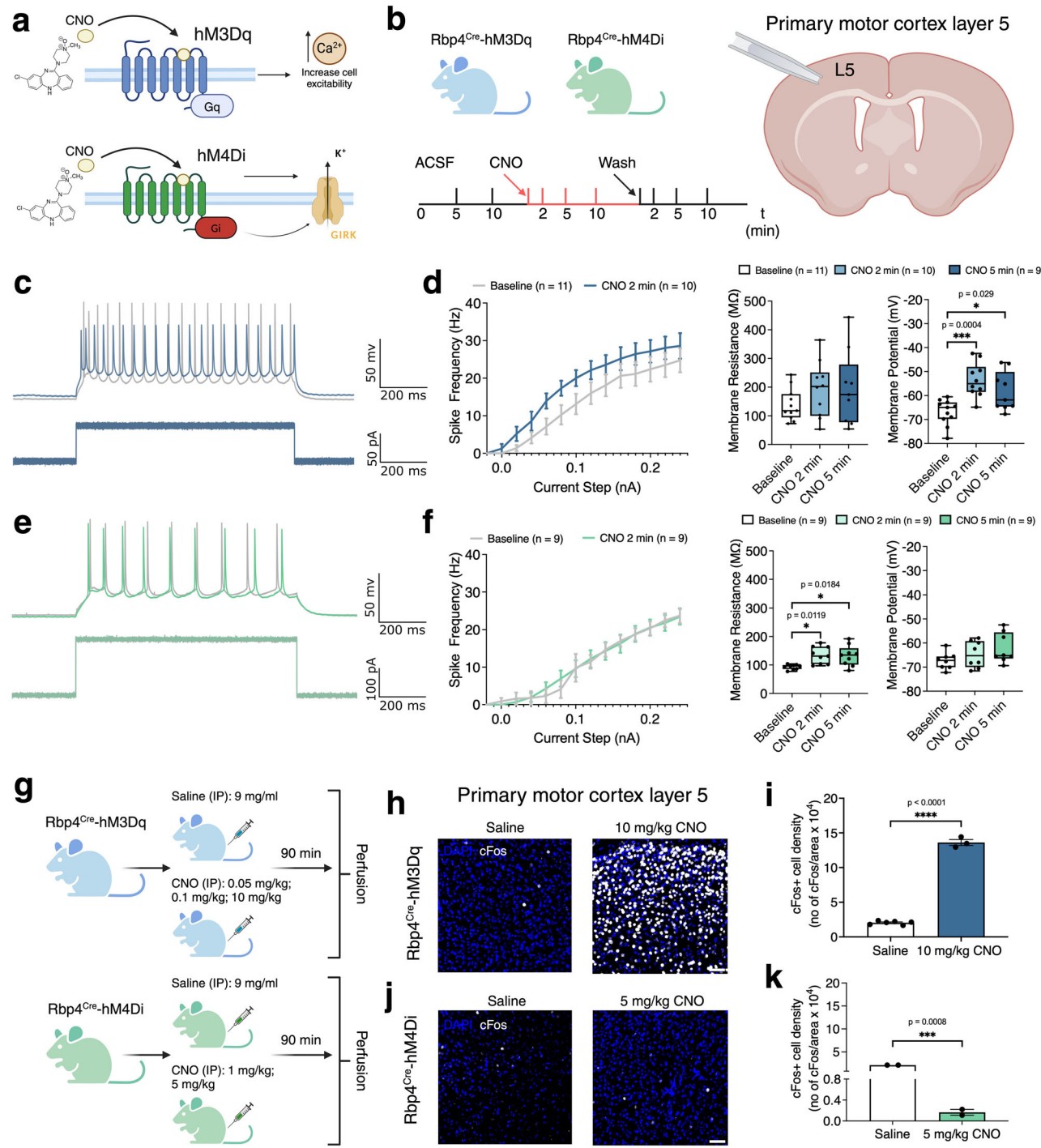

lower in V1 L5 and SC regions (0.05 mg/kg and 10 mg/kg CNO administration). No significant changes were observed in CA1 and CA3 (Fig. 3a, b). This suggests that acute activation of Rbp4[Cre] neurons leads to a reduction in excitatory synapses to compensate for increased cellular activity. In Rbp4[Cre]-hM4Di, synaptic density increased (CNO effect: $F_{(2,12)} = 163.7$, $p < 0.0001$), with dose-dependency in V1 L5 and CA1, yet no effect in SC and CA3 (Fig. 3c, d). This suggests that silencing Rbp4[Cre] neurons increases the number of excitatory synapses to compensate for reduced cellular activity.

### Acute chemogenetic manipulations of neuronal activity leads to rapid changes in microglial density and morphology

To evaluate whether acute manipulations of neuronal activity affect the dynamics of microglial cells, we measured microglial density, CD68+

lysosomes of Iba1+ microglia, and microglial morphology. No significant difference in microglial density was observed in V1 and laminar distribution was unaltered in Rbp4[Cre]-hM3Dq after CNO application (CNO effect: $F_{(3,7)} = 3.375$, $p = 0.0839$, Fig. 4a–c). However, a significant decrease in density was observed in subcortical regions (CNO effect: $F_{(3,9)} = 28.78$, $p < 0.0001$, Fig. 4d). This decrease was dose-dependent in SC, but only observed for the highest dose of CNO (10 mg/kg) in the CA1–CA3 region (Fig. 4d). This decrease in density contrasted with an increase in morphological signs of microglial response to CNO application, as well as an increase in CD68+ phagocytic lysosomes in Iba1+ cells (CNO effect: $F_{(3,9)} = 108.1$, $p < 0.0001$, Fig. 4e, f). CD68+ phagocytic lysosomes increased in Iba1+ cells in SC (0.1 mg/kg and 10 mg/kg CNO), CA1 (0.05 mg/kg, 0.1 mg/kg, and 10 mg/kg CNO) and CA3

**Fig. 1 | Electrophysiological validation of Rbp4$^{Cre}$-hM3Dq and Rbp4$^{Cre}$-hM4Di DREADD function in cortical layer 5 pyramidal neurons. a, b** Schematics of excitatory (blue) and inhibitory (green) DREADD receptors and experimental procedures. Patch-clamp electrophysiological recordings were done in consecutive fashion. Two baseline recordings at 5 and 10 min under constant ACSF application were followed by administration of CNO, and three recordings at 2, 5 and 10 min post application, followed by a washout step. During this last step, control recordings at different timepoints were obtained. **c** Representative example of Rbp4$^{Cre}$-hM3Dq membrane voltage response to a step current application. **d** Input-Output curve before and after CNO administration, in response to increasing current applications. Gray represents before, and blue traces after CNO administration. Membrane resistance of recorded neurons before and after CNO administration in Rbp4$^{Cre}$-hM3Dq brain slices. No significant differences between baseline and CNO administrations (left). Resting membrane potential of recorded neurons before and after CNO administration. After CNO administration, the membrane is tonically depolarized (right). $n$ = 7 mice (baseline 11 cells, CNO 2 min 10 cells, CNO 5 min 9 cells recorded), one-way ANOVA via Dunnett's multiple comparison test.

**e** Representative example of Rbp4$^{Cre}$-hM4Di membrane voltage response to a step current application. **f** Membrane resistance of neurons recorded before and after CNO administration in Rbp4$^{Cre}$-hM4Di brain slices. CNO administration significantly increases membrane resistance (left). Resting membrane potential of recorded neurons before and after CNO administration. No significant difference in resting membrane potential (right). $n$ = 7 mice (baseline 9 cells, CNO 2 min 9 cells, CNO 5 min 9 cells recorded), one-way ANOVA via Dunnett's multiple comparison test. **g** Schematics of excitatory and inhibitory DREADD experiments. **h, i** Representative images of cFos+ immunolabelling in M1 of Rbp4$^{Cre}$-hM3Dq mice show increased cFos+ cell density after CNO application. $n$ = 6 saline and $n$ = 3 10 mg/kg CNO mice, two-tailed unpaired Student's $t$ test. **j, k** Representative images of cFos+ immunolabelling in M1 of Rbp4$^{Cre}$-hM4Di mice shows decreased cFos+ cell density after CNO application. $n$ = 2 saline and $n$ = 2 5 mg/kg CNO mice, Mann–Whitney test. All data presented as mean ± SEM, *$p < 0.05$, **$p < 0.01$, ***$p < 0.001$, ****$p < 0.0001$. Scale bar of **h, j** - 100 μm. Detailed statistical information is listed in Supplementary Data 1. Source data is provided as a Supplementary Data 2 file. Created with BioRender.com.

(10 mg/kg CNO), but no significant changes were observed in V1 L5 (Fig. 4f). In contrast, in Rbp4$^{Cre}$-hM4Di, microglial density in V1 L5 increased (CNO effect: $F_{(2,3)} = 22.26$, $p = 0.0159$, Fig. 4g) with no dose-dependent effect (Fig. 4h), while V1 was not significantly altered overall (Fig. 4i). Microglial density increased after CNO application in subcortical regions (CNO effect: $F_{(2,3)} = 17,28$, $p = 0.0226$), but there were no significant dose-dependent effects in individual regions (Fig. 4j). In Rbp4$^{Cre}$-hM4Di, a significant decrease in CD68+ in Iba1+ cells was observed (CNO effect: $F_{(2,12)} = 20.20$, $p = 0.0001$) (Fig. 4k, l), specifically in SC with 1 mg/kg CNO, yet no significant differences were observed in V1 L5, CA1–CA3 (Fig. 4l). No significant cortical changes were observed in microglial cell density in Rbp4$^{Cre}$ mice injected with 10 mg/kg CNO compared to saline (CNO effect: $F_{(1,7)} = 0.01378$, $p = 0.9098$, Fig. 4m–o). However, we observed an increase in density in CA3 (Fig. 4p). Unexpectedly, we observed an increase in CD68 in Iba1+ microglia (CNO effect: $F_{(1,28)} = 32.86$, $p < 0.0001$, Fig. 4q, r) in V1 L5, SC and CA1, suggesting an acute microglial response to CNO application alone (Fig. 4q, r). We further compared changes in microglial density between Rbp4$^{Cre}$ and Rbp4$^{Cre}$-hM3Dq injected with the highest dose of CNO. No significant difference in density was observed in V1 laminar distribution, across V1 and SC when comparing Rbp4$^{Cre}$ and Rbp4$^{Cre}$-hM3Dq mice injected with saline or CNO (genotypes x treatment: laminar $F_{(3,12)} = 0.9723$, $p = 0.4378$; V1: $F_{(1,12)} = 1.753$, $p = 0.2102$; SC: $F_{(1,14)} = 0.04187$, $p = 0.8408$, Supplementary Fig. 3a–c). We observed statistically significant interaction between genotypes and treatment conditions in microglial density in the CA1 region (genotypes x treatment: $F_{(1,14)} = 7.196$, $p = 0.0179$, Supplementary Fig. 3d). In CA3, microglial density was increased in saline Rbp4$^{Cre}$-hM3Dq compared with saline Rbp4$^{Cre}$ but decreased in Rbp4$^{Cre}$-hM3Dq after 10 mg/kg CNO application compared to Rbp4$^{Cre}$ injected with the same concentration of CNO (genotypes x treatment: $F_{(1,14)} = 56,49$, $p < 0.0001$, Supplementary Fig. 3e). Statistically significant interactions were also observed when assessing the percentage of Iba1+ microglia that were also CD68+. Specifically, we found no statistically significant interaction between genotypes and treatment conditions in CD68 in Iba1 microglia in V1 L5 and SC, but it was statistically significant in CA1–CA3 regions (genotypes × treatment: V1 L5 ($F_{(1,14)} = 0.8998$, $p = 0.3589$; SC: $F_{(1,14)} = 2.611$, $p = 0.1284$; CA1: $F_{(1,14)} = 22.05$, $p = 0.0003$; CA3: $F_{(1,14)} = 43.66$, $p < 0.0001$, Supplementary Fig. 3f–i). Moreover, in all brain regions assessed, the percentage of CD68 in Iba1+ microglia was increased in saline injected Rbp4$^{Cre}$-hM3Dq compared with Rbp4$^{Cre}$, and in 10 mg/kg CNO injected Rbp4$^{Cre}$-hM3Dq compared with Rbp4$^{Cre}$ (Supplementary Fig. 3f–i). These data suggest that the high dose of CNO has no effect with or without activation in V1 and Rbp4$^{Cre}$ projection region to SC but has an effect both by itself and in combination with chemogenetic activation in the CA1–CA3 regions. Given the effect of high doses of CNO alone in the Rbp4$^{Cre}$, we cannot be certain whether the reduction in microglial density observed in CA1 and CA3 in Rbp4$^{Cre}$-hM3Dq after 10 mg/kg CNO (Fig. 4d) is the result of CNO, or chemogenetic activation.

To further investigate microglial response to acute manipulations, we measured microglial volume, soma area, and the number of primary branches in V1 L5 and CA3 (Fig. 5a), in Rbp4$^{Cre}$-hM3Dq and Rbp4$^{Cre}$-hM4Di. In Rbp4$^{Cre}$-hM3Dq, we observed changes in microglial volume (CNO effect: $F_{(2.608,67.81)} = 12.17$, $p < 0.0001$), primary branches (CNO effect: $F_{(2.462,64.01)} = 14.25$, $p < 0.0001$), and soma area (CNO effect: $F_{(2.415,62.80)} = 3.403$, $p = 0.0314$). Microglial volume decreased in V1 L5 (0.05 mg/kg CNO) and in CA3 (0.05 mg/kg, 0.1 mg/kg CNO), but not at higher CNO doses (Fig. 5b, d). The number of primary branches decreased in both regions, but only with the highest dose of CNO, while microglial soma size did not differ upon CNO administration (Fig. 5b, d). In Rbp4$^{Cre}$-hM4Di, only primary branches were affected by CNO application (CNO effect: microglial volume $F_{(1.802,39.63)} = 0.5798$, $p = 0.5474$; primary branches $F_{(1.717,56.66)} = 3.885$, $p = 0.032$; soma area $F_{(1.928,42.42)} = 0.4575$, $p = 0.6289$; Fig. 5c, d).

To better understand whether acute manipulations of neuronal activity impact microglial processes, we measured the number of intersections and Sholl's regression coefficients (Fig. 5e–g). In Rbp4$^{Cre}$-hM3Dq, acute activation led to decreased microglial cells' processes (CNO effect: 3D Sholl intersections $F_{(2.534,125.0)} = 4.646$, $p < 0.0001$; regression coefficient $F_{(2.672,131.8)} = 14.59$, $p < 0.0001$; Fig. 5f). The number of intersections decreased only in CA3 (0.1 mg/kg and 10 mg/kg CNO, Fig. 5f). Sholl's regression coefficient, a measure of the change in process density as a function of distance from the cell body, was increased for V1 L5 (0.1 mg/kg) and CA3 (0.05 mg/kg and 0.1 mg/kg). In Rbp4$^{Cre}$-hM4Di, microglial processes also decreased (CNO effect: intersections $F_{(1.796,59.27)} = 41.49$, $p < 0.000$; regression coefficient $F_{(1.892,41.62)} = 1.413$, $p = 0.2545$; Fig. 5g). The intersections of microglial cells' processes decreased in a dose-dependent manner in V1 L5 and CA3, however no significant differences were identified in the regression coefficients (Fig. 5f). Collectively, these changes suggest that acute manipulations of neuronal activity led to rapid microglial responses.

## Changes in astrocyte reactivity and function after acute chemogenetic manipulations of neuronal activity

To evaluate changes in astrocytes after acute chemogenetic manipulations, we used GFAP and S100β immunolabelling. S100β is a marker of mature astrocytes and inflammation[33,34], and GFAP is associated with reactive astrocytes[21,35,36]. In Rbp4$^{Cre}$-hM3Dq, astrocyte density and reactivity was increased in V1 (CNO effect: GFAP+ $F_{(3,6)} = 7.300$, $p = 0.0199$; S100β+ $F_{(3,6)} = 11.03$, $p = 0.0074$; GFAP+ cells expressing S100β $F_{(3,6)} = 10.36$, $p = 0.0087$; Fig. 6a–d). This effect was due to an increase in S100β+ cell density with 10 mg/kg dose of CNO (Supplementary Fig. 4a). There were no effects of chemogenetic activation on GFAP+ cell density

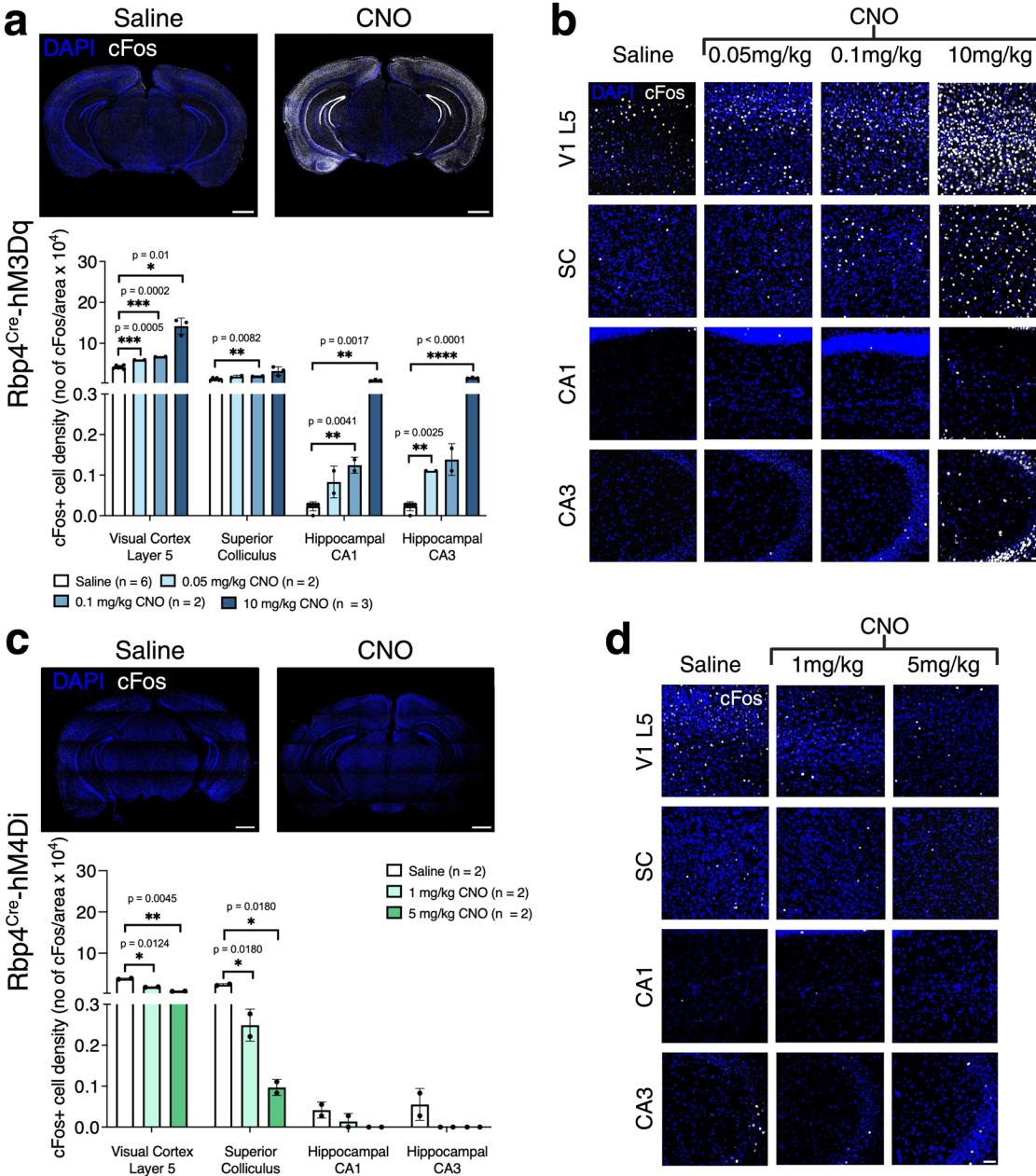

**Fig. 2 | Changes in cFos+ immunoreactivity in Rbp4^Cre-hM3Dq and Rbp4^Cre-hM4Di mice. a** Representative images of cFos+ immunostained sections of saline and CNO injected Rbp4^Cre-hM3Dq mouse shows an increase in cFos+ cell density in the CNO-injected mouse visual cortical layer 5 (V1 layer 5), superior colliculus (SC), and hippocampal CA1 and CA3 regions. Mice: $n = 6$ saline, $n = 2$ CNO 0.05 mg/kg, $n = 2$ CNO 0.1 mg/kg, $n = 3$ CNO 10 mg/kg, with an average of 3 sections per region, evaluated using mixed-effects ANOVA via the corrected method of Benjamini and Yekutieli. **b** Representative images of CNO dose-dependent changes of cFos+ cells (white) in Rbp4^Cre-hM3Dq mouse V1 layer 5, SC, CA1 and CA3 regions. **c** Representative images of cFos+ immunostained sections of saline and CNO

injected Rbp4^Cre-hM4Di mouse show a decrease in cFos+ cell density in CNO injected mouse V1 layer 5, SC, CA1 and CA3 regions. Mice: $n = 2$ saline, $n = 2$ CNO 1 mg/kg, $n = 2$ CNO 5 mg/kg, with an average of 3 sections per region, evaluated using mixed-effects ANOVA via the corrected method of Benjamini and Yekutieli. **d** Representative images of CNO dose-dependent changes of cFos+ cells (white) in Rbp4^Cre-hM4Di mouse V1 layer 5, SC, CA1 and CA3 regions. All data presented as mean ± SEM, false discovery rate of 0.05 adjusted using Benjamini and Yekutieli, $*p < 0.05$, $**p < 0.01$, $***p < 0.001$, $****p < 0.0001$ with scale bars of **a, c** - 1000 μm and **b, d** - 100 μm. Detailed statistical information is listed in Supplementary Data 1. Source data is provided as a Supplementary Data 2 file. Created with BioRender.com.

alone, or on the layer distribution of any of these cell types (Fig. 6b–d). In Rbp4^Cre-hM4Di, no significant changes in astrocyte reactivity were observed (CNO effect: GFAP+ $F_{(2,3)} = 3.752$, $p = 0.1527$; S100β+ $F_{(2,3)} = 1.672$, $p = 0.3251$; GFAP+ cells expressing S100β $F_{(2,3)} = 5.527$, $p = 0.0986$; Fig. 6e–h, Supplementary Fig. 4b). Additionally, we measured astrocytes' response to CNO in Rbp4^Cre (Fig. 6i) and surprisingly observed a significant increase in astrocyte reactivity to CNO itself in the absence of any DREADDs that could be activated by it (CNO effect: GFAP+ $F_{(1,7)} = 12.66$, $p = 0.0092$; S100β+ $F_{(1,7)} = 3.622$, $p = 0.0988$; GFAP+ cells expressing

S100β $F_{(1,7)} = 7.150$, $p = 0.0318$; Fig. 6i–l). A significant increase in GFAP+ density was observed in L1 and L6 after CNO application, yet no significant changes in S100β+ density were observed (Fig. 6h, Supplementary Fig. 4c). The proportion of GFAP+ cells expressing S100β significantly increased only in L1. These changes in Rbp4^Cre suggest that CNO application alone leads to increased astrocyte response (Fig. 6h, i, Supplementary Fig. 4c). We further investigated changes in GFAP+, S100β+ and the proportion of GFAP+ cells expressing S100β between Rbp4^Cre and Rbp4^Cre-hM3Dq injected with saline and the highest dose of CNO. In V1, we found no

**Fig. 3 | Changes in synaptic density in Rbp4^Cre-hM3Dq and Rbp4^Cre-hM4Di mice. a** In excitatory DREADD mice (blue), vGlut1/PSD95 density decreases in V1 layer 5 and CA3. Mice: $n = 4$ saline, $n = 2$ CNO 0.05 mg/kg, $n = 2$ CNO 0.1 mg/kg, $n = 3$ CNO 10 mg/kg, with an average of 3 sections per region and per mouse, evaluated using mixed-effects ANOVA via the corrected method of Benjamini and Yekutieli. **b** Representative images of vGlut1 and PSD95 immunolabeling in saline and CNO injected Rbp4^Cre-hM3Dq mice, as well as a schematic illustration of changes in pre- and post- synaptic terminals during short-term chemogenetic activation. **c** In inhibitory DREADD mice (green), vGlut1/PSD95 density increases in V1 layer 5, SC, CA1 and CA3 regions. Mice: $n = 2$ saline, $n = 2$ CNO 1 mg/kg, $n = 2$ CNO 5 mg/kg, with an average of 3 sections per region and per mouse, evaluated using two-way ANOVA via the corrected method of Benjamini and Yekutieli. **d** Representative images of vGlut1 (magenta) and PSD95 (cyan) immunolabeling in saline and CNO injected Rbp4^Cre-hM3Dq mice, as well as a schematic illustration of changes in pre- and post- synaptic terminals during short-term chemogenetic silencing. All data presented as mean ± SEM, false discovery rate of 0.05 adjusted using Benjamini and Yekutieli, *$p < 0.05$, **$p < 0.01$, ***$p < 0.001$, ****$p < 0.0001$ with scale bars of **b, d** - 100 μm. Detailed statistical information is listed in Supplementary Data 1. Source data is provided as a Supplementary Data 2 file. Created with BioRender.com.

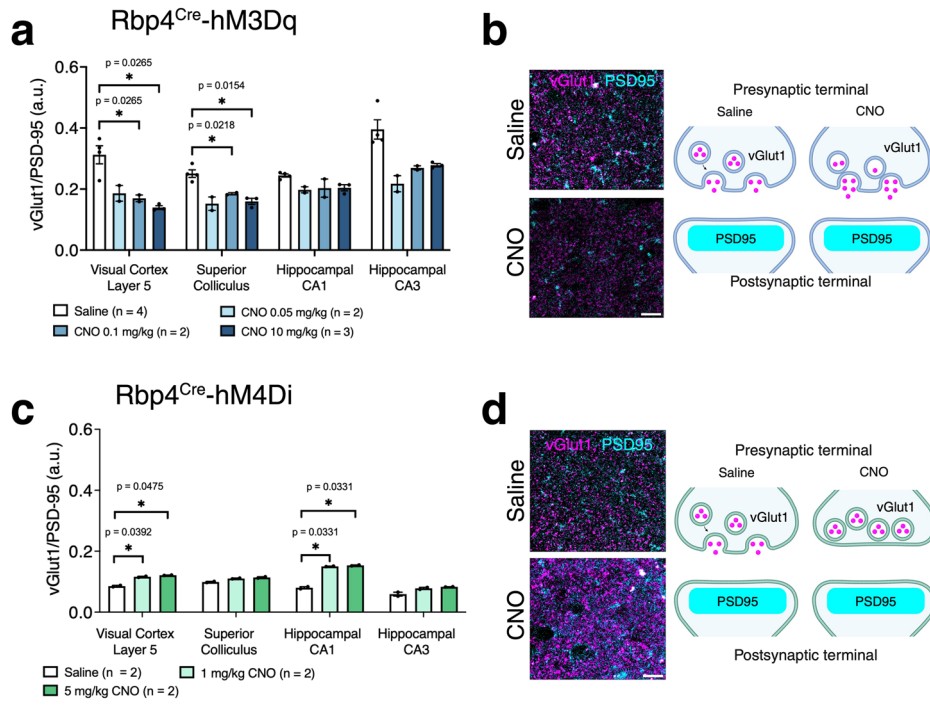

statistically significant interaction between genotypes and treatment conditions in laminar distribution of GFAP+ cells or across V1 (genotypes x treatment: laminar $F_{(3,12)} = 0.9723$, $p = 0.4378$; across V1: $F_{(1,11)} = 3.055$, $p = 0.1083$), but we observed significant changes in the laminar distribution of S100β+ and GFAP+ cells expressing S100β, but not across V1 (genotypes x treatment: laminar S100β+: $F_{(3,11)} = 6.909$, $p = 0.007$, across $F_{(1,11)} = 3.417$, $p = 0.0916$; laminar GFAP+ cells expressing S100β $F_{(3,11)} = 5.605$, $p = 0.014$, across $F_{(1,11)} = 2.559$, $p = 0.1379$ Supplementary Fig. 5a, b, f, j).

We furthermore investigated astrocyte responses in subcortical regions. In Rbp4^Cre-hM3Dq, only GFAP+ density increased (CNO effect: GFAP+ $F_{(3,6)} = 11.50$, $p = 0.0067$; S100β+ $F_{(3,6)} = 4.255$, $p = 0.0623$; GFAP+ cells expressing S100β $F_{(3,6)} = 2.536$, $p = 0.1531$; Fig. 7a–d). This effect was confined to the CA3 region and the highest dose of CNO (10 mg/kg) (Fig. 7b). In Rbp4^Cre-hM4Di, there were no significant changes in GFAP+ density (CNO effect: $F_{(2,3)} = 7.355$, $p = 0.0697$) or the proportion of GFAP+ cells expressing S100β (CNO effect: $F_{(2,3)} = 3.341$, $p = 0.1725$; Fig. 7e–h), yet we observed a significant decrease in S100β+ density alone (CNO effect: $F_{(2,3)} = 30.98$, $p = 0.0099$). We only observed a decrease in S100β+ density in SC and CA1 with 5 mg/kg CNO (Fig. 7g). In Rbp4^Cre mice injected with CNO we surprisingly observed rapid changes in astrocytes' responses in the absence of any DREADDs that could be activated by it (CNO effect: GFAP+ $F_{(1,7)} = 57.29$, $p = 0.0001$; S100β+ $F_{(1,7)} = 131.7$, $p < 0.0001$; GFAP+ cells expressing S100β $F_{(1,21)} = 125.1$, $p < 0.0001$; Fig. 7i–l). Specifically, GFAP+ density significantly increased in CA1 and CA3, with no changes in SC (Fig. 7j). S100β+ density and the proportion of GFAP+ cells expressing S100β increased in SC, CA1, and CA3 after CNO application (Fig. 7k, l). In CA1, we only observed a statistically significant interaction between genotypes and treatment conditions in GFAP+ and S100β+ cell densities and GFAP+ cells expressing S100β in CA1 (genotypes x treatment: GFAP+ $F_{(1,11)} = 16.74$, $p = 0.0018$; S100β+ $F_{(1,11)} = 8.632$, $p = 0.0135$; GFAP+ cells expressing S100β: $F_{(1,11)} = 17.25$, $p = 0.0016$, Supplementary Fig. 5d, h, l), but not in SC (genotypes x treatment

conditions: GFAP+ $F_{(1,11)} = 0.9638$, $p = 0.3473$; S100β+ $F_{(1,11)} = 3.071$, $p = 0.9957$; GFAP+ cells expressing S100β: $F_{(1,11)} = 1.470$, $p = 0.2508$, Supplementary Fig. 5c, g, k). GFAP+ and S100β+ cell densities were increased in Rbp4^Cre-hM3Dq compared to Rbp4^Cre injected with saline, and the density of GFAP+ cells expressing S100β was lower in Rbp4^Cre-hM3Dq compared to Rbp4^Cre injected with 10 mg/kg CNO (Supplementary Fig. 5d, h, l). Finally, in CA3, we observed a statistically significant interaction between genotypes and treatment conditions (genotypes x treatment: GFAP+ $F_{(1,11)} = 6.654$, $p = 0.0256$; S100β+ $F_{(1,11)} = 13.38$, $p = 0.0038$; GFAP+ cells expressing S100β: $F_{(1,11)} = 30.60$, $p = 0.0002$, Supplementary Fig. 5e, i, m). GFAP+ density was increased in Rbp4^Cre-hM3Dq mice compared with Rbp4^Cre injected with saline and CNO. Moreover, following CNO injection, GFAP+ cells expressing S100β decreased in Rbp4^Cre-hM3Dq mice compared to Rbp4^Cre.

Altogether, astrocyte responses to altered neuronal activity appear to be region-specific, with changes in Rbp4^Cre suggesting that astrocytes are also sensitive to CNO application itself. Chemogenetic activation via CNO has no effect on Rpb4^Cre L5 and its projections to SC yet the highest dose of CNO affects the DG Rpb4^Cre cells' projection region of CA3 and CA1 in the absence of any DREADDs that could be activated by it.

### Chemogenetic manipulations of Rbp4^Cre neuron activity in layer 5 has no effect on PV+ neurons

We previously described changes in the laminar distribution of the parvalbumin interneurons after chronic abolition of evoked vesicle release from L5 cortical projection neurons[32]. To further evaluate how acute chemogenetic manipulations affect the same cortical cells, we investigated effects on GABAergic interneurons. PV interneurons are fast-spiking GABAergic cells that provide perisomatic innervation in the cortex[37] and play a crucial role in cortical inhibition. We used PV immunolabeling to evaluate acute activity-dependent changes in interneurons, and plant-based lectins (*Vicia villosa* agglutinin (VVA)) for detecting components of PNN. PNN are extracellular matrix elements surrounding PV interneurons in various regions of the

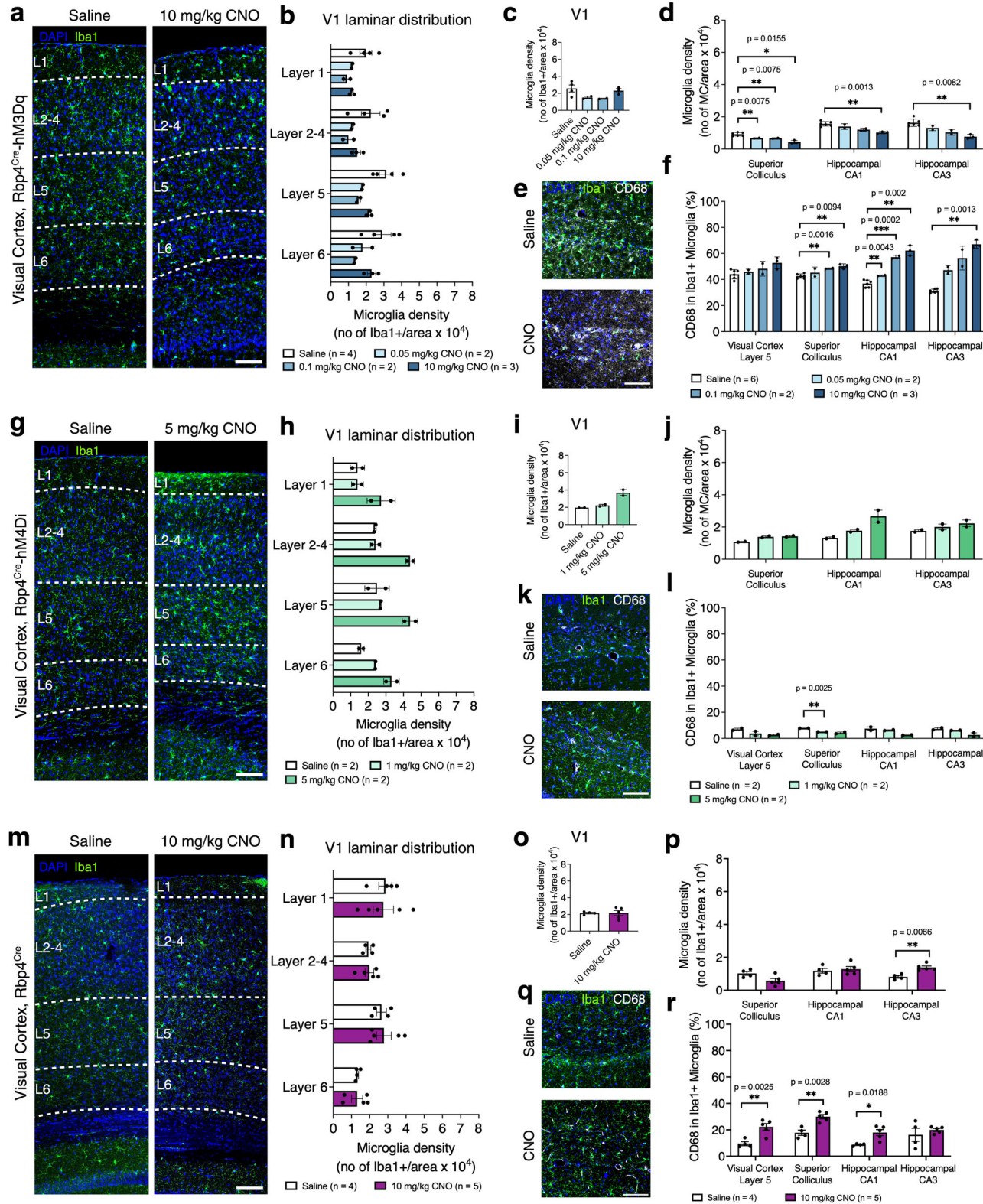

brain, including the cortex, and are responsible for synaptic homeostasis and stabilising synaptic plasticity[38]. Here, we evaluated PV+ interneuron density, VVA labelling and PV+ cell colocalization with VVA in different cortical and subcortical regions. We measured PV+ density in the RSP, S1 and posterior parietal association area (PTLp).

In Rbp4$^{Cre}$-hM3Dq, we analysed PV+ density in saline and 10 mg/kg CNO brains (Fig. 8a, b). No significant changes in PV+ density was

observed after CNO application in RSP, PTLp and S1 (CNO effect: $F_{(1,4)} = 2.613$, $p = 0.1813$ Fig. 8c). We measured laminar PV+ distribution and observed no significant effects of CNO administration on PV+ interneuron densities and laminar distributions within and between brain regions (CNO effect: RSP $F_{(1,4)} = 1.907$, $p = 0.2394$; PTLp $F_{(1,4)} = 0.8506$, $p = 0.4086$; S1 $F_{(1,4)} = 5.841$, $p = 0.073$; Fig. 8a–l). However, after CNO application, PV + VVA− cell densities increased in RSP L6a,

**Fig. 4 | Changes in microglial density in response to CNO application in Rbp4[Cre]-hM3Dq and Rbp4[Cre]-hM4Di mice and in Rbp4[Cre] without DREADD receptor.**
**a** Representative images of microglial density (green) in V1 of excitatory DREADD mice in saline and CNO injected conditions. **b–d** Changes in microglial density in V1, V1 layer 5, SC, CA1 and CA3 regions. Mice: $n = 4$ saline, $n = 2$ CNO 0.05 mg/kg, $n = 2$ CNO 0.1 mg/kg, $n = 3$ CNO 10 mg/kg, with an average of 3 sections per region and per mouse, evaluated using mixed-effects ANOVA via the corrected method of Benjamini and Yekutieli. **e, f** Representative images of CD68+ phagocytic lysosomes (grey) in Iba1+ positive microglia (green) in Rbp4[Cre]-hM3Dq animals in saline and CNO injection conditions. In CNO-injected mice, morphological features indicate that microglial response increases in a dose-dependent manner. Mice: $n = 4$ saline, $n = 2$ CNO 0.05 mg/kg, $n = 2$ CNO 0.1 mg/kg, $n = 3$ CNO 10 mg/kg, with an average of 3 sections per region and per mouse and with statistics involving mixed-effects ANOVA via the corrected method of Benjamini and Yekutieli. **g** Representative images of microglial density (green) in V1 of inhibitory DREADD animals in saline and CNO injected conditions. **h–j** Changes in microglial density in V1, V1 layer 5, SC, CA1 and CA3 in Rbp4[Cre]-hM4Di mice injected with 5 mg/kg CNO. Mice: $n = 2$ saline, $n = 2$ CNO 1 mg/kg, $n = 2$ CNO 5 mg/kg, with an average of 3 sections per region and per mouse, evaluated using mixed-effects ANOVA via the corrected method of Benjamini and Yekutieli. **k, l** Representative images of CD68+ phagocytic lysosomes (grey) in Iba1+ positive microglia (green) in Rbp4[Cre]-hM4Di in saline and CNO injected mice. In CNO injected mice, microglial activity decreases in a dose-dependent manner in V1, but only with 5 mg/kg CNO in SC, CA1, CA3. Mice: $n = 2$ saline, $n = 2$ CNO 1 mg/kg, $n = 2$ CNO 5 mg/kg, with an average of 3 sections per region and per mouse, evaluated using mixed-effects ANOVA via the corrected method of Benjamini and Yekutieli. **m** Representative images of microglial density (green) in V1 of Rbp4[Cre] mice in saline and CNO injected conditions. **n–p** Changes in microglial cell density in V1, V1 layer 5, SC, CA1 and CA3 regions of Rbp4[Cre] mice. Mice: $n = 4$ saline, $n = 5$ CNO 10 mg/kg, with an average of 3 sections per region and per mouse, evaluated using mixed-effects ANOVA via Holm–Šidak's multiple comparisons test. **q, r** Representative images of CD68+ phagocytic lysosomes (grey) in Iba1+ positive microglia (green) in Rbp4[Cre] animals in saline and CNO injection conditions. In CNO-injected mice, morphological features indicate that microglial response increases in a dose-dependent manner. Mice: $n = 4$ saline, $n = 5$ CNO 10 mg/kg, with an average of 3 sections per region and per mouse, evaluated using mixed-effects ANOVA via Holm–Šidak's multiple comparisons test. All data are presented as mean ± SEM, false discovery rate of 0.05 adjusted using Benjamini and Yekutieli, *$p < 0.05$, **$p < 0.01$, ***$p < 0.001$, ****$p < 0.0001$ using scale bars of **a, e, g, k, m, q** - 100 μm. Detailed statistical information is listed in Supplementary Data 1. Source data is provided as a Supplementary Data 2 file. Created with BioRender.com.

while PV + VVA+ cells were not affected and VVA+ cell densities decreased in RSP L2-3 and 5 (Supplementary Fig. 4a–c). In PTLp, no significant changes in PV + VVA−, PV + VVA+, and VVA+ cell densities were observed after CNO application (Supplementary Fig. 4d-f). In S1, there were no changes in PV + VVA− and VVA+ cell densities within different cortical layers, but PV + VVA+ cell densities increased in L4-5 after CNO application (Supplementary Fig. 4g–i). We measured PV+ cell density in different hippocampal regions and subcortical areas and found no significant changes (Supplementary Fig. 5a, b).

In Rbp4[Cre]-hM4Di, we analysed PV+ density in saline and 1 mg/kg CNO brains (Fig. 9a, b). No significant changes in PV+ cell densities were observed after CNO application in RSP, PTLp, and S1 (CNO effect: $F_{(1,2)} = 0.4246$, $p = 0.5815$, Fig. 9c). Moreover, no differences were observed in the laminar distribution of PV+ cells in RSP, PTLp, and S1 (CNO effect: RSP $F_{(1,2)} = 0.04222$, $p = 0.8562$; PTLp $F_{(1,2)} = 2.154$, $p = 0.2799$; S1 $F_{(1,2)} = 0.001787$, $p = 0.9701$; Fig. 9a–l). Furthermore, no significant changes in PV + VVA−, PV + VVA+, and VVA+ cell densities were found in RSP, PTLp and S1 (Supplementary Fig. 6a–i) or subcortical regions (Supplementary Fig. 7a, b). These results suggest that acutely activating and silencing Rbp4[Cre] neurons does not affect PV+ interneuron densities, with only minor effects on PNNs.

## Discussion
In this study, we report how acute chemogenetic manipulations of specific neuronal populations in cortical L5 and hippocampal DG rapidly influence neuron-glial interactions. Using Rbp4[Cre]-hM3Dq and Rbp4[Cre]-hM4Di mice, we show that activating and inhibiting cell-type-specific neuronal activity leads to substantial and rapid changes in neuron-glia communications. We showed that CNO alone has no influence on neuronal activity as measured by immediate early gene expression. However, previous reports describe an effect of CNO on sleep[39]. We demonstrate that acute chemogenetic activation decreases excitatory synaptic density, with silencing having the opposite effect. Additionally, acute neuronal activation rapidly increases microglia and astrocyte reactivity. Conversely, both glial cells exhibit decreased reactivity with acute neuronal silencing. We also show that, surprisingly, microglia and astrocytes rapidly respond to CNO application even without DREADDs present. However, the effect we observed with CNO in the presence of hM4Di is in a direction opposite to the effect observed with CNO alone. We therefore believe that DREADD-mediated activity itself does influence glial reactivity.

First, we selected the Rbp4[Cre] projection neuron population for manipulation because of our previous investigations on long-term silencing[28,29,31,32,40]. We crossed the Rbp4[Cre] line with a stop-floxed DREADD transgenic line for several reasons. Firstly, the timing of acute chemogenetic intervention is ideal compared to longer-term viral injection. Secondly, viral injections can only influence a relatively small population of neurons which is limited to the site of administration. Our model addresses this limitation as all Cre+ cells in genetically manipulated animals express DREADDs and thus could all be subjected to chemogenetic modulation. We used different CNO concentrations for various reasons. For hM3Dq, 10 mg/kg has been used as the default dose of CNO in similar studies[41]. For our initial experiments, we used 1 mg/kg and 10 mg/kg CNO. Both concentrations had a severe adverse effect on some animals. With 10 mg/kg CNO we observed a range of adverse effects ranging from death to a sedated, immobile state without movements of limbs. Further EEG recording would be required to study whether this immobilised posture is indicative of non-convulsive seizures. When we decreased the CNO dose to 0.1 mg/kg, adverse effects were less severe. Mice still seemed less active but were easily arousable. At 0.05 mg/kg, we still observed slower locomotion but no signs of distress. Prior studies have raised concerns about the behaviour of animals expressing DREADDs following CNO administration[42], and in this study, we suspect that adverse effects including death (on two occasions) were caused by strong receptor activation in addition to neuron and glial reactions to CNO itself. For future experiments, lower doses, or alternative actuators such as compound 21[39,43] should be considered in combination with EEG recording. No behavioural changes were observed in Rbp4[Cre]-hM4Di after CNO injections in the timeframe studied, which is consistent with results found previously[42]. We investigated changes in targeted neurons themselves. Activating Rbp4[Cre] neurons in Rbp4[Cre]-hM3Dq mice decreased synaptic density in V1 L5, but not in projection regions, while Rbp4[Cre]-hM4Di exhibited a dose-dependent increase in synaptic density in V1 L5 and CA1. We speculate that excitatory synapses try to compensate for suddenly increased neuronal activity by reducing in number. Increased synaptic density following hM4Di activation might relate to potassium channel activation and the inhibition of neurotransmitter release, which would indicate that excitatory synapses also compensate for reduced cellular activity

A very exciting observation involved rapid changes in microglial dynamics. Acute activation of neurons decreases microglial density. It is unlikely that the microglial response is directly due to CNO as higher doses of CNO in hM4Di mice had no effect. Moreover, we also demonstrated that CNO administration alone does not affect cortical microglia density in Rbp4[Cre], with only minor changes observed in CA3. Thus, we conclude that the number of Iba1+ microglia was reduced due to Rbp4[Cre] neuron activation. Previous work suggests that direct microglial activation in CX3CR1-hM3Dq mice upregulated both Iba1 expression and the number of Iba1+

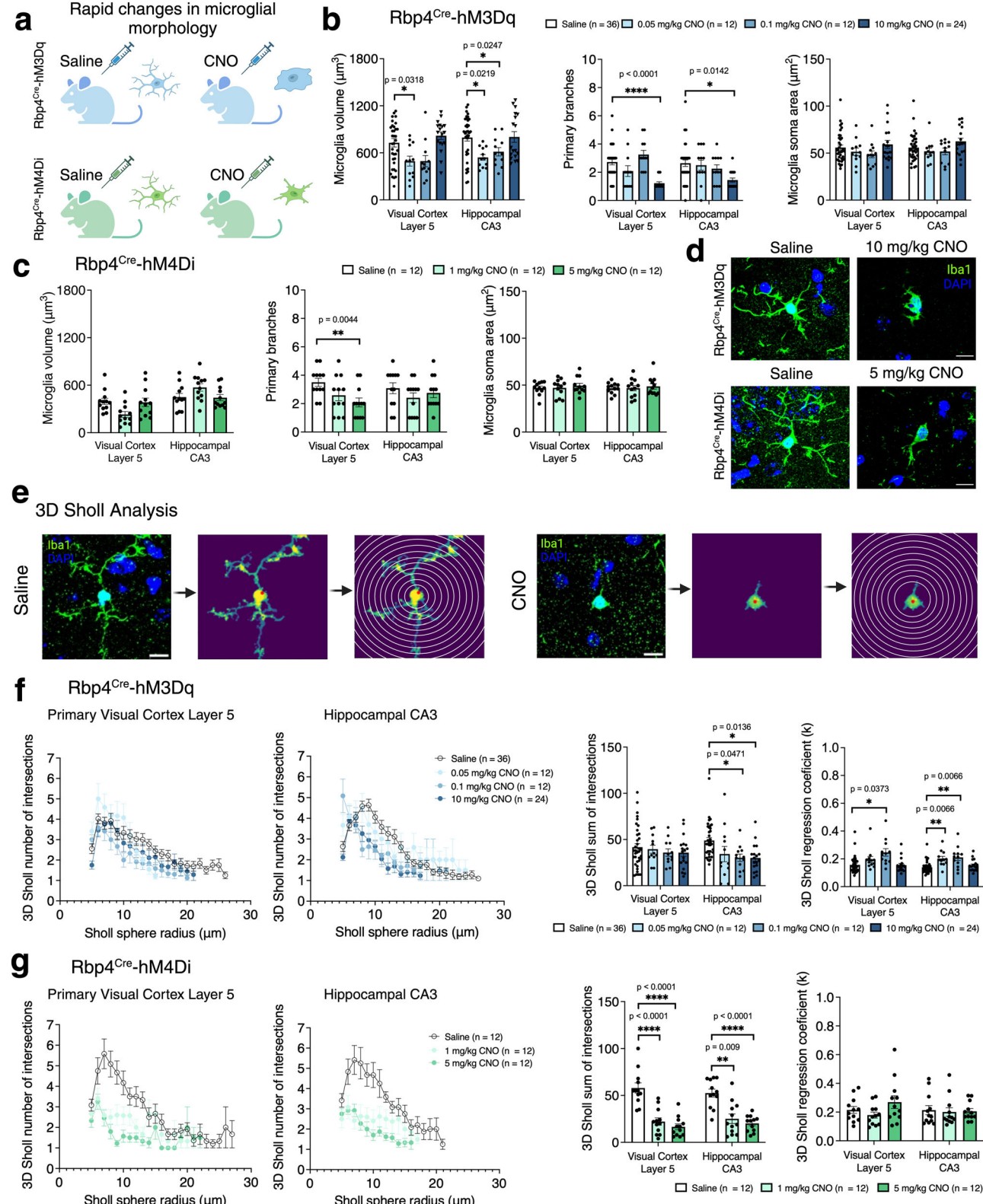

**Figure panels a–g: Rapid changes in microglial morphology; Rbp4Cre-hM3Dq and Rbp4Cre-hM4Di analyses; 3D Sholl Analysis.**

microglia[44]. Activation of nearby neurons versus microglia themselves therefore appears to have opposite effects on microglial cell density.

We tested phagosomal marker CD68 co-expression with Iba1, which is expressed at high levels in reactive microglia[41,45]. Increased microglial reactivity was only observed in subcortical regions (SC, CA1, CA3) in Rbp4Cre-hM3Dq, suggesting that microglia are particularly sensitive to

Rbp4Cre projection neuron activation. Previous studies have shown that inducing hM4Di in microglia themselves suppresses microglial reactivity[46], which aligns with our observed decrease in microglia reactivity in CA1 after silencing DG neurons. We also showed increased microglia reactivity to CNO itself in Rbp4Cre, which surprisingly indicates that microglia react to both neuronal activity and CNO. We further compared microglial cell

**Fig. 5 | Rapid changes in microglial morphology after acute chemogenetic manipulations. a** Schematic of experimental paradigms used to explore the rapid change in microglia morphology in excitatory (blue) and inhibitory (green) DREADDs after CNO injection. **b, c** In Rbp4[Cre]-hM3Dq mice (blue) microglial volume and primary branches changes yet no differences were observed in microglial soma area. Mice: $n = 4$ saline, $n = 2$ CNO 0.05 mg/kg, $n = 2$ CNO 0.1 mg/kg, $n = 3$ CNO 10 mg/kg, with 6–9 cells per animal. In Rbp4[Cre]-hM4Di mice (green) microglial volume and primary branches changes yet no differences were observed in microglial soma area. Mice: $n = 2$ saline, $n = 2$ CNO 1 mg/kg, $n = 2$ CNO 5 mg/kg, with 6–9 cells per animal. Evaluated using mixed-effects ANOVA via the corrected method of Benjamini and Yekutieli. **d** Representative images of Iba1-immunolabeled microglia in Rbp4[Cre]-hM3Dq and Rbp4[Cre]-hM4Di mice injected with saline and CNO. **e** Representative images of 3D Sholl analysis of microglial cells after CNO application. By applying Sholl sphere radii, microglia process retraction can be measured by quantifying branch intersections with each radius. **f, g** 3D Sholl analysis showed that in excitatory DREADD (blue), the number intersections were lower in CA3 but no changes in V1 L5, and regression coefficient was higher in both regions, but no dose-dependent changes were observed. Mice: $n = 4$ saline, n = 2 CNO 0.05 mg/kg, $n = 2$ CNO 0.1 mg/kg, $n = 3$ CNO 10 mg/kg, with 6–9 cells per animal. In inhibitory DREADD (green), the number intersections decreased in dose-dependent manner in both regions, but no changes in regression coefficient were observed. Mice: $n = 2$ saline, $n = 2$ CNO 1 mg/kg, $n = 2$ CNO 5 mg/kg, with 6–9 cells per animal. Evaluated using mixed-effects ANOVA via the corrected method of Benjamini and Yekutieli. All data presented as mean ± SEM, false discovery rate of 0.05 adjusted using Benjamini and Yekutieli, $*p < 0.05$, $**p < 0.01$, $***p < 0.001$, $****p < 0.0001$. Scale bars for **d** and **e** - 100 μm. Detailed statistical information is listed in Supplementary Data 1. Source data is provided as a Supplementary Data 2 file. Created with BioRender.com.

dynamics in Rbp4[Cre] and Rbp4[Cre]-hM3Dq mice injected with saline or 10 mg/kg CNO. In Rbp4[Cre] L5 and projection region to SC the high dose of CNO has no effect, whether or not the DREADD is present. Yet it affects microglial dynamics in the CA1–CA3 regions, even in the absence of DREADDs. However, we cannot confirm if changes in microglial cells observed in CA1–CA3 are the result of CNO or chemogenetic manipulation or a combination of both. Acute activation of Rbp4[Cre] neurons also strongly influences microglial morphology in V1 L5 and CA3. Acute activation reduces microglial volume in a dose-dependent manner, with the highest dose of CNO having no impact on microglia volume, only reducing the number of primary branches. In addition, Sholl analysis showed strong reduction in the number of microglial processes in CA3. Interestingly, acute silencing of neurons reduced microglia size and primary branches only in the cortex, while retraction of microglial processes was observed in microglia in both V1 and CA3. Changes in microglial distribution and dynamics might not only be influenced by manipulations of L5 neurons but also due to increased neuronal activity of other cortical layers. Our results suggest that microglia respond to neuronal activation or silencing by adapting their morphology and reactivity, and that manipulations of neuronal activity alter neuron-microglia interactions.

In addition to microglial changes, we also show that astrocytes quickly respond to acute changes in neuronal activity. Acute L5 activation leads to increased astrocyte reactivity and inflammation in the cortex. However, acute neuronal silencing does not affect astrocyte reactivity. In Rbp4[Cre], CNO administration increases astrocyte reactivity but no changes in inflammation were observed. In subcortical regions, acute activation of neurons increased astrocyte reactivity only in CA3 at the highest dose, with inflammation being absent in SC, CA1–CA3. Astrocyte reactivity was affected by acute silencing in SC and CA1. GFAP+ co-expression with S100β+ decreased in CA1, suggesting that the CA1 region was more sensitive to acute silencing. Surprisingly, and like microglia, control experiments with Rbp4[Cre] showed that CNO itself hugely affects astrocytes. Astrocyte reactivity and inflammation increase upon CNO application in all tested subcortical regions. Moreover, when quantifying the density of GFAP+ and S100β+ astrocytes in Rbp4[Cre] and Rbp4[Cre]-hM3Dq with or without CNO, we observed no changes. However, like for microglia, genotype and CNO effects on astrocytes were observed in CA1–CA3 regions. Previous studies reported that reactive astrogliosis occurs during brain injury and other pathologies[47,48] and that hM4Di activation leads to neuroinflammation[49–51]. This, and our present findings, suggest that neuron-to-astrocyte interactions might follow different mechanisms. Here we show that astrogliosis can be triggered by acute changes in neuronal activity through DREADD activation and even by CNO administration itself.

Lastly, we showed that acute chemogenetic manipulations have minor effects on PV+ cells and PNNs. It is known that changes in neuronal-activity can modify PV expression and PNNs[32], but acute activation of Rbp4[Cre] neurons seems to have no effect on PV+ expression yet decreased the density of PNNs in L2-3 and 5 and increased the proportion of PV+ cells without perineuronal nets in cortical L6a. Moreover, DREADD activation leads to increased PV+ co-localisation with perineuronal nets in L4-5 of S1. However, acute silencing of Rbp4[Cre] neurons did not result in notable differences in PV expression and PNNs.

In summary, our study revealed rapid and complex neuron-neuron and neuron-glia interactions after acute chemogenetic activation of selected cortical cell populations, as well as some concerning effects of 10 mg/kg CNO itself.

## Methods
### Breeding and maintenance
All animal experiments were approved by a local ethical review committee and conducted in accordance with the UK Animals (Scientific Procedures) Act, 1986 (ASPA), under valid personal and project licences. We have complied with all relevant ethical regulations for animal use. Animals were held in individually ventilated cages (IVCs) on a 12-hour light/dark cycle in the Biomedical Sciences Building (BSB), Oxford. Water and food were given *ad libitum*. Both, male and female mice were used (Supplementary Table 4). To understand the effect of acute activation and silencing on layer 5 neurons, we used a chemogenetic approach and the layer 5 driver line Rbp4[Cre28,29]. We crossed the layer 5 driver mouse expressing Cre recombinase Tg(Rbp4-cre) KL100Gsat/Mmucd (Rbp4-Cre) with the inhibitory DREADD line B6.129-Gt(ROSA)26Sortm1(CAG-CHRM4*,-mCitrine)Ute/J (The Jackson Laboratory, No: 026219) to generate recombinant 'inhibitory DREADD' mice abbreviated as Rbp4[Cre];hM4Di and with the excitatory DREADD line B6N;129-Tg(CAG-CHR3*, mCitrine)1Ute/J (The Jackson Laboratory, No: 026220) to generate recombinant 'excitatory DREADD' mice abbreviated as Rbp4[Cre];hM3Dq. In the inhibitory DREADD line, previously, the excitatory DREADD receptor was designed as a targeted insertion into the *Gt(ROSA)26Sor* locus. However, in 2017, The Jackson Laboratory reported and confirmed a randomly integrated construct CAG-LSL-hM3Dq-pta-mCitrine instead of *Gt(ROSA)26Sor* locus and that the random insertion has no effect on the functionality of the allele. All lines used were on C57BL/6 background. All genotyping was performed by Transnetyx. Immunohistochemistry, imaging, and analysis were carried out blind to the animal genotype and condition.

### CNO preparation
Clozapine-N-oxide dihydrochloride (Torcis, 6329) was used to prepare CNO solution. CNO solution was prepared under sterile conditions and passed through a Millipore filter. CNO was kept at -20°C and defrosted on the day of use and diluted using sterile saline.

### Electrophysiology
Adult mice aged 41 ± 16d ($n = 7$) were used for this section of the study. After induction of terminal anaesthesia with 5% Isoflurane (Zoetis, 50019100), brains were retrieved and sliced to 300–400 μm vibratome (Leica VT 1200 s) sections in ice-cold artificial CSF (ACSF). Brain sections were left for incubation in ACSF for one hour before recording procedures.

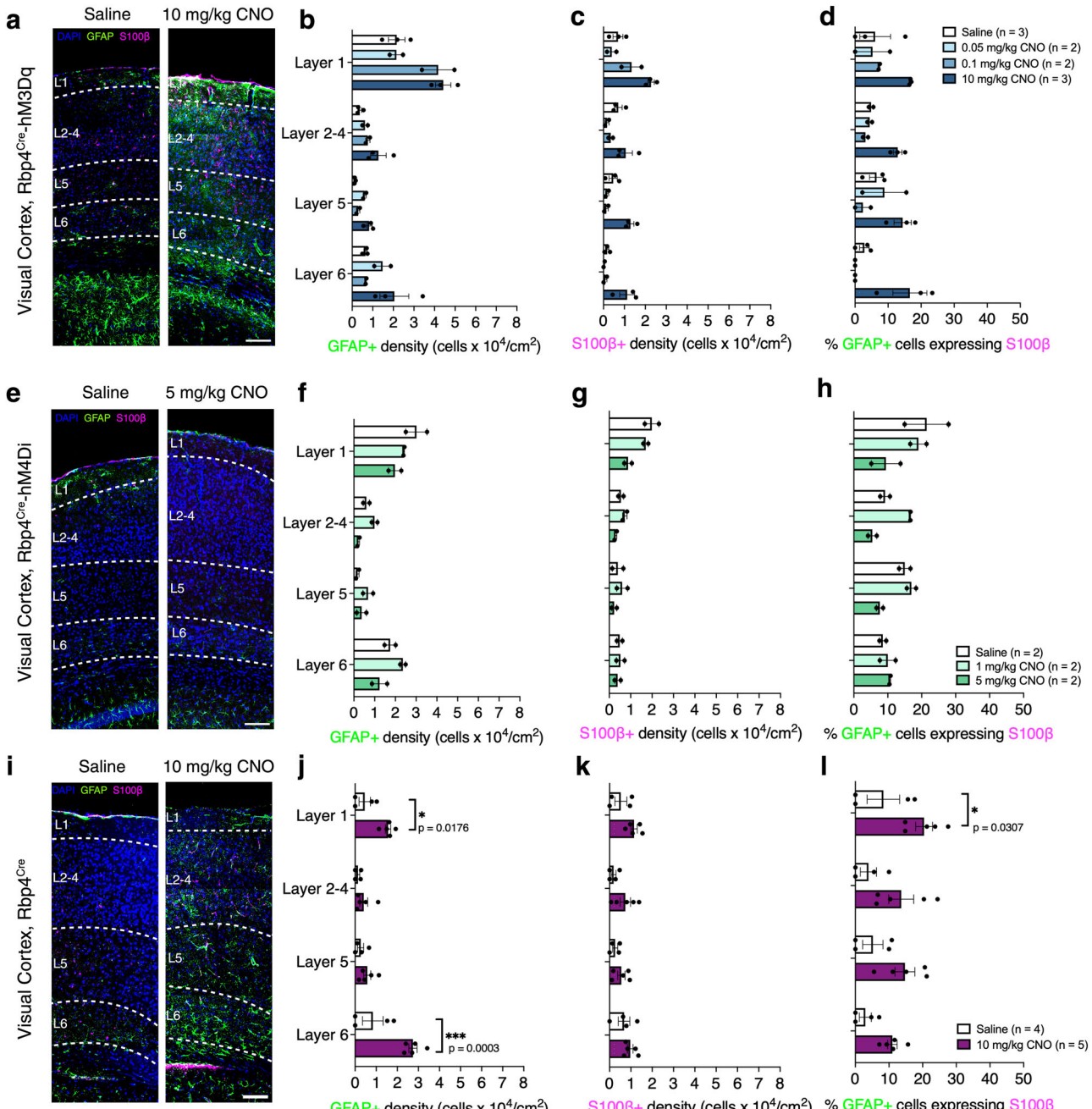

**Fig. 6 | Acute chemogenetic manipulations of neurons affect astrocytic responses to CNO in cortex. a** Representative images of GFAP (green) and S100β (magenta) immunolabeled astrocytes in Rbp4Cre-hM3Dq mice injected with saline and CNO. **b–d** GFAP+, S100β+ and GFAP+ cells that also express S100β in the primary visual cortex. Mice: *n* = 4 saline, *n* = 2 CNO 0.05 mg/kg, *n* = 2 CNO 0.1 mg/kg, *n* = 3 CNO 10 mg/kg, with an average of 3 sections per region and per mouse, evaluated using mixed-effects ANOVA via the corrected method of Benjamini and Yekutieli. **e** Representative images of GFAP (green) and S100β (magenta) immunolabeled astrocytes in Rbp4Cre-hM4Di mice injected with saline and CNO. **f–h** No changes in the density of GFAP+, S100β+ and GFAP+ cells that also express S100β in the primary visual cortex. Mice: *n* = 2 saline, *n* = 2 CNO 1 mg/kg, *n* = 2 CNO 5 mg/kg, with an average of 3 sections per region and per mouse, evaluated using mixed-

effects ANOVA via the corrected method of Benjamini and Yekutieli.
**i** Representative images of GFAP (green) and S100β (magenta) immunolabeled astrocytes in Rbp4Cre mice injected with saline and CNO. **j–l** Increased density of GFAP+ and GFAP+ cells that also express S100β but no changes in S100β+ in the primary visual cortex. Mice: *n* = 4 saline, *n* = 5 CNO 10 mg/kg, with an average of 3 sections per region and per mouse, evaluated using mixed-effects ANOVA via Holm–Šidák's multiple comparisons test. All data are presented as mean ± SEM, false discovery rate of 0.05 adjusted using Benjamini and Yekutieli, *$p < 0.05$, **$p < 0.01$, ***$p < 0.001$, ****$p < 0.0001$. Scale bars for **a, e, i** - 100 µm. Detailed statistical information is listed in Supplementary Data 1. Source data is provided as a Supplementary Data 2 file. Created with BioRender.com.

Recording consisted in whole-cell patch-clamp experiments using a borosilicate patch-clamp pipette containing intracellular potassium gluconate recording solution and 0.5% biocytin solution for post-hoc neuron identification and morphological reconstructions. Patch-Clamp experiments

were performed using a MultiClamp 700B microelectrode amplifier using pClamp Software for voltage measurement. The recording procedure consisted of increasing step-current injections ranging from −140 pA to 240 pA with 20 pA step increments. Data analysis was done using custom-made

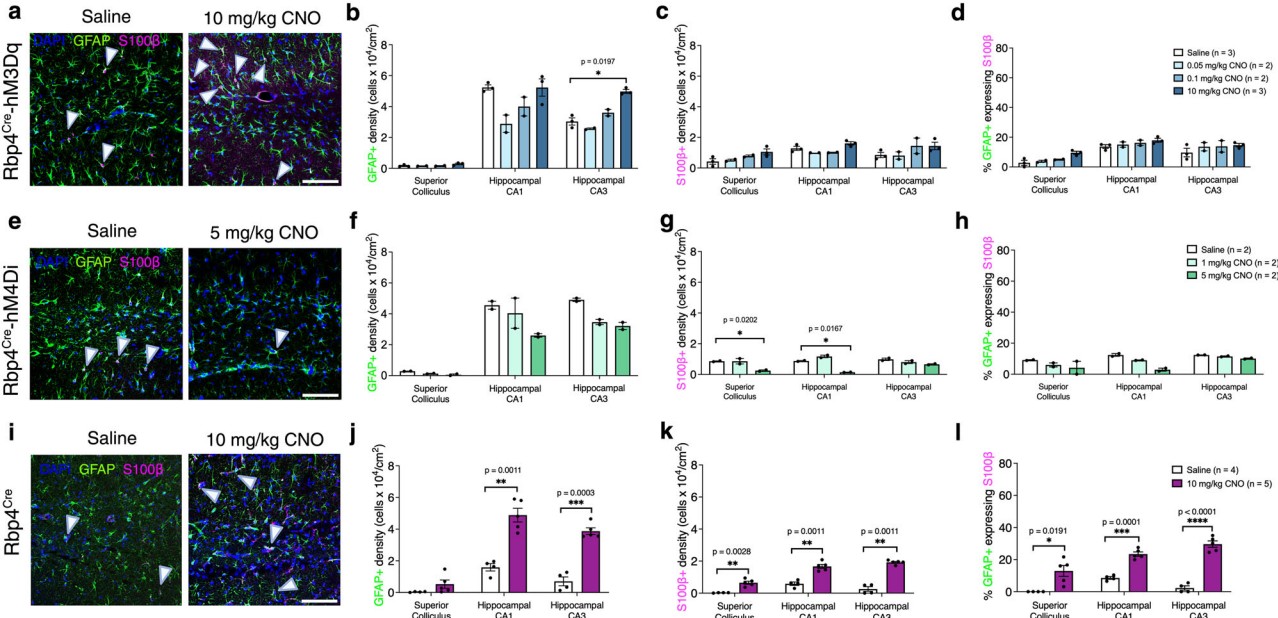

**Fig. 7 | Rapid changes of astrocytes response after acute chemogenetic manipulations in subcortical regions. a** Representative images of GFAP (green) and S100β (magenta) immunolabeled astrocytes in Rbp4$^{Cre}$-hM3Dq mice injected with saline and CNO. **b–d** Changes in GFAP+, S100β+ and GFAP+ cells that also express S100β in SC, CA1 and CA3. Mice: $n = 4$ saline, $n = 2$ CNO 0.05 mg/kg, $n = 2$ CNO 0.1 mg/kg, $n = 3$ CNO 10 mg/kg, with an average of 3 sections per region and per mouse, evaluated using mixed-effects ANOVA via the corrected method of Benjamini and Yekutieli. **e** Representative images of GFAP (green) and S100β (magenta) immunolabeled astrocytes in subcortical regions of Rbp4$^{Cre}$-hM4Di mice injected with saline and CNO. **f–h** Decrease in S100β+ density but no changes in GFAP+ density and GFAP+ cells that also express S100β in SC, CA1 and CA3. Mice: $n = 2$ saline, $n = 2$ CNO 1 mg/kg, $n = 2$ CNO 5 mg/kg, with an average of

3 sections per region and per mouse, evaluated using mixed-effects ANOVA via the corrected method of Benjamini and Yekutieli. **i** Representative images of GFAP (green) and S100β (magenta) immunolabeled astrocytes in subcortical regions of Rbp4$^{Cre}$ mice injected with saline and CNO. **j–l** Increase in GFAP+ and S100β+ density and GFAP+ cells that also express S100β in SC, CA1 and CA3. Mice: $n = 4$ saline, $n = 5$ CNO 10 mg/kg, with an average of 3 sections per region and per mouse, evaluated using mixed-effects ANOVA via Holm–Šidak's multiple comparisons test. All data are presented as mean ± SEM, false discovery rate of 0.05 adjusted using Benjamini and Yekutieli, *$p < 0.05$, **$p < 0.01$, ***$p < 0.001$, ****$p < 0.0001$. Scale bars for **a, e, i** - 100 μm. Detailed statistical information is listed in Supplementary Data 1. Source data is provided as a Supplementary Data 2 file. Created with BioRender.com.

---

Igor scripts for feature extraction, and a combination of Python and GraphPad Prism for statistical analysis (https://doi.org/10.6084/m9.figshare.26963674).

## Tissue collection

Adult mice were anaesthetised with 0.6 mL/kg Pentobarbital administered via intraperitoneal (IP) injection. Anaesthesia was confirmed by a pedal reflex test before perfusion. Animals were transcardially perfused with 0.1 M phosphate buffered saline (PBS, pH 7.4) and 4% formaldehyde (PFA diluted in PBS, F8775; Sigma Aldrich). Afterwards, brains were removed and post-fixed in 4% PFA overnight at 4 °C. Perfused brains were transferred to 1x PBS with 0.05% sodium azide (26628-22-8; Sigma Aldrich) the following day and stored at 4 °C until further usage. PFA-fixed brains were embedded in 5% agarose (Bioline, DM50-113D) and cut with a vibrating microtome (VT1000S, Leica Systems) into 50 μm thick coronal sections.

## Immunohistochemistry

To investigate immediate early gene cFos, microglia and astrocytes density, CD68 and S100β markers, and synaptic density, brain sections of Rbp4$^{Cre}$-hM4Di, Rbp4$^{Cre}$-hM3Dq and Rbp4$^{Cre}$ mice were blocked with 10% normal goat serum (Sigma Aldrich) and 0.3% TritonX-100 in PBS (blocking solution, 2 h at RT) and incubated with one of the following primary antibody combinations (Supplementary Table 1): rabbit anti-cFos; rabbit anti-Iba1 and mouse anti-CD68; mouse anti-PSD95 and guinea pig anti-vGlut1; rabbit anti-GFAP and mouse anti-S100β. Primary antibody combinations underwent overnight incubation, for vGlut1 and PSD95 underwent 48 h incubation at 4 °C. Sections were then washed in 0.1 M PBS before incubating with secondary antibodies (goat anti-rabbit A488, goat anti-mouse A488, goat anti-guinea pig A633) (Supplementary Table 1) in blocking

solution at room temperature for 2 h. Immunolabeled sections were counterstained with DAPI, mounted with FluorSave™ (Merck Millipore). For interneurons and perineuronal nets analysis, the same brain sections were blocked with blocked with 2% normal donkey serum (Sigma Aldrich) and 0.2% TritonX-100 in PBS (blocking solution, 2 h at RT) and incubated with one of the following primary antibody combination (Supplementary Table 1): rabbit anti-PV and biotinylated VVA at 2 ug/ml diluted in VVA blocking solution (VVA requires a special buffer containing CaCl2 and the antibodies are therefore not diluted in simply 0.1 M PBS). Primary antibody combinations underwent overnight incubation at 4 °C. Sections were then washed in 0.1 M PBS before incubating with secondary antibodies (donkey anti-rabbit A488 and Cy5 streptavidin-conjugated) in blocking solution at room temperature for 2 h (Supplementary Table 1). Immunolabeled sections were counterstained with DAPI, mounted with Prolong Glass (Invitrogen) and imaged as described below.

## Laser scanning confocal microscopy

Imaging and analysis were carried out blind to the animal genotype and condition. To evaluate glial cell density, S100β+ cell density and microglial activity in visual cortex layer 5 ($n = 3$ images/animal), superior colliculus ($n = 6$ images/animal) and hippocampus CA1 and CA3 regions ($n = 3$ images/animal), immunolabeled sections were imaged with a laser-scanning confocal microscope (Zeiss LSM710) using 20×/0.6 air objective and 1× optical zoom at 0.42 μm pixel size, and frame size 1024 ×1024. Image size for cortex was 424.7 ×1275.29 μm, and for superior colliculus and hippocampus CA1 and CA3: 424.7 ×424.7 μm. To examine individual microglial cell morphological features and vGlut1/PSD95 synaptic density in visual cortex layer 5 ($n = 9$ images/animal) and hippocampus CA3 region ($n = 9$ images/animal) the same immunolabeled sections were imaged with

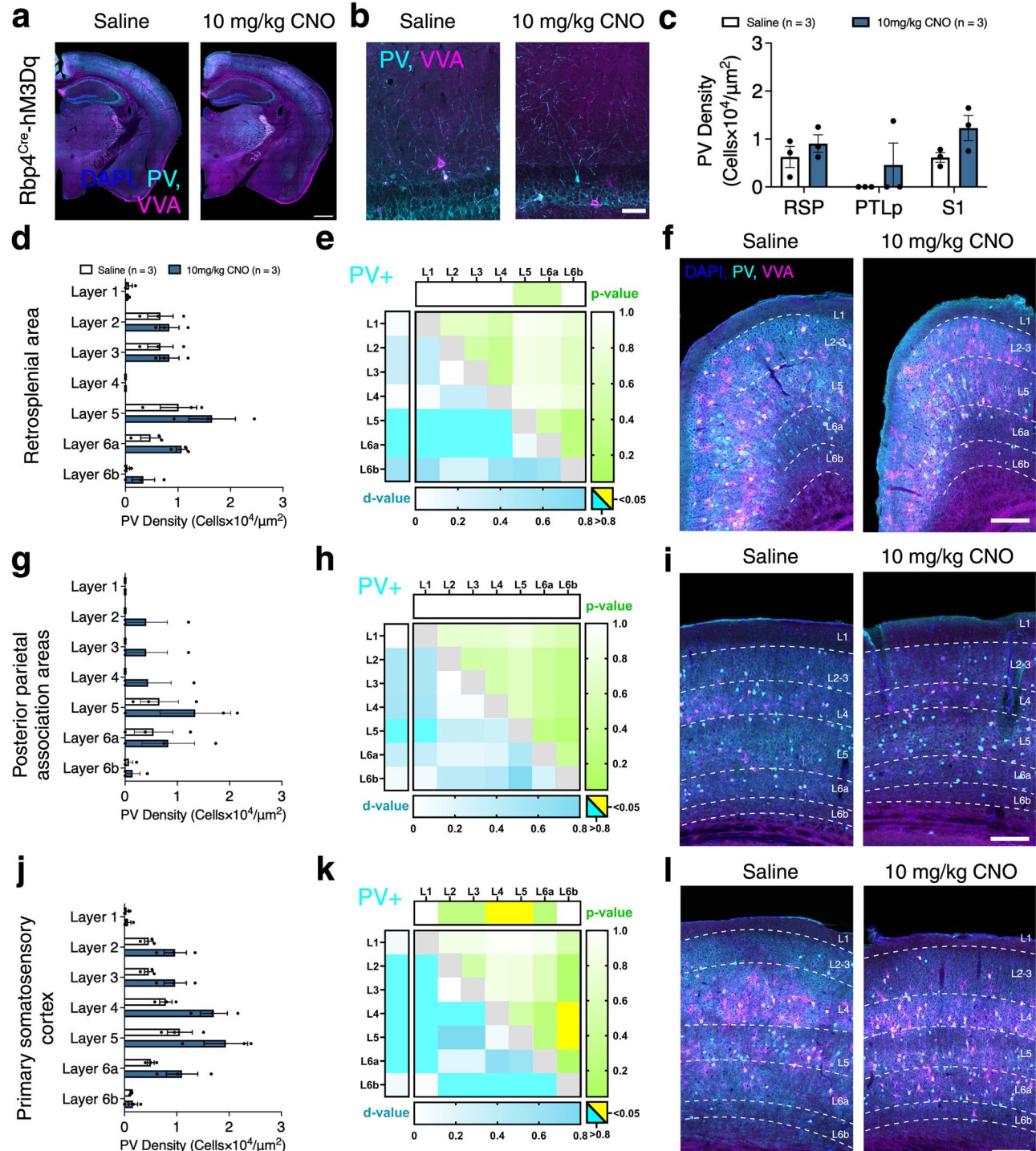

**Fig. 8 | No changes in PV+ interneuron densities in Rbp4[Cre]-hM3Dq mice.**
**a, b** Representative images of PV (cyan) and VVA (magenta) immunolabeling in Rbp4[Cre]-hM3Dq mice injected with saline and 10 mg/kg CNO. **c** Changes observed in PV+ interneuron density upon CNO administration in different cortical areas using $n = 3$ mice/dose, with 3 coronal sections per region per mouse, evaluated using mixed-effects ANOVA via the corrected method of Benjamini and Yekutieli. **d, g, j** Laminar distribution of PV+ interneurons in the retrosplenial cortex, posterior parietal association areas, and the primary somatosensory cortex using $n = 3$ mice/dose, with 3 coronal sections per region per mouse, evaluated using mixed-effects ANOVA via the corrected method of Benjamini and Yekutieli. **e, h, k** Heat maps illustrate the effects of CNO administration within and between brain regions on PV+ interneuron densities. Shades of blue indicate effect size as Cohen's d-values (cyan for large effect sizes >0.8), and shades of green indicate statistical significance

as $p$-values (yellow for $p < 0.05$). Horizontal bars indicate the overall within-region/layer statistical significance of CNO administration evaluated using two-way ANOVA and vertical bars indicate the overall within-region/layer effect size of CNO administration evaluated using Cohen's d. Pairwise comparisons between cortical layers/regions are represented in the body of each heat map, with statistical significance calculated by applying the Student's $t$ test to outcomes of the two-stage step-up method of Benjamini, Krieger and Yekutieli and effect size calculated via Cohen's d. **f, j, l** Representative images of excitatory DREADD RSP, PTLp and S1. All data are presented as mean ± SEM, *$p < 0.05$, **$p < 0.01$, ***$p < 0.001$, ****$p < 0.0001$. Scale bars are presented as **a**, - 1000 µm and **f, i, l**, - 100 µm. Detailed statistical information is listed in Supplementary Data 1. Source data is provided as a Supplementary Data 2 file. Created with BioRender.com.

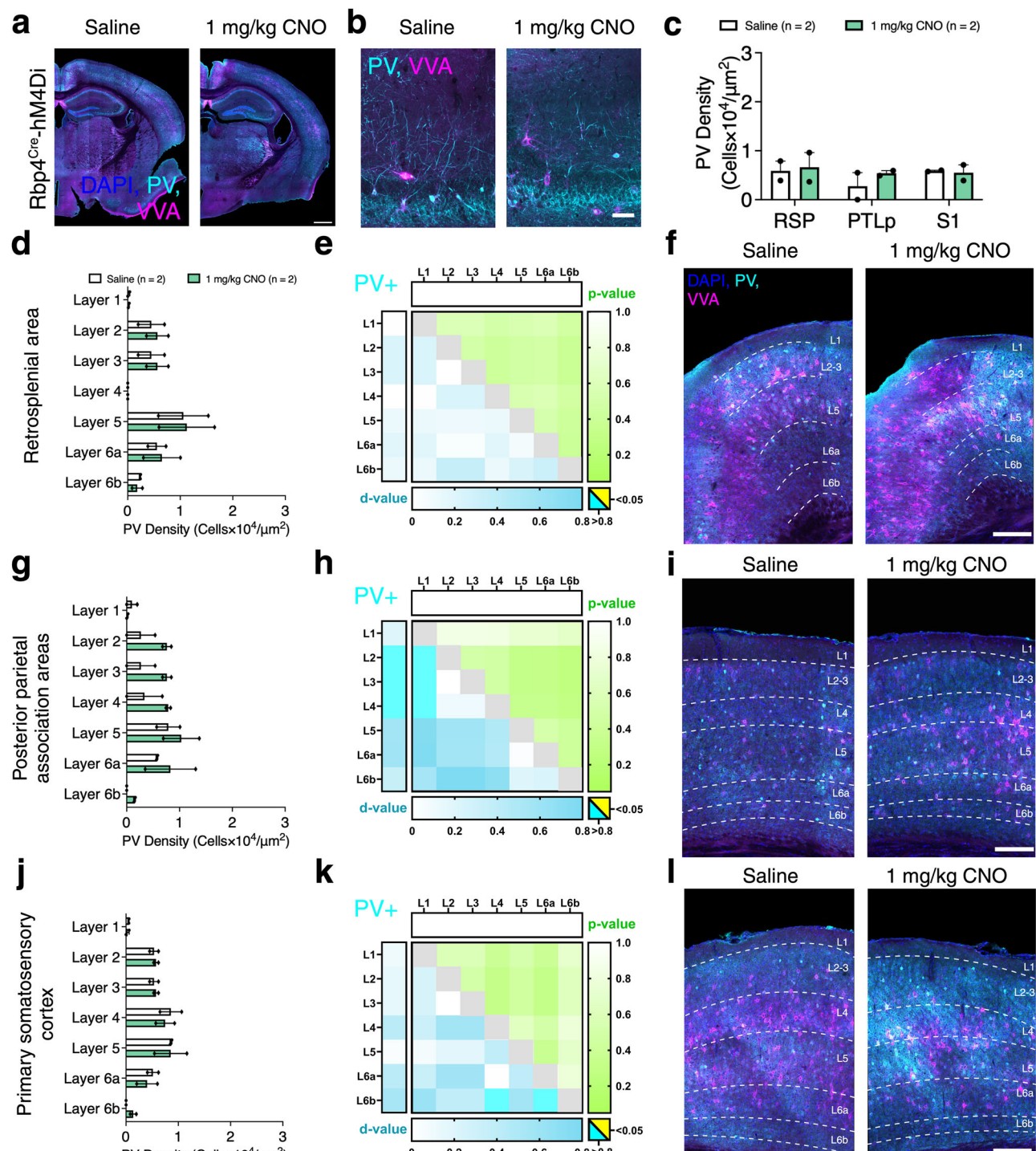

**Fig. 9 | Acute silencing has no impact of cortical PV+ interneurons in Rbp4[Cre]-hM4Di mice. a, b** Representative images of PV (cyan) and VVA (magenta) immunolabeling in Rbp4[Cre]-hM4Di mice injected with saline and 1 mg/kg CNO. **c** Changes observed in PV+ interneuron density upon CNO administration in different cortical areas using n = 2 mice/dose, with 3 coronal sections per region per mouse, evaluated using mixed-effects ANOVA via the corrected method of Benjamini and Yekutieli. **d, g, j** Laminar distribution of PV+ interneurons in the retro-splenial cortex, posterior parietal association areas, and the primary somatosensory cortex using n = 2 mice/dose, with 3 coronal sections per region per mouse, evaluated using mixed-effects ANOVA via the corrected method of Benjamini and Yekutieli. **e, h, k** Heat maps illustrate the effects of CNO administration within and between brain regions on PV+ (top left) interneuron densities. Shades of blue indicate effect size as Cohen's d-values (cyan for large effect sizes >0.8), and shades of

green indicate statistical significance as p-values (yellow for p < 0.05). Horizontal bars indicate the overall within-region/layer statistical significance of CNO administration evaluated using two-way ANOVA and vertical bars indicate the overall within-region/layer effect size of CNO administration evaluated using Cohen's d. Pairwise comparisons between cortical layers/regions are represented in the body of each heat map, with statistical significance calculated by applying the Student's t test to outcomes of the two-stage step-up method of Benjamini, Krieger and Yekutieli and effect size calculated via Cohen's d. **f, j, l** Representative images of excitatory DREADD RSP, PTLp and S1. All data are presented as mean ± SEM, *p < 0.05, **p < 0.01, ***p < 0.001, ****p < 0.0001. Scale bars are presented as **a**, - 1000 μm and **f, i, l**, - 100 μm. Detailed statistical information is listed in Supplementary Data 1. Source data is provided as a Supplementary Data 2 file. Created with BioRender.com.

laser-scanning confocal (Zeiss LSM710) using 63×/1.4NA oil-immersion objective and 2× optical zoom.

## Spinning disk confocal microscopy

Interneuron cell densities were assessed using spinning disk confocal microscopy (Olympus SpinSR SoRa, Yokogawa CSU-W1 SoRa unit B525/50 μm pinhole disc at 1600 rpm with glass camera port and aperture lens of 18.5 mm). Full-coronal images were obtained using 20×/0.8 air objective and 1× SD-MGCA magnification changer at 0.325 μm pixel size, with variable total frame dimensions depending on the anatomical shape of the tissue (around 30,000 × 20,000, or 0.975 × 0.650 cm). Three z-stacks were acquired from the tissue-slide interface, moving towards the coverslip at a step size of 7.49 μm. Lasers used include 405 nm at 10% with an exposure time of 50 ms, 488 nm at 40% with an exposure time of 100 ms, and 640 nm at 40% with an exposure time of 100 ms. The IX3 halogen lamp was set to 3 V, the aperture stop to 75%, and the DIC slider to 2950. Exposure time was set to 50 ms with a pixel clock of 480 MHz and a resolution of 2304 × 2304 (sub-array of 1500 × 1500 to avoid quilting). cellSens software was used for microscope operation and to apply high-density focus maps. Eight additional points were initialised (bilaterally at S1, the hippocampus, the striatum, and the globus pallidus) for each coronal section to furthermore improve focus maps in regions of interest. Imaging and analyses were conducted blind to animal genotype.

## Cell counter

To evaluate immediate early gene cFos and microglial cell density, GFAP+ and S100β+ cells density, we created an automatic cell counter program with cell overlapping nuclei (overlap between DAPI (nucleus) and Iba1) and based on signal intensity only for cFos, GFAP and S100β to count the total number of Iba1+, cFos+, GFAP+ and S100β+ per field of view. The image processing was done using Python language package Scikit-Image, in particular the *skimage.morphology* module to find, filter and count connected volumes[52]. Image filtering operations (Gaussian filter) were performed using skimage.filters module (https://doi.org/10.6084/m9.figshare.26963674).

## Automatic 3D microglia volume counter

To visualise 3D microglial cells, anti-Iba1 immunohistochemical labelling was used. Iba1+ signal volume was computed by counting all voxels in the connected element (this was done by using the morphology module of the skimage (Python package[52]) corresponding to the cell. Connected element segmentation was done by using 26 neighbour's connectivity (that is the voxels are connected if their edges, corners, or faces touch). Since small ($>0.7 \mu m^3$, ~ 300 voxels) connected, volumes cannot correspond to a whole cell, connected regions with smaller voxel count than 300 and considered to be noise are removed. Voxel depth and pixel size was taken from laser-scanning confocal microscope (Zeiss LSM710) image acquisition settings (voxel depth z = 0.55 μm, pixel size x = y = 0.0658941 μm). Iba1 channel was smoothed using Gaussian kernels (with standard deviation of 2 voxels) and binarized using intensity threshold:

$$I_{th} = 2\sigma = 2\sqrt{\sum_{x,y,z} \frac{\left(I_{x,y,z} - \bar{I}\right)^2}{N}}$$

where x, y, z - intensity channel 3D coordinates, N - voxel count, where σ is standard deviation of the channel 3D matrix, $\underline{I}$ - average intensity (https://doi.org/10.6084/m9.figshare.26963674).

## 3D Sholl analysis

This analysis was performed with the following steps: 1) by applying 3D Gaussian blur to reduce high spatial frequency noise, 2) thresholding the image to obtain a 3D segmentation mask of the cell, 3) removing small connected components, 4) generating binary masks of a fixed radius and thickness of x pixels, 5) multiplication of spherical shell mask with 3D mask

of the cell (the sphere is centred in the centre of mass of the cell or nucleus of the cell). Next, we counted the connected components in spherical shell mask intersection with the mask of the cell and repetition of steps 4–6 with increasing radius of the spherical shell mask results in radial distribution of spherical shell intersections. Lastly, we mapped the intersections distribution to: $y = \ln\ln\left(\frac{N_{intersections}}{V_{sphere}}\right)$; $x = r_{sphere}$ and fitting a linear regression model $\hat{y} = m - kr$ we obtain the 3D Sholl regression coefficient k (https://doi.org/10.6084/m9.figshare.26963674).

## Manual analysis of microglia

Analysis of microglial soma and primary branches, CD68, S100β colocalization with GFAP and synaptic density was performed using ImageJ (NIH). Somata were manually measured using a polygon selection tool. Primary branches were manually quantified based on visual investigation. Primary branches are described as the main branches that are directly connected to the soma of microglia. Iba1+ and CD68+, and GFAP+ and S100β+ co-localization was manually counted[66]. To count the percentage of CD68+/Iba1+ microglia, and S100β+/GFAP+ astrocytes, CD68+ and S100β+ cells were divided by total number of Iba1+ microglia and GFAP+ astrocytes respectively per field of view and multiplied by 100.

## Synaptic density

vGlut1 and PSD95 particle number and volume were analysed with ImageJ software (NIH) using 3D ROI manager after subtracting background to remove spatial variations of the background intensities for 3D particle reconstruction, applying Gaussian blur 3D smoothing filter (x sigma = 2, y sigma = 2, z sigma = 2) and manual thresholding. Afterwards, data was normalised based on number of planes in image and image volume. Colocalization between PSD95 and vGlut1 particles was quantified using the ImageJ JACoP plugin. All analysis and calculations are summarised in the reference guide provided for JACoP plugin: https://imagej.net/plugins/jacop and additional associated references[53].

## Interneuron density

Interneuron densities were assessed using a modified version of the QUINT pipeline[54]. This broadly involved image pre-processing, cell detection, atlas registration, atlas mapping, and output reformatting for visualisation and statistical analysis.

Following image acquisition, raw vsi files were first pre-processed using a customised ijm script in Fiji/ImageJ (2.9.0 v1.54b). Coronal sections were imported at a downsampled factor of 4 (Series 3) with split channels followed by maximum-intensity z-stack projections, auto-brightness thresholding, and background subtraction. LUTs were reassigned before being merged and exported as a tiff composite image.

.tiff outputs were imported into QuPath (v0.4.2)[55]. where a custom groovy script conducted automated cell detection and image export in two separate formats: the first for the results of cell detections (used instead of ilastik in QUINT's "Feature Extraction" workflow), and the second for composite image compatibility with atlas mapping software (QUINT's "Atlas-Registration" workflow). Four sets of cell detections were performed for fluorophore combinations corresponding to 1) all cells with PV signal (PV+), 2) all cells with VVA signal (VVA+), 3) cells with both PV and VVA signal (PV+/VVA+), and 4) cells with PV signal but no VVA signal (PV+/VVA−) (Supplementary Tables 2 and 3 for cell detection and object classification parameters respectively). Detections were visualised with filled areas and without the underlying composite coronal image, then exported as a svg file. A short custom Fiji/ImageJ ijm script converted svg files to a binary mask, thereby creating a one-pixel outline around adjacent cell detections. Images were inverted such that cell detections presented as black on a white background for subsequent conversion and export as RGB png files. Within QuPath, composite images were additionally saved as RGB png files at a downsampled factor of 8 (32 overall when also accounting for 4x downsampling during Fiji/ImageJ pre-processing) for compatibility with the requirements of the QuickNII atlas registration software (RRID:SCR_016854)[56].

Downsampled png composite images from QuPath were imported into QuickNII's FileBuilder and merged to xml files for each animal (i.e., including all technical replicate sections from the same brain). xml files were opened in QuickNII such that each coronal section could be aligned and registered to the Allen Mouse Common Coordinate Framework 3D reference atlas (2017 CCFv3)[57]. Registrations were exported as json files, then imported into VisuAlign software for nonlinear refinement (v0.8 RRID:SCR_017978). Refined VisuAlign json anchoring files and flat brain atlas maps were saved as the final product of atlas registration.

Atlas mapping was then conducted by importing QuPath/Fiji/ImageJ cell detection png images, VisuAlign json anchoring files, and VisuAlign flat atlas maps into Nutil software[58]. Custom brain regions were defined using the QUINT template and incorporated into the Nutil Quantifier alongside png, json, and flat files. All object reports were generated with a minimum object size of 8 pixels, pixel scale of 1.69 pixels/$\mu m^2$ the detection colour set to black, a point cloud density of 4, with custom masks in cases of bilateral discrepancy in the coronal plane, without object splitting, and extracting all coordinates for 3D registration. The Nutil Quantifier was run for each brain, generating output 3D point cloud json files and csv object reports. Point cloud json files were directly visualised using MeshView software (ABA Mouse CCFv3 2017 25um RRID:SCR_017222) while csv reports required re-formatting for further quantification.

A custom MATLAB script (R2022b, MathWorks, Natick, MA, USA) extracted and re-formatted Nutil outputs for each brain section for each animal of each genotype and DREADD injection condition for region-specific quantification. This custom script (quint_postprocessing_subdivisions.m) has been deposited on (https://doi.org/10.6084/m9.figshare.26963674). Re-formatted data was imported into GraphPad Prism for statistical analysis and graphical representation (v.9.3.1, GraphPad Software, San Diego, CA, USA).

## Statistics and reproducibility

Data are reported as means ± standard error of the mean (SEM). Microglial cells and astrocytes analyses were performed on data from at least 2 brains per group and for each analysis a minimum of 3 sections per brain and injected condition for each DREADD. Results were averaged over technical replicates (visual cortex: 3 images/animal, superior colliculus: 6 images/animal, hippocampus CA3 region: 3 images/animal, 3D data of layer 5 and CA3 area: 6–9 images/animal). Interneuron densities were computed for each brain with 1–3 technical replicate sections per animal, genotype, and DREADD injection condition (1–2 for Rbp4[Cre] saline controls; 3 for Rbp4[Cre]-hM4Di 5 mg/kg CNO administration; 1–3 for Rbp4[Cre]-hM4Dq saline controls; and 1–2 for Rbp4[Cre]-hM4Dq 10 mg/kg CNO administration - see the Source Data file for more details). Statistical analyses were conducted using GraphPad Prism. To compare differences of neuronal activity after CNO application, we used mixed-effects ANOVA with Šidák's multiple comparisons test or one-way ANOVA via Dunnet's multiple comparisons test and these tests were used for analysis of electrophysiology data. For any single comparison between two groups, we used an unpaired Student's $t$ test for normally distributed data, and Mann–Whitney for not normally distributed data. Normality was verified using the Shapiro–Wilk test (assessed by QQ plot). Comparison between different CNO concentrations was performed using mixed-effects ANOVA via Benjamini and Yekutieli multiple comparisons test for Rbp4[Cre]-hM3Dq and Rbp4[Cre]-hM4Di, and mixed-effects ANOVA via Holm-Šidák's multiple comparisons test for Rbp4[Cre], these tests were used for analysis of microglial density and astrocyte density in the primary visual cortex with different concentrations of CNO. Comparison between different brain regions and CNO concentrations was performed using linear mixed-effects modelling, with brain region as a within-subjects factor and CNO concentration as a between-subjects factor. Models were fitted using the restricted maximum likelihood method and included a Geisser-Greenhouse correction for sphericity. *Post hoc* testing was used where appropriate to compare CNO groups against the control (saline) group for each brain region, with a Benjamini–Yekutieli correction to control the false discovery rate at 5%. This strategy was used for analysis of the density, morphology and co-localisation of immediate early genes, glial cells densities, microglial morphology, and synaptic and interneuron densities. *p*-values are reported after Benjamini–Yekutieli adjustment; for uncorrected *p*-values see Supplementary material. Mixed-effects ANOVA via Holm-Šidák's multiple comparisons test was used for Rbp4[Cre] control experiments for microglial density and astrocyte density. A custom MATLAB script was used to re-format outputted Prism statistics (prism_anova_reformatting_sidak.m) and is available on (https://doi.org/10.6084/m9.figshare.26963674).

Interneuron density assessments were repeated for combinations of PV and VVA expression/co-expression including PV+ (PV, irrespective of VVA), VVA+ (VVA, irrespective of PV), PV+/VVA+ (PV and VVA co-expression), and PV+/VVA− (PV and a lack of VVA expression). To additionally evaluate pairwise differences in PV/VVA expression/co-expression between brain regions (i.e., cortical layers, hippocampal subdivision, or other subcortical subdivisions), the Student's $t$ test was applied to outcomes of the two-stage step-up method of Benjamini, Krieger and Yekutieli (controlling for false discovery rate). To supplement this, the effect size of both the main influence of CNO administration and pairwise regional differences were evaluated using Cohen's d. Interneuron assessments furthermore included three-way ANOVA with CNO administration (saline vs CNO), brain region (e.g., cortical layer or subsets of subcortical areas), and VVA co-expression (i.e., VVA+ vs VVA−) defined as main factors. A custom MATLAB script (prism_anova_reformatting_sidak.m) was written to conduct these calculations and is available (https://doi.org/10.6084/m9.figshare.26963674). The threshold for statistical significance was set at $p < 0.05$.

All figures were created with BioRender.com

## Reporting summary
Further information on research design is available in the Nature Portfolio Reporting Summary linked to this article.

## Data availability
A sample dataset and electrophysiological recordings, and results used to generate key analyses presented in this paper is available on Figshare (https://doi.org/10.6084/m9.figshare.26963638). Supplementary data 1 (Supplementary Statistics) and 2 (Source data) provided with this paper. Images are available from the corresponding authors upon reasonable request.

## Code availability
Custom-made codes for data analysis are deposited on Figshare (https://doi.org/10.6084/m9.figshare.26963674).

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

## Acknowledgements
A.V.'s graduate studies were funded by the State Study Foundation of Lithuania, Goodger and Schorstein Scholarship from University of Oxford; A.V. is currently funded through a Junior Research Fellowship with St John's College, Oxford. E.M.'s graduate studies were funded through the O'Sullivan Family Graduate Scholarship. R.T. has been funded through European Union's Horizon 2020 research and innovation program under the Marie Skłodowska-Curie grant agreement No 890457-L6b (RT, EOM, ZM). M.M. is supported through a Rhodes Scholarship; F.S. is supported from an Anatomical Society Graduate Studentship awarded to A.H.S. and Z.M., and Goodger and Schorstein Scholarship from University of Oxford; F.M. is a postdoctoral Fellow supported by an MRC Project Grant (G00900901 Z.M., E.M.). L.B.K. was supported by Wellcome Trust PhD studentship 203971/Z/16/Z; Part of V.V. salary is supported by MRC/S01134X/1; The work in Z.M.'s laboratory was supported by Research Grants from St John's College Research Centre No 21138077 (A.V., Z.M.). Z.M. is also associated to Charité-Universitätsmedizin Berlin (Host Prof Britta Eickholt) as Einstein Visiting Fellow (2020-2024).

## Author contributions
Conceptualisation: A.V., A.H.S., Z.M.; Experimental design: A.V., E.M., R.T.N., A.H.S., Z.M.; Transgenic line development and breeding: A.V., L.B.K., A.H.S.; Electrophysiology experiment design: A.V., R.T.N., recordings: R.T.N., data analysis: R.T.N., F.M.; Injections, tissue collection and preparation: A.V., E.M.; Immunohistochemistry for cFos: A.V., E.M.; Iba1, CD68, GFAP, S100β, vGlut1 and PSD95: A.V.; P.V. and V.V.A.: M.M., F.S.; imaging: A.V., M.M.; data analysis: A.V., M.M., A.J.; Expert advice on statistical analysis: O.B. Provided funding: A.V., R.T.N., E.O.M., L.B.K., V.V., Z.M. Wrote the first draft of the paper: A.V. All authors discussed the results and commented or edited the paper.

## Competing interests
The authors declare no competing interests
