## [Peer Review File · Communications Biology]

Reviewers' comments:

Reviewer #1 (Remarks to the Author):

In this study, the authors investigated activity-dependent neuron-glia interactions by using DREADD-based chemogenetic methods. The results demonstrate that acute changes in neuronal activity elicit rapid effects on the interactions between glial cells, glutamatergic and GABAergic neuronal networks. The authors thoroughly analyzed glial cell morphologies, aspects of PV interneurons and synaptic status. In particular, they extensively examined the effects of short-term neuronal activity, which is a novel feature. However, there are some problems in terms of conceptual flow and interpretation of results.

Major issues:

1. In this study Rbp4-Cre lines were used to upregulate and downregulate neuronal activity in layer 5 cortical neurons and a subset of hippocampal neurons. The interpretation is based on the assumption that layer-specific manipulation was achieved. However, Figures 1 and 2 show that cFos expression is found in not only layer 5 but also other layers. They should demonstrate the laminar distribution of cFos-positive cells, and reconsider the descriptions and interpretations throughout the result and discussion. If strong effects are present on layer 5 neurons, they should show the evidence.
2. They showed the decrease in the number of PV interneurons by the chemogenetic manipulation, but the interpretation is not described. It is unlikely that PV neurons were subjected to cell death.
3. Each results section mostly begins with introductory statements. These statements should be placed in the introduction section, and should be included in the aim of this study. Redundant statements in the discussion should be reduced.

Minor issues:

1. The first part of the results were carried out in slice preparations. However, there is no description in the text that the result was obtained in vitro.
2. Many panels of figures (such as 1a, 1g, 1h, 1j, 1k, 2h, 3a, 3e, 3g, 3k, 4d, 4e, 5a, 5b, 6a, 6b, 6d, many figure 7 and 8 panels) are not cited in the text.

Reviewer #3 (Remarks to the Author):

This is a very interesting manuscript with a huge amount of data incorporated, but there are some ambiguities and contradictions left that need clarification. I also think a main control group with CNO is missing for all experiments.

1/ What is exactly the rationale behind the choice for targeting specifically Rbp4 neurons? It does not seem a commonly used neuronal marker?

2/ Electrophysiological data in Figure 1: it is not clear what the number of patched cells was, n=? Any clarification for the large variation in panel d?

3/ Figure 1: To validate dose-dependent CNO effects on both excitatory and inhibitory DREADDs, the authors injected mice with different doses of CNO (excitatory: 0.05 mg/kg, 0.1 mg/kg and 10 mg/kg; inhibitory: 1 mg/kg and 5 mg/kg) or with saline as a control (panel g), but the panels h,i,j,k show only one dose; which dose? It is not mentioned?

4/ Figure 3, CD68 in Iba1+ cells in saline conditions: why is there such a huge difference between the baseline counts in saline conditions between Rbp4Cre-hM3Dq and Rbp4Cre-hM4Di mice (panel f versus panel l)?

5/ Figure 4 panel b, how do you explain the lack of dose dependent effect? Especially no effect of 10 mg/kg CNO in this specific experiment which is unexpected in view of the effects at lower CNO concentrations...

6/ Please pay attention to the new terminology in the microglia field (doi:

10.1016/j.neuron.2022.10.020); terminology “activated” is not used anymore (e.g. lines 287-288, Collectively, these changes in microglia suggest that, in Rbp4Cre-hM3Dq mice injected with CNO, microglia shift from a ramified to an activated state and decrease in cell density in subcortical regions...); Refer to microglia in your paper as “reactive to” or “responding to” while describing the particular signals they respond to (i.e., the context) instead of using the widely used broad term “activated,” as microglia are always active in both health and disease.

Idem on other occasions throughout the text, e.g. line 529 “enables cytological changes suggesting microglial activation”

Idem for astrocytes and changes described in astrocytes.

7/ Were the experimenters blinded for the genotypes and treatments upon analysis of the data?

8/ Lines 906-908 “Data are reported as means \pm standard error of the mean (SEM).

Microglial cells and astrocytes analyses were performed on data from at least 2 brains per group and for each analysis a minimum of 3 sections per brain”. I think it is fair to say that the experiments whose data are shown from just 2 mice really cannot be representative and would be better removed. Two mice are really not enough independent samples. Or else, I think it should be clear to the reader exactly which results were obtained in 2 mice and which in larger groups of mice. Now it is often indicated for different groups as n=2-6 Statistics parametric student t-test is in my opinion not correct in view of the small sample

sizes for some groups; better use non parametric tests.

9/ I am puzzled by the following statement (lines 385-386) “However, ‘silencing’ Rbp4Cre neurons leads to the inhibition of astrocyte activation”; in comparison with healthy resting conditions? In healthy resting conditions astrocytes are not activated???. How can you thus explain this?

What is the effect of the high doses of CNO alone on the morphology/markers of the astrocytes and the microglia? This is an important control that is missing throughout the paper, and actually the study's biggest limitation. It would be best to add these additional control experiments. Or else at least describe in detail this is a weakness of the current study.

10/ Lines 519-521 “To further support this, we demonstrate that acute activation of Rbp4Cre cortical layer 5 and dentate gyrus neurons affects Rbp4Cre-hM3Dq animal activity and locomotor function”, and “density as well as reduced motor activity and the onset of a sedation-like state...”

Which figure? I cannot find these behavioural data?

11/ Lines 549-553; “sing DREADD viral injections, microglia and astrocytes first migrate to the injection site to detect and react to the virus as a potential pathogen. We anticipated that it would be difficult to study changes in glial cells caused by DREADD neuromodulation itself without being confounded by the effects of the injection itself.”

Two points to handle here, first the typo “sing???”; second, the statement is not true, when injecting viral vectors, possible confounding effects on injection can be easily circumvented by using appropriate control groups with injections of control vector!!!! This is usually done and gives the appropriate control group. Here in this paper the appropriate control group, control litter mice with injections of CNO have not been performed, the saline control group cannot replace this.

12/Discrepant discussion on the doses of CNO. If the discussion is about the effects on behaviour then it is said that the doses of CNO of 10 mg/kg are too high and may even be toxic and lead to seizures and death, but many of the described effects on astrocytes and microglia also become apparent only at the higher doses of CNO, so may as well be a toxic effect, no? I think the discussion should be revisited globally and be more nuanced, and also that it should only focus on clear effects that have been proven in the paper in sufficient animals

13/ Reference to the Li paper; are the effects of Gi DREADDs observed only in pathological conditions (pain) or also in healthy mice?

14/ Throughout the paper the main results are obtained by neuronal DREADD modulation on other cell types, so it seems logical to me that the authors describe throughout the text about “neuron-glia interactions” and not “glial-neuron interactions”

Reviewer #4 (Remarks to the Author):

In this study, Vadisiute et al. aim to determine how acute changes in neuronal activity in layer 5 of cortex affect the number of excitatory synapses, the activation of microglia and astrocytes, and the number of parvalbumin-positive neurons and perineuronal nets. To this end, they crossed the Rbp4-Cre mouse line, which specifically expresses Cre in layer 5 cortex and in dentate gyrus, with mouse lines expressing the chemogenetic DREADD receptors hM3Dq or hM4Di. They then injected the DREADD ligand CNO into adult mice, and 90 minutes later they assessed the above parameters by immunohistochemical analysis. From their results, the authors conclude that acute chemogenetic activation leads to activation of microglia and astrocytes, increased parvalbumin expression, and reduced synaptic markers, whereas acute chemogenetic inhibition has the opposite effects.

Overall, this study addresses an intriguing and timely question, i.e. how rapidly can changes in neuronal activity alter the functionality of the surrounding neural tissue. The manuscript is largely well-written, and the figures are mostly beautifully prepared. The topic will be of interest to a wide range of neuroscientists. However, there are some critical concerns regarding the execution and statistical analysis of the experiments that render the study difficult to interpret in its current state.

1. A key concern is the extremely low numbers of animals used in this manuscript. The entire study rests on a single set of mice, and this set contains only $n=2$ animals in most groups. This n is far too low to accurately assess the parameters analyzed here, which inevitably show substantial biological variance between subjects. It is also challenging, if not impossible, to conduct correct statistical analysis on groups of $n=2$ animals (see paragraph below). In order to enable appropriate analysis of their data, the authors need to increase the animal numbers to at least 4-5 animals per group.

2. A second key concern is the statistical analysis conducted, which is incomplete or incorrect for many of the figures. In particular, the following points need to be addressed:

a) The authors state that they analysed e.g. two animals per group and three sections per animal for their immunohistochemical analysis, i.e. a total of six sections per group. Did

they then average the values from the sections for each animal and conduct statistics on $n=2$ animals (i.e. two values per group in the statistical analysis), or did they conduct statistics on $n=6$ sections (i.e. six values per group in the analysis)? In the latter case, this analysis would be incorrect, since it violates the assumption of independence between the samples. The three sections per animal should be considered as technical replicates, and the values from these samples should be averaged to generate one final value per animal.

b) In addition, the individual values should be plotted on the graphs in addition to the averages and error bars, to allow a faster assessment of the number and variance of data points.

c) The authors state that for comparisons of more than three groups, they used a two-way ANOVA (line 925). However, it is incorrect to use a two-way ANOVA simply because there are more than three groups being compared. A two-way ANOVA analyses the influence of two independent factors that affect a given parameter. The authors should either clearly identify the two independent factors analysed in each panel (e.g. in figures 7 and 8, the two factors are (1) drug treatment and (2) brain region / cortical layer), or they should use a one-way ANOVA where there are no two independent factors (figures 2-6).

d) In addition, for every ANOVA conducted, the authors need to list the p-value, F-value, and degrees of freedom for the main effects and, in the case of two-way ANOVAs, for the interaction, in addition to the multiple comparisons analysis.

Additional comments:

3. Please provide additional information on the mice used in this study. Which genetic background were the lines on? Were the experimental mice males, females, or mixed groups, and if mixed groups, exactly how many males and females were in each group?

4. Please explicitly state in the methods section whether the experimenters were blind to genotype at all points of the experiment.

5. Please add further information on the CNO. Where was it obtained, and how was it prepared and administered? Which dose of CNO was used in figures 7-8?

6. The fonts on some of the figures are extremely small and hard to read in the printed version. Moreover, in some figures, the resolution is not sufficient to be able to interpret even the digital version (e.g. figure 4f). I would recommend increasing the font size to a

minimum of 6 pt in the printed version.

7. For the immunohistochemistry methods, please add precise blocking / permeabilization / incubation time conditions for each antibody, rather than generic statements such as '2-10% normal goat serum or donkey serum' or 'Primary antibody combinations underwent overnight, 20 h or 48 h incubation'. I would recommend adding a table that provides the exact conditions for each antibody.

8. In supplementary tables 3 and 4, what is meant by a saline concentration of 0.25 mg/ml? Does this mean that the authors used 0.25 mg/ml NaCl (instead of 0.9% NaCl, i.e. 9 mg/ml NaCl, as is usual for physiological saline)? Please specify in the table or figure legend.

9. In supplementary table 3, the authors state that red asterisks indicate mice that died within 90 min of CNO administration, before they could be perfused. For the administration of 10 mg/kg CNO, this is very confusing. It looks like the authors administered CNO to 3 mice, 4 of which subsequently died. Clearly this is not what they meant, but it would be helpful to express this more accurately.

10. What is the difference between figure 3b and 3c? For the purpose of clarity, the authors should describe these two graphs separately in the figure legend, instead of combining the description as (b, c, d).

11. The authors often don't discuss the panels in the order in which they are arranged in the figures, and many panels do not seem to be discussed anywhere in the text. This makes it very hard to match what is written in the results section with the figures and data. This is particularly pronounced for (but not limited to) Figures 7 and 8, where there seems to be no description at all of the results shown in the majority of the panels. The heat maps in 7e, 7h, 7k, 8e, 8h, and 8k are poorly described and the low resolution makes them largely illegible and hard to understand.

12. The discussion is very hard to follow, since it is one long text with no obvious structure into paragraphs or concepts. It is challenging to identify the main points that the authors are making here. The authors should re-arrange this section into shorter paragraphs with clearly identifiable conclusions / statements / concepts.

Department of Physiology, Anatomy and Genetics
University of Oxford
South Parks Road, Oxford, OX1 3QX, UK

Reception: +44-1865-272168
Fax: +44-1865-272420

Zoltán Molnár MD DPhil
Professor of Developmental Neuroscience
Tutor of Human Anatomy – St John's College

Direct Line: +44-1865-282664
Laboratory: +44-1865-282663
E-mail: Zoltan.Molnar@dpag.ox.ac.uk

Oxford, 1/4/24

Dario Ummarino, PhD
Senior Editor
Communications Biology

Re: Response to referees

Reviewers' comments:

Reviewer #1 (Remarks to the Author):

In this study, the authors investigated activity-dependent neuron-glia interactions by using DREADD-based chemogenetic methods. The results demonstrate that acute changes in neuronal activity elicit rapid effects on the interactions between glial cells, glutamatergic and GABAergic neuronal networks. The authors thoroughly analyzed glial cell morphologies, aspects of PV interneurons and synaptic status. In particular, they extensively examined the effects of short-term neuronal activity, which is a novel feature. However, there are some problems in terms of conceptual flow and interpretation of results.

Major issues:

1. In this study Rbp4-Cre lines were used to upregulate and downregulate neuronal activity in layer 5 cortical neurons and a subset of hippocampal neurons. The interpretation is based on the assumption that layer-specific manipulation was achieved. However, Figures 1 and 2 show that cFos expression is found in not only layer 5 but also other layers. They should demonstrate the laminar distribution of cFos-positive cells, and reconsider the descriptions and interpretations throughout the result and discussion. If strong effects are present on layer 5 neurons, they should show the evidence.

Answer: We state that the chemogenetic manipulation is layer-specific, but the activation is indeed affecting many other cortical layers, since layer 5 projection neurons have indeed extensive cortical connectivity as the referee pointed out. We included low power images across the entire cortical thickness, and we demonstrate cFos+ cells laminar distribution across the cortical depth in the mice where the excitatory DREADD was activated in primary visual cortex (see Supplementary Figure 1e), and in mice where inhibitory DREADD was activated in primary visual cortex (see Supplementary Figure 1f). Indeed, these figures demonstrate that layer 5 subtype activation elicited response across the entire depth of the

cortex and changes are also expected across the target regions of the layer 5 projections across the brain, including striatum, thalamus, and brainstem. This is now discussed to increase clarity.

2. They showed the decrease in the number of PV interneurons by the chemogenetic manipulation, but the interpretation is not described. It is unlikely that PV neurons were subjected to cell death.

Answer: We agree that it is unlikely that PV+ neurons were subjected to cell death. Changes in PV+ protein expression is most likely behind the changes. PV+ interneurons might be able to compensate for the rapid changes in excitatory neuron activity in L5 projection neurons by regulating their level of parvalbumin protein expression and/or proteins of the extracellular matrix when there is a transient perturbation in network activity. The onset of expression and the maintenance of the calcium-binding protein parvalbumin have been shown to be regulated by cell-extrinsic factors including activity, thalamocortical and dopaminergic afferents, and neurotrophins as well (Patz et al., 2004).

In Rbp4^{Cre}-hM3Dq mice, we analysed PV+ density in saline and 10 mg/kg CNO brains (Fig. 8a,b). No significant changes in PV+ density were observed after CNO application in RSP, PTLp and S1 (CNO effect: $F_{(1,4)} = 2.613$, $p = 0.1813$ Fig. 8c). We measured laminar PV+ distribution where heat maps illustrate no significant effects of CNO administration on PV+ interneuron densities and laminar distributions within and between brain regions (CNO effect: RSP $F_{(1,4)} = 1.907$, $p = 0.2394$; PTLp $F_{(1,4)} = 0.8506$, $p = 0.4086$; S1 $F_{(1,4)} = 5.841$, $p = 0.073$; Fig. 8a-l). However, after CNO application, PV+/VVA- cell densities increased in RSP L6a, while PV+/VVA+ cells were not affected and VVA+ cell densities decreased in RSP layers 2, 3 and 5 (Supplementary Fig. 4a,b,c). In PTLp, no significant changes in PV+/VVA-, PV+/VVA+, and VVA+ cell densities were observed after CNO application (Supplementary Fig. 4d,e,f). In S1, there were no changes in PV+/VVA- and VVA+ cell densities within different cortical layers, but PV+/VVA+ cell densities increased in L4-5 after CNO application (Supplementary Fig. 4g,h,i). We measured PV+ cell density in different hippocampus regions and subcortical areas and found no significant changes (Supplementary Fig. 5a,b). In Rbp4^{Cre}-hM4Di, we analysed PV+ density in saline and 1 mg/kg CNO brains (Fig. 9a,b). No significant changes in PV+ cell densities were observed after CNO application in RSP, PTLp, and S1 (CNO effect: $F_{(1,2)} = 0.4246$, $p = 0.5815$, Fig. 9c). Moreover, no differences were observed in the laminar distribution of PV+ cells in RSP, PTLp, and S1 (CNO effect: RSP $F_{(1,2)} = 0.04222$, $p = 0.8562$; PTLp $F_{(1,2)} = 2.154$, $p = 0.2799$; S1 $F_{(1,2)} = 0.001787$, $p = 0.9701$; Fig. 9a-l). Furthermore, no significant changes in PV+VVA-, PV+/VVA+, and VVA+ cell densities were found in RSP, PTLp and S1 (Supplementary Fig. 6a-i) as well as subcortical regions (Supplementary Fig. 7a,b). These results suggest that acutely activating and silencing Rbp4-Cre neurons does not affect PV+ interneuron densities, with only minor effects on PNNs.

3. Each results section mostly begins with introductory statements. These statements should be placed in the introduction section, and should be included in the aim of this study. Redundant statements in the discussion should be reduced.

Answer: We changed some of the introduction and introduced glial cells in more detail. This placed some of the introductory statements into the introduction. We find the short primers at the beginning of results useful to place the experiments into context.

Minor issues:

1. The first part of the results were carried out in slice preparations. However, there is no description in the text that the result was obtained in vitro.

Answer: We included a description that electrophysiology experiments were carried out in acute slices in vitro. The following text was introduced on the details:

“Adult inhibitory and excitatory designer receptors exclusively activated by designer drugs (DREADDs) (Fig. 1a) mice were used for validation and all experiments presented in this study. To validate the effects that were elicited by inhibitory and excitatory DREADDs (Rbp4^{Cre}-hM3Dq and Rbp4^{Cre}-hM4Di) following CNO application, we recorded the in-vitro responses and activity of layer 5 neurons in the primary motor cortex (M1) to an increasing current administration (Fig. 1b).”

2. Many panels of figures (such as 1a, 1g, 1h, 1j, 1k, 2h, 3a, 3e, 3g, 3k, 4d, 4e, 5a, 5b, 6a, 6b, 6d, many figure 7 and 8 panels) are not cited in the text.

Answer: We made sure that every single panel is cited in the text, and these appear across the entire text of the results section to help the reader.

Reviewer #3 (Remarks to the Author):

This is a very interesting manuscript with a huge amount of data incorporated, but there are some ambiguities and contradictions left that need clarification. I also think a main control group with CNO is missing for all experiments.

1/ What is exactly the rationale behind the choice for targeting specifically Rbp4 neurons? It does not seem a commonly used neuronal marker?

Answer: We have a long-standing interest in the manipulation of various cortical projection neuron subtypes by chronically silencing these populations. We have been exploring the consequences of silencing Rbp4-Cre⁺ layer 5 neurons of the cortex by ablating Snap25 protein expression and we compared this paradigm with silencing layer 6b (Drd1a-Cre) and layer 6a (Ntsr1-Cre) cell groups. The Rbp4-Cre⁺ population was the most interesting for us because of our various previously analysed and published anatomical and physiological phenotypes; see the following papers):

- Cell-specific loss of SNAP25 from cortical projection neurons allows normal development but causes subsequent neurodegeneration A Hoerder-Suabedissen, KV Korrell, S Hayashi, A Jeans, DMO Ramirez, ...*Cerebral Cortex* 29 (5), 2148-2159
- A role for the cortex in sleep-wake regulation LB Krone, T Yamagata, C Blanco-Duque, MCC Guillaumin, MC Kahn, ...*Nature neuroscience* 24 (9), 1210-1215
- Maturation of complex synaptic connections of layer 5 cortical axons in the posterior thalamic nucleus requires SNAP25 S Hayashi, A Hoerder-Suabedissen, E Kiyokage, C Maclachlan, K Toida, ...*Cerebral Cortex* 31 (5), 2625-2638
- The role of snare proteins in cortical development A Vadasiute, E Meijer, F Szabó, A Hoerder-Suabedissen, E Kawashita, ...*Developmental Neurobiology* 82 (6), 457-475
- Differential effect on myelination through abolition of activity-dependent synaptic vesicle release or reduction of overall electrical activity of selected cortical

projections ...KV Korrell, J Disser, K Parley, A Vadisiute, MC Requena-Komuro, ...*Journal of anatomy* 235 (3), 452-467

We aimed to perform the chemogenetic manipulation experiments (DREADD+ and -) in exactly the same cortical projection neuronal population as the Snap25 cKO. This is why our choice had fallen again on the Rbp4-Cre+ layer 5 population. We also explored the hippocampus because of the Cre recombinase expression in the dentate gyrus with projections to CA3 region. This provides an additional system where the activated neurons and their projections can be studied. We now explain this in the introduction.

Regarding the lack of controls, we added a completely new control group. We agree with the referee that one can control for various conditions and therefore we included a control group that does not have a DREADD receptor but has Cre expression and still received CNO and saline. As you will see from the experiments below, the referee was right, the CNO itself produced glial cell response. We previously published that CNO itself can have influence on sleep (without DREADD receptor, <https://doi.org/10.7554/eLife.84740>) and now we extend these observations to microglia behavior. We are grateful for these comments, this additional observation will further caution researchers who use these models.

2/ Electrophysiological data in Figure 1: it is not clear what the number of patched cells was, n=? Any clarification for the large variation in panel d?

Answer: We included “n” size and every datapoint appears separate at every single panel in Figure 1. In Figure 1d, the large variation is due to two high resistance neurons that are consistently over the median (in baseline and both CNO conditions). There does not seem to be anything in particular with these neurons other than higher resistance and therefore were not removed as outliers. We checked the CVs, and they were around 0.5 nA which is a realistic value for this type of data. Therefore, we did not remove any datapoint from the analysis.

3/ Figure 1: To validate dose-dependent CNO effects on both excitatory and inhibitory DREADDs, the authors injected mice with different doses of CNO (excitatory: 0.05 mg/kg, 0.1 mg/kg and 10 mg/kg; inhibitory: 1 mg/kg and 5 mg/kg) or with saline as a control (panel g), but the panels h,I,j,k show only one dose; which dose? It is not mentioned?

Answer: We included CNO concentrations to Figure 1 panels h,l,j,k.

4/ Figure 3, CD68 in Iba1+ cells in saline conditions: why is there such a huge difference between the baseline counts in saline conditions between Rbp4Cre-hM3Dq and Rbp4Cre-hM4Di mice (panel f versus panel I)?

Answer: One possibility for this difference is that the excitatory (hM3Dq) and inhibitory (hM4Di) DREADDs have different genetic backgrounds. Previously both excitatory and inhibitory DREADDs were designed as targeted insertions into the *Gt(ROSA)26Sor* locus. However, in 2017, The Jackson Laboratory reported and confirmed a randomly integrated construct CAG-LSL-hM3Dq-pta-mCitrine instead of *Gt(ROSA)26Sor* locus for excitatory DREADD. We discuss this in the methods and the discussions sections.

5/ Figure 4 panel b, how do you explain the lack of dose dependent effect? Especially no effect of 10 mg/kg CNO in this specific experiment which is unexpected in view of the effects at lower CNO concentrations...

Answer: (We changed a few figures and now it is Fig 6b.) We speculate that microglial cells respond to the highest concentration of CNO by retracting their primary branches and becoming bigger. Figure 6 panel b has three graphs: first graph shows decrease in microglia volume in mice injected with 0.05 mg/kg and 0.1 mg/kg CNO, but the microglia volume was like the saline group with the highest 10 mg/kg CNO dose. Second graph in b shows a reduced number of primary branches only in 10 mg/kg CNO. That is why we propose that the cells become bigger by retracting their primary branches when the highest concentration of CNO is applied. We now discuss this in the text.

6/ Please pay attention to the new terminology in the microglia field (doi: 10.1016/j.neuron.2022.10.020); terminology “activated” is not used anymore (e.g. lines 287-288, Collectively, these changes in microglia suggest that, in Rbp4Cre-hM3Dq mice injected with CNO, microglia shift from a ramified to an activated state and decrease in cell density in subcortical regions...); Refer to microglia in your paper as “reactive to” or “responding to” while describing the particular signals they respond to (i.e., the context) instead of using the widely used broad term “activated,” as microglia are always active in both health and disease.

Idem on other occasions throughout the text, e.g. line 529 “enables cytological changes suggesting microglial activation”. Idem for astrocytes and changes described in astrocytes.

Answer: We would like to thank the referee for noticing this mistake. We changed ‘activated’ to ‘reactive to’ and/or ‘responding to’ throughout the manuscript.

7/ Were the experimenters blinded for the genotypes and treatments upon analysis of the data?

Answer: Yes, we stated this in the methods section that the immunohistochemistry, imaging and all data analysis were performed blind to the genotype and treatment condition.

8/ Lines 906-908 “Data are reported as means \pm standard error of the mean (SEM). Microglial cells and astrocytes analyses were performed on data from at least 2 brains per group and for each analysis a minimum of 3 sections per brain”. I think it is fair to say that the experiments whose data are shown from just 2 mice really cannot be representative and would be better removed. Two mice are really not enough independent samples. Or else, I think it should be clear to the reader exactly which results were obtained in 2 mice and which in larger groups of mice. Now it is often indicated for different groups as n=2-6 Statistics parametric student t-test is in my opinion not correct in view of the small sample sizes for some groups; better use non parametric tests.

Answer: We added n size to every figure and re-did the statistical analysis. The issue is as follows: two-way ANOVA assumes that all the variances can be explained by the sum of fixed effects and an independent error term which comes from the same distribution for all observations. In this case, the model is: [cFos density] = [effect of region] + [effect of treatment] + [interaction effect of region-by-treatment] + [random error which explains variation between mice]. As the spread of values is very different between regions (see from the y-axis), this will most likely break the assumption that the random error in each observation looks similar and make p-values artificially smaller. Many biological effects of this form are exponential in nature, so we used sqrt-transform the data to address this issue. When we used a mixed-effects ANOVA (e.g., split-plot ANOVA) as slices from different regions were taken from the same mice and the selection of regions was based on neuronal projections, and for multiple comparisons we used Benjamini and Yekutieli method.

Statistical analyses were conducted using GraphPad Prism. To compare differences of neuronal activity after CNO application, we used mixed-effects ANOVA with Šidak's multiple comparisons test or one-way ANOVA via Dunnett's multiple comparisons test and these tests were used for analysis of electrophysiology data. For any single comparison between two groups, we used an unpaired Student's *t*-test for normally distributed data, and Mann-Whitney for not normally distributed data. Normality was verified using the Shapiro-Wilk test (assessed by QQ plot). Comparison between different CNO concentrations was performed using mixed-effects ANOVA via Benjamini and Yekutieli multiple comparisons test for Rbp4^{Cre}-hM3Dq and Rbp4^{Cre}-hM4Di, and mixed-effects ANOVA via Holm-Šidak's multiple comparisons test for Rbp4^{Cre}, these tests were used for analysis of microglial density and astrocyte density in the primary visual cortex with different concentrations of CNO. Comparison between different brain regions and CNO concentrations was performed using linear mixed-effects modelling, with brain region as a within-subjects factor and CNO concentration as a between-subjects factor. Models were fitted using the restricted maximum likelihood method and included a Geisser-Greenhouse correction for sphericity. *Post hoc* testing was used where appropriate to compare CNO groups against the control (saline) group for each brain region, with a Benjamini-Yekutieli correction to control the false discovery rate at 5%. This strategy was used for analysis of the density, morphology and co-localisation of immediate early genes, glial cells densities, microglial morphology, and synaptic and interneuron densities. *p*-values are reported after Benjamini-Yekutieli adjustment; for uncorrected *p*-values see *Supplementary material*. Mixed-effects ANOVA via Holm-Šidak's multiple comparisons test was used for Rbp4^{Cre} control experiments for microglial density and astrocyte density.

9/ I am puzzled by the following statement (lines 385-386) "However, 'silencing' Rbp4Cre neurons leads to the inhibition of astrocyte activation"; in comparison with healthy resting conditions? In healthy resting conditions astrocytes are not activated??? How can you thus explain this?

What is the effect of the high doses of CNO alone on the morphology/markers of the astrocytes and the microglia? This is an important control that is missing throughout the paper, and actually the study's biggest limitation. It would be best to add these additional control experiments. Or else at least describe in detail this is a weakness of the current study.

Answer: The referee wanted a more detailed description of the astrocytes and microglia. Moreover, the referee also wanted to have additional controls regarding CNO effects. We agree with these requests and include a great deal of new experimental data.

We performed additional experiments using only Rbp4^{Cre} mouse line and injected 4 animals with saline and 5 with 10 mg/kg CNO. Supplementary figure 2 demonstrates that CNO does not elicit cFos activation, therefore neurons are not excited.

Furthermore, we performed Iba1+ cell density and Iba1 + CD68 analysis, GFAP+ density and S100β+ density, and analysis of co-expression of GFAP+ cells and S100β+ cells in primary visual cortex, superior colliculus, hippocampal CA1 and CA3 regions. (Previously Figure 3, now Figure 4): We show no changes in microglia density in primary visual cortex and no changes in laminar distribution between saline and 10 mg/kg CNO groups (Fig 4m,n,o). In superior colliculus microglia density was decreased in mouse brains injected with 10 mg/kg CNO compared to saline groups but increased in hippocampal CA3 (Fig. 4p). Moreover, the percentage of CD68 in Iba1+ microglia was increased in mouse brains treated with 10 mg/kg CNO compared to saline in primary visual cortex layer 5, superior colliculus and hippocampal CA1 region, but no differences were observed in hippocampal CA3 region

(Fig 4r). (Previously Figure 5, now Figure 6): Increase in GFAP+ density in primary visual cortex layer 1 and layer 6 was observed in Rbp4^{Cre} mice treated with 10 mg/kg CNO compared to saline (Fig 6h), S100 β + density was not affected by CNO treatment (Fig 6h) and the percentage of GFAP+ cells expressing S100 β was increased only in layer 1 (Fig 6h). Average data of the entire primary visual cortex showed an increase in GFAP+ density and GFAP+ cells expressing S100 β in mice treated with 10 mg/kg CNO compared to saline (Supplementary Figure 3c). Figure 7: GFAP+ density was increased in hippocampal CA1 and CA3 regions in mice treated with 10 mg/kg CNO (Fig 7j), S100 β + density and GFAP+ expressing S100 β + was increased in all tested subcortical regions after CNO treatment (Fig 7k,l).

We agree with the referee, that the added control and more detailed description improves the paper.

10/ Lines 519-521 “To further support this, we demonstrate that acute activation of Rbp4Cre cortical layer 5 and dentate gyrus neurons affects Rbp4Cre-hM3Dq animal activity and locomotor function”, and “density as well as reduced motor activity and the onset of a sedation-like state...” Which figure? I cannot find these behavioural data?

Answer: The referee is correct; we do not have detailed quantified behavioural observations on the excitatory DREADD animals. We only have qualitative observations. However, we have detailed analysis of the inhibitory DREADD effects, together with EEG, but these are not included into our study. The reason why we still mention animal behaviour is to warn future experimentalists from using the high dose of CNO that produce sedation-like state and may have adverse effects on the animals. This information, although not quantitative, could be useful for researchers using similar paradigms.

11/ Lines 549-553; “sing DREADD viral injections, microglia and astrocytes first migrate to the injection site to detect and react to the virus as a potential pathogen. We anticipated that it would be difficult to study changes in glial cells caused by DREADD neuromodulation itself without being confounded by the effects of the injection itself.” Two points to handle here, first the typo “sing????”; second, the statement is not true, when injecting viral vectors, possible confounding effects on injection can be easily circumvented by using appropriate control groups with injections of control vector!!!! This is usually done and gives the appropriate control group. Here in this paper the appropriate control group, control litter mice with injections of CNO have not been performed, the saline control group cannot replace this.

Answer: we performed control experiments with control litter mice with saline and CNO injections. We answered it in detail above (your questions 9). In future experiments when we employ injections we shall follow the referee’s advice for controls.

12/Discrepant discussion on the doses of CNO. If the discussion is about the effects on behaviour then it is said that the doses of CNO of 10 mg/kg are too high and may even be toxic and lead to seizures and death, but many of the described effects on astrocytes and microglia also become apparent only at the higher doses of CNO, so may as well be a toxic effect, no? I think the discussion should be revisited globally and be more nuanced, and also that it should only focus on clear effects that have been proven in the paper in sufficient animals.

Answer: We discuss the effects of doses of CNO of 10 mg/kg on behaviour. We state that this dose is too high and may even be toxic and lead to seizures and death. We see changes

in synaptic density (Figure 3a-d) with lower doses, we also see decreased microglia density in hippocampus, we see dose dependent increase in CD68 immunoreactivity (Figure 4 d and f). The change in the microglial processes is dose dependent (Figure 5 f and g). We mention all this in the discussion.

13/ Reference to the Li paper; are the effects of Gi DREADDs observed only in pathological conditions (pain) or also in healthy mice?

Answer: not only pathological conditions. It has been shown that Gi-DREADDs in CA1 astrocytes impaired memory recall and decreased activity in the anterior cingulate cortex during retrieval. Also it induces long-lasting synaptic potentiation in CA1.

- Kol, A., Adamsky, A., Groysman, M., Kreisel, T., London, M., and Goshen, I. (2020). Astrocytes contribute to remote memory formation by modulating hippocampal–cortical communication during learning. *Nat. Neurosci.* 23:10. doi: 10.1038/s41593-020-0679-6
- Van Den Herrewegen Y, Sanderson TM, Sahu S, De Bundel D, Bortolotto ZA, Smolders I. Side-by-side comparison of the effects of Gq- and Gi-DREADD-mediated astrocyte modulation on intracellular calcium dynamics and synaptic plasticity in the hippocampal CA1. *Mol Brain.* 2021;14:144. <https://doi.org/10.1186/s13041-021-00856-w> . - DOI - PubMed

14/ Throughout the paper the main results are obtained by neuronal DREADD modulation on other cell types, so it seems logical to me that the authors describe throughout the text about “neuron-glia interactions” and not “glial-neuron interactions”.

Answer: we agree with this comment and changed the text accordingly.

We thank the referee for the constructive criticisms that helped to improve clarity in our presentation.

Reviewer #4 (Remarks to the Author):

In this study, Vadisiute et al. aim to determine how acute changes in neuronal activity in layer 5 of cortex affect the number of excitatory synapses, the activation of microglia and astrocytes, and the number of parvalbumin-positive neurons and perineuronal nets. To this end, they crossed the Rbp4-Cre mouse line, which specifically expresses Cre in layer 5 cortex and in dentate gyrus, with mouse lines expressing the chemogenetic DREADD receptors hM3Dq or hM4Di. They then injected the DREADD ligand CNO into adult mice, and 90 minutes later they assessed the above parameters by immunohistochemical analysis. From their results, the authors conclude that acute chemogenetic activation leads to activation of microglia and astrocytes, increased parvalbumin expression, and reduced synaptic markers, whereas acute chemogenetic inhibition has the opposite effects. Overall, this study addresses an intriguing and timely question, i.e. how rapidly can changes in neuronal activity alter the functionality of the surrounding neural tissue. The manuscript is largely well-written, and the figures are mostly beautifully prepared. The topic will be of

interest to a wide range of neuroscientists. However, there are some critical concerns regarding the execution and statistical analysis of the experiments that render the study difficult to interpret in its current state.

1. A key concern is the extremely low numbers of animals used in this manuscript. The entire study rests on a single set of mice, and this set contains only $n=2$ animals in most groups. This n is far too low to accurately assess the parameters analyzed here, which inevitably show substantial biological variance between subjects. It is also challenging, if not impossible, to conduct correct statistical analysis on groups of $n=2$ animals (see paragraph below). In order to enable appropriate analysis of their data, the authors need to increase the animal numbers to at least 4-5 animals per group.

Answer: There are several reasons for the low n numbers for some, but not all of the experimental paradigms. These experiments were started during Covid, and we decided to keep these initial numbers minimal. After, we noticed that the high doses of CNO caused adverse effects on mice's wellbeing, and we decided not to include more animals due to ethical reasons. These mice have not been bred since and now we are not in a position to resurrect the colony. Some measures show a dose-response curve, and thus while each data point contains few animals, the trend is statistically robust. However, we are working on a new inhibitory DREADD colony.

2. A second key concern is the statistical analysis conducted, which is incomplete or incorrect for many of the figures. In particular, the following points need to be addressed:

a) The authors state that they analysed e.g. two animals per group and three sections per animal for their immunohistochemical analysis, i.e. a total of six sections per group. Did they then average the values from the sections for each animal and conduct statistics on $n=2$ animals (i.e., two values per group in the statistical analysis), or did they conduct statistics on $n=6$ sections (i.e., six values per group in the analysis)? In the latter case, this analysis would be incorrect, since it violates the assumption of independence between the samples. The three sections per animal should be considered as technical replicates, and the values from these samples should be averaged to generate one final value per animal.

Answer: We average sections per animal and conducted statistical analysis on $n = 2$. We agree with the referee that it is not ideal, but we are confident of our results that show a dose-response curve, and thus while each data point contains few animals, the trend is statistically robust.

b) In addition, the individual values should be plotted on the graphs in addition to the averages and error bars, to allow a faster assessment of the number and variance of data points.

Answer: We agree with the referee, we plotted individual values on all graphs in all figures. Please see new figures.

c) The authors state that for comparisons of more than three groups, they used a two-way ANOVA (line 925). However, it is incorrect to use a two-way ANOVA simply because there are more than three groups being compared. A two-way ANOVA analyses the influence of two independent factors that affect a given parameter. The authors should either clearly identify the two independent factors analysed in each panel (e.g. in figures 7 and 8, the two

factors are (1) drug treatment and (2) brain region / cortical layer), or they should use a one-way ANOVA where there are no two independent factors (figures 2-6).

Answer: we used two-way ANOVA for two factors: drug treatment and brain region/cortical layer. We re-did all statistical analysis.

d) In addition, for every ANOVA conducted, the authors need to list the p-value, F-value, and degrees of freedom for the main effects and, in the case of two-way ANOVAs, for the interaction, in addition to the multiple comparisons analysis.

Answer: we listed all values in Supplementary Statistics and Source data excel files, and figure legends refer to them accordingly. We added all F-value and p-value for the interaction for each graph to the main text, and multiple comparisons analysis p-value(s) was added to each figure or figure legends.

Additional comments:

3. Please provide additional information on the mice used in this study. Which genetic background were the lines on? Were the experimental mice males, females, or mixed groups, and if mixed groups, exactly how many males and females were in each group?

Answer: Detailed information can be found in Supplementary Table 4. Lines were on C57BL/6 background. Genotype, sex, drug doses were all added to Table 4 (We added this table below to support our answer to your question 9).

4. Please explicitly state in the methods section whether the experimenters were blind to genotype at all points of the experiment.

Answer: we added additional information to methods sections that immunohistochemistry, imaging and data analysis were performed blind to the genotype and treatment condition.

5. Please add further information on the CNO. Where was it obtained, and how was it prepared and administered? Which dose of CNO was used in figures 7-8?

Answer: Information about CNO preparation was added to methods. CNO concentrations were added to both figure 7 and 8.

“CNO preparation: Clozapine-N-oxide dihydrochloride (Torcis, 6329) was used to prepare CNO solution. CNO solution was prepared under sterile conditions and passed through a Millipore filter. CNO was kept at -20°C and defrosted on the day of use and diluted using strike saline.”

6. The fonts on some of the figures are extremely small and hard to read in the printed version. Moreover, in some figures, the resolution is not sufficient to be able to interpret even the digital version (e.g., figure 4f). I would recommend increasing the font size to a minimum of 6 pt in the printed version.

Answer: we increased font size in all figures, rearranged some figures and transferred some of the graphs to supplementary figures to keep the remaining panels larger.

7. For the immunohistochemistry methods, please add precise blocking / permeabilization / incubation time conditions for each antibody, rather than generic statements such as ‘2-10% normal goat serum or donkey serum’ or ‘Primary antibody combinations underwent overnight, 20 h or 48 h incubation’. I would recommend adding a table that provides the exact conditions for each antibody.

Answer: We thank for this suggestion. Detailed information can be found in Supplementary Table 1.

Table 1: Primary and secondary antibodies

Primary Antibodies			
Target	Host-Species	Concentration	Manufacturer
Anti-cFos	Rabbit	1:500	Synaptic Systems, 226-003
Anti-Iba1	Rabbit	1:500	FUJIFILM Wako 019-19741
Anti-CD68	Mouse	1:500	Abcam, ab955
Anti-vGlut1	Guinea Pig	1:500	Merck Millipore AB5905
Anti-PSD95	Mouse	1:500	Thermo Xe3-1B8
Anti-GFAP	Rabbit	1:500	Dako Z0334
Anti-S100 β	Mouse	1:500	Sigma-Aldrich, S2532
Anti-PV	Rabbit	1:500	Swant, PV27
Biotinylated VVA		2 ug/ml	Vector laboratories, B-1235-2
Secondary Antibodies			
Fluorophore	Species	Concentration	Manufacturer
Alexa Fluor®488	Goat Anti-Rabbit IgG (H+L)	1:500	ThermoFisher, A11034
Alexa Fluor®488	Goat Anti-Mouse IgG (H+L)	1:500	Lifeteck, A21041
Alexa Fluor®633	Goat Anti-Guinea Pig IgG (H+L)	1:500	Molecular Probes, A21105
Alexa Fluor®488	Donkey Anti-Rabbit IgG (H+L)	1:500	Invitrogen, A21206
Cy5 streptavidin-conjugated		1:200	Invitrogen, SA1011

8. In supplementary tables 3 and 4, what is meant by a saline concentration of 0.25 mg/ml?

Does this mean that the authors used 0.25 mg/ml NaCl (instead of 0.9% NaCl, i.e. 9 mg/ml NaCl, as is usual for physiological saline)? Please specify in the table or figure legend.

Answer: mistake on our part, we used 0.9% NaCl, figures were corrected.

9. In supplementary table 3, the authors state that red asterisks indicate mice that died within 90 min of CNO administration, before they could be perfused. For the administration of 10 mg/kg CNO, this is very confusing. It looks like the authors administered CNO to 3 mice, 4 of which subsequently died. Clearly this is not what they meant, but it would be helpful to express this more accurately.

Answer: We agree with the referee, we clarify it and put more detailed information to Supplementary Table 4. Two mice died within 90 min of CNO administration, and before perfusion could be initiated. The brains of these mice were collected but were not included in further data analysis. We added information about dead animals (Indicated in red) to the comments section.

Table 4: Animal information

Two mice died within 90 min of CNO administration, and before perfusion could be initiated. The brains of these mice were collected but were not included in further data analysis. All animals used for this part of experiments were adults and littermates.

Rbp4^{Cre}-hM3Dq					
Animal ID	Sex	Genotype	Drug	Dose	Comments and Behaviour
RCDB2.2A	Male	Rbp4 ^{Cre} -hM3Dq	CNO	10mg/kg	Animal died, was not used for analysis
RCDB2.2B	Male	Rbp4 ^{Cre} -hM3Dq	Saline	0.9% NaCl	No changes in behaviour
RCDB2.2C	Male	Rbp4 ^{Cre} -hM3Dq	CNO	10mg/kg	Sedation-like state
RCDB2.2D	Male	Rbp4 ^{Cre} -hM3Dq	Saline	0.9% NaCl	No changes in behaviour
RCDB2.2E	Female	Rbp4 ^{Cre} -hM3Dq	Saline	0.9% NaCl	No changes in behaviour
RCDB2.2F	Female	Rbp4 ^{Cre} -hM3Dq	CNO	10mg/kg	Sedation-like state
RCDB2.2G	Female	Rbp4 ^{Cre} -hM3Dq	CNO	10mg/kg	Sedation-like state
RCDB2.2H	Female	Rbp4 ^{Cre} -hM3Dq	Saline	0.9% NaCl	No changes in behaviour
RCDB3.2B	Male	Rbp4 ^{Cre} -hM3Dq	CNO	0.5mg/kg	Seemed less active but were easily arousable
RCDB3.2D	Male	Rbp4 ^{Cre} -hM3Dq	CNO	0.1mg/kg	
RCDB3.2E	Male	Rbp4 ^{Cre} -hM3Dq	Saline	0.9% NaCl	No changes in behaviour
RCDB3.1A	Male	Rbp4 ^{Cre} -hM3Dq	CNO	1mg/kg	Animal died, was not used for analysis
RCDB3.1C	Male	Rbp4 ^{Cre} -hM3Dq	CNO	0.05mg/kg	Mild behavioural effect of reduced activity and slower locomotor activity

RCDB3.1D	Male	Rbp4 ^{Cre} -hM3Dq	CNO	0.05mg/kg	with no signs of distress	
RCDB3.1E	Female	Rbp4 ^{Cre} -hM3Dq	CNO	0.1mg/kg	Seemed less active but were easily arousable	
RCDB3.1F	Female	Rbp4 ^{Cre} -hM3Dq	Saline	0.9% NaCl	No changes in behaviour	
Rbp4^{Cre}-hM4Di						
RCDA9.2A	Male	Rbp4 ^{Cre} -hM4Di	CNO	5mg/kg	No changes in behaviour or welfare were observed	
RCDA9.2B	Male	Rbp4 ^{Cre} -hM4Di	CNO	1mg/kg		
RCDA9.2C	Male	Rbp4 ^{Cre} -hM4Di	Saline	0.9% NaCl		
RCDA9.2D	Female	Rbp4 ^{Cre} -hM4Di	CNO	5mg/kg		
RCDA9.2E	Female	Rbp4 ^{Cre} -hM4Di	CNO	1mg/kg		
RCDA9.2F	Female	Rbp4 ^{Cre} -hM4Di	Saline	0.9% NaCl		
Control animals						
TNPW12.1e	Female	hM3Dq	Saline	0.9% NaCl	No changes in behaviour or welfare were observed	
TNPW12.1f	Female	hM3Dq	Saline	0.9% NaCl		
SABX11.1a	Male	hM4Di	Saline	0.9% NaCl		
SABX11.1b	Male	hM4Di	Saline	0.9% NaCl		
RSAB12.2c	Male	Rbp4 ^{Cre}	Saline	0.9% NaCl		
RSAB12.2f	Female	Rbp4 ^{Cre}	Saline	0.9% NaCl		
TGJP43.1a	Male	Rbp4 ^{Cre}	CNO	10mg/kg		
TGJP43.1b	Male	Rbp4 ^{Cre}	CNO	10mg/kg		
TGJP43.1c	Female	Rbp4 ^{Cre}	CNO	10mg/kg		
TGJP43.1d	Male	Rbp4 ^{Cre}	Saline	0.9% NaCl		
TGJP43.1e	Male	Rbp4 ^{Cre}	Saline	0.9% NaCl		
TGJP43.1f	Male	Rbp4 ^{Cre}	CNO	10mg/kg		
TGJP43.1g	Female	Rbp4 ^{Cre}	CNO	10mg/kg		
All animals were perfused 90 min after saline or CNO administration						

10. What is the difference between figure 3b and 3c? For the purpose of clarity, the authors should describe these two graphs separately in the figure legend, instead of combining the description as (b, c, d).

Answer: (We changed figures, Figure 3 → Figure 4). Figure 4b shows laminar distribution within V1; Figure 4c – shows numbers within the entire thickness of the cortex.

11. The authors often don't discuss the panels in the order in which they are arranged in the figures, and many panels do not seem to be discussed anywhere in the text. This makes it very hard to match what is written in the results section with the figures and data. This is particularly pronounced for (but not limited to) Figures 7 and 8, where there seems to be no description at all of the results shown in the majority of the panels. The heat maps in 7e, 7h, 7k, 8e, 8h, and 8k are poorly described and the low resolution makes them largely illegible and hard to understand.

Answer: In the result section we now discuss all panels in the order they appear in the figures. We discuss figures 7 and 8 in more detail (now figures 8 and 9).

To increase clarity of figures, some panels from Figures 8 and 9 were moved to supplementary material, so the main figures appear larger. For interneurons (old Figures 7 and 8 – new figures 8 and 9) we transferred the heatmaps to supplementary material and increased the size of the remaining panels. We removed the p and d values from the heatmaps, and the numbers are listed in the supplementary statistics.

The following text has been extensively edited:

Chemogenetic manipulations of Rbp4^{Cre} neurons activity in layer 5 affected immunohistochemical markers of GABAergic neurons.

To further evaluate how acute chemogenetic manipulations affect interneurons, we investigated GABAergic neurons. Parvalbumin (PV) interneurons are fast-spiking GABAergic cells that provide perisomatic innervation in the cortex³⁶ and play a crucial role in cortical inhibition. PV interneuron dysfunction can affect the excitability of pyramidal neurons' firing in oscillatory neuronal networks in the cortex³⁷ and hippocampus³⁸. We used PV immunolabelling to evaluate acute activity-dependent changes in interneurons, as well as plant-based lectins for detecting components of perineuronal nets (PNN). PNN are extracellular matrix elements surrounding PV interneurons in various regions of the brain, including the cortex and are responsible for synaptic homeostasis and stabilising synaptic plasticity³⁹. PNN can be detected based on the lectin-binding capacity of their glycan components by *Vicia villosa* agglutinin (VVA)⁴⁰. PV itself can regulate the PNN expression and both PV and VVA are activity-dependent markers. Here, we evaluated PV+ interneuron density, VVA labelling and PV+ cell co-localisation with VVA signal in different cortical and subcortical regions. We measured PV+ density in retrosplenial area (RSP), posterior parietal association area (PTLp), and primary somatosensory cortex (S1).

In Rbp4^{Cre}-hM3Dq, we analysed PV+ density in saline and 10 mg/kg CNO brains (Fig. 8a,b), but no significant changes in PV+ density were observed after CNO application in RSP, PTLp and S1 (CNO effect: $F_{(1,4)} = 2.613$, $p = 0.1813$ Fig. 8c). We further measured laminar PV+ distribution and heat maps illustrate the effects of CNO administration within and between brain regions on PV+ interneuron densities, but no significant differences were observed in laminar distribution of PV+ in the cortex (CNO effect: RSP $F_{(1,4)} = 1.907$, $p = 0.2394$; PTLp $F_{(1,4)} = 0.8506$, $p = 0.4086$; S1 $F_{(1,4)} = 5.841$, $p = 0.073$; Fig. 8a-l). However, after CNO application, PV+VVA- was increased in RSP layer 6a, PV+VVA+ was not affected and VVA+ density was decreased in RSP layer 2, 3 and 5 (Supplementary Fig. 4a,b,c). In PTLp, after CNO application, no significant changes in PV+VVA-, PV+VVA+

and VVA+ density were observed (Supplementary Fig. 4d,e,f). In S1, no changes in PV+VVA- and VVA+ density within different cortical layers were observed, but PV+VVA+ increased in layer 4 and 5 after CNO application (Supplementary Fig. 4g,h,i). Moreover, we measured PV+ cell density in different regions of the hippocampus and subcortical areas (Supplementary Fig. 5a,b) but no significant changes were observed. In Rbp4^{Cre}-hM4Di, we analysed PV+ density in saline and 1 mg/kg CNO brains (Fig. 9a,b), but no significant changes in PV+ density were observed after CNO application in RSP, PTLp and S1 (CNO effect: $F_{(1,2)} = 0.4246$, $p = 0.5815$, Fig. 9c). Moreover, no significant differences were observed in laminar distribution of PV+ in RSP, PTLp and S1 (CNO effect: RSP $F_{(1,2)} = 0.04222$, $p = 0.8562$; PTLp $F_{(1,2)} = 2.154$, $p = 0.2799$; S1 $F_{(1,2)} = 0.001787$, $p = 0.9701$; Fig. 9a-l). Furthermore, no significant changes in PV+VVA-, PV+VVA+ and VVA+ densities in RSP, PTLp and S1 (Supplementary Fig. 6a-i) and PV+ density in subcortical regions were observed (Supplementary Fig. 7a,b). These results suggest that acute activation and silencing has no effect on PV+ interneurons, and only minor effect on PNN.

12. The discussion is very hard to follow, since it is one long text with no obvious structure into paragraphs or concepts. It is challenging to identify the main points that the authors are making here. The authors should re-arrange this section into shorter paragraphs with clearly identifiable conclusions / statements / concepts.

We thank for this suggestion. We structured the discussion into separate paragraphs following coherent and transparent logic according to these subheadings (that were removed after editing according to editorial guidelines):

Summary of findings

Validation of the models

Synaptic density changes

Changes in microglia and astrocytes

Changes in PV

Communications Biology is committed to improving transparency in authorship. As part of our efforts in this direction, we are now requesting that all authors identified as ‘corresponding author’ create and link their Open Researcher and Contributor Identifier (ORCID) with their account on the Manuscript Tracking System prior to acceptance. ORCID helps the scientific community achieve unambiguous attribution of all scholarly contributions. You can create and link your ORCID from the home page of the Manuscript Tracking System by clicking on ‘Modify my Springer Nature account’ and following the instructions in the link below. Please also inform all co-authors that they can add their ORCID to their accounts and that they must do so prior to acceptance.

We added ORCID details to all co-authors of the paper.

With kind regards,

Zoltán Molnár MD DPhil
Professor of Developmental Neuroscience
University of Oxford
<https://www.dpag.ox.ac.uk/team/zoltan-molnar>

Auguste Vadisiute MSc DPhil
Junior Research Fellow in Physiology and Medicine
University of Oxford
<https://www.dpag.ox.ac.uk/team/auguste-vadisiute>

Reviewers' comments:

Reviewer #1 (Remarks to the Author):

The manuscript has been improved, but there are still some problems.

First, the authors showed the laminar distribution of cFos+ cells after CNO application, but did not provide an adequate interpretation in the text. They still describe as if the effect is due to layer 5 projection neurons and specific cell type (see L143, "This increase was layer-specific as a function of dose as..." and the first paragraph in Discussion). Certainly neuronal activation could be originated from layer 5 neurons and a subset of hippocampal neurons, but the activation should immediately spread to cortical and subcortical regions. In fact, the new data clearly showed that cFos+ cells are densely distributed in other layers. The density appears to be rather higher in other layers than layer 5. It is therefore difficult to say the origins of various effects of chemogenetic activation, including behaviors and cellular morphology. In particular, it is likely that the effect on glial cell distribution and morphology is attributed to dramatic activation of all layer neurons. A logical interpretation should be described.

An interpretation of the results regarding PV neurons has been added at the end of the discussion. However, the description is too general to make sense of the result. Moreover, the results of statistical analysis are exaggerated despite the small sample size.

Furthermore, as the reviewer #2 pointed out, the high dose of CNO (10 mg/kg) has some effect without chemogenetic activation. This should also be considered with regard to effect on PV neurons. The authors emphasize that chemogenetic silencing did affect neither glial cells nor PV neurons even at high doses of CNO. However, it is possible that the CNO effect may be canceled by suppression of neuronal activity. It would be best to remove the result at high concentrations of CNO or to list them separately. In relation to this issue, the following statement in Discussion seems far from logical conclusion. "Our study also raises concerns regarding microglia and astrocyte responses that can be elicited even without chemogenetic activation, while solely CNO application causes clear changes in glial cells. Collectively, this shows that acute changes in the neuronal activity of a subset of cortical L5 and hippocampal neurons can have an immense global effect on glial cells."

Reviewer #2 (Remarks to the Author):

First of all, I would like to thank the authors for conducting the requested additional control

experiments including these with CNO alone in Rbp4Cre mice. They also answered most questions adequately. A few concerns however arise/remain.

As shown in the new results, the higher dose of CNO (10 mg/kg) by itself has several effects that may affect the readouts from the experiments in the DREADDed mice. Despite the fact that the authors clearly describe these unexpected, toxic (?) effects, however, they take little or no account of these results when describing the effects of chemogenetic modulation on astrocytes and microglia with this highest dose.

It is clear that 10 mg/kg CNO has effects on several read-outs, wouldn't it be better NOT to use this highest dose precisely in those experiments with DREADD modulation looking at the same read-outs? In such cases, how can one separate the CNO effects from the DREADD-mediated effects? For example, suppl Fig 3, the effect of 10 mg/kg CNO is very pronounced in Rbp4Cre mice (panel c) and therefore makes it very difficult, if not impossible, to correctly interpret panel a of the same figure. Panel a second graph S100beta, with low doses rather decreased density, only at the high dose an effect, impossible to tell whether this effect arises from CNO alone or from Gq-DREADD activation by CNO in my opinion. In panel b the 10 mg/kg dose was not used, only lower doses.

It is very important to describe the effects of CNO 10 mg/kg in the article and indicate that they are way too high. However, I also think it is important to describe the real, true DREADD mediated effects, so maybe with the low CNO doses but not with 10 mg/kg. Would the authors be willing to critically revise their data and description, and add more nuance here? There are certainly DREADD mediated effects at lower CNO doses, and some read-outs, such as cFos in neurons, are not affected by 10 mg/kg CNO. But in other places an unambiguous interpretation or description is not possible in my opinion, especially on the microglial and astrocytic markers. For example page 3 lines 18-23. Another example is Fig 4 with description on page 7 lines 13-18.

Also in the discussion the observation that microglia and astrocytes respond to (high?) CNO application is mentioned, but these effects are not further dissected from the real DREADD mediated responses, and therefore make the discussion unclear and sometimes a bit incoherent. E.g. page 11 lines 8-14. On what basis do the authors link the effects of CNO application alone also to neuronal activity changes? E.g. page 12 lines 16-29, the effect of CNO alone on microglia cannot be denied, maybe in the Gi-DREADD experiments the effects are opposite and therefore compensating or antagonising the CNO effect? E.g. end of second paragraph on page 13, maybe the authors could bring in some nuance in the observed effects, not all doses of CNO per se might trigger astrogliosis, as far as I can

understand from the data, this holds true for the 10 mg/kg dose, but is this also observed for the lower doses?

Are the authors sure that activation of hM3Dq receptors 'sedated' some animals and observed epileptic convulsions in other animal? It may very well be that animals that appear 'sedated' may have epileptic non-convulsive seizures, i.e. without effects on their limbs? Non-convulsive seizures can result in staring and immobilised posture.

Moreover, the statement on page 5 line 2-3 does not seem to be in line with the table 4 animal information; there I see that also 3 Rbp4cre-hM3Dq mice treated with 10 mg/kg CNO exhibited sedation like behaviour???

Reviewer #3 (Remarks to the Author):

The authors have addressed all of my concerns.

The issue of the low sample number (n = 2 in many experiments), which according to the authors resulted from a combination of Covid-related problems and from ethical concerns due to adverse side effects of the CNO, has been adequately addressed by showing individual samples on the bar graphs. This allows the reader to take the low sample number into consideration when evaluating the reliability of the results.

Minor points:

In their description of the CNO preparation, the authors now added the following sentence: "CNO was kept at -20°C and defrosted on the day of use and diluted using strike saline." What is strike saline? Was sterile saline meant?

The authors did not highlight the changes made within the manuscript as is customary, which made it difficult to identify what has been altered with respect to the original manuscript. It would be helpful to include these highlights in future revised manuscripts.

Department of Physiology, Anatomy and Genetics
University of Oxford
South Parks Road, Oxford, OX1 3QX, UK

Reception: +44-1865-272168
Fax: +44-1865-272420

Zoltán Molnár MD DPhil
Professor of Developmental Neuroscience
Tutor of Human Anatomy – St John's College

Direct Line: +44-1865-282664
Laboratory: +44-1865-282663
E-mail: Zoltan.Molnar@dpag.ox.ac.uk

Oxford, 21/6/24

Re: Response to referees:

Point by point summary of the revisions implemented in Vadisiute et al., manuscript

Reviewers' comments appear in blue, our answers in black:

Reviewer #1 (Remarks to the Author):

The manuscript has been improved, but there are still some problems.

First, the authors showed the laminar distribution of cFos+ cells after CNO application, but did not provide an adequate interpretation in the text. They still describe as if the effect is due to layer 5 projection neurons and specific cell type (see L143, "This increase was layer-specific as a function of dose as..." and the first paragraph in Discussion). Certainly neuronal activation could be originated from layer 5 neurons and a subset of hippocampal neurons, but the activation should immediately spread to cortical and subcortical regions. In fact, the new data clearly showed that cFos+ cells are densely distributed in other layers. The density appears to be rather higher in other layers than layer 5. It is therefore difficult to say the origins of various effects of chemogenetic activation, including behaviors and cellular morphology. In particular, it is likely that the effect on glial cell distribution and morphology is attributed to dramatic activation of all layer neurons. A logical interpretation should be described.

An interpretation of the results regarding PV neurons has been added at the end of the discussion. However, the description is too general to make sense of the result. Moreover, the results of statistical analysis are exaggerated despite the small sample size.

Furthermore, as the reviewer #2 pointed out, the high dose of CNO (10 mg/kg) has some effect without chemogenetic activation. This should also be considered with regard to effect on PV neurons. The authors emphasize that chemogenetic silencing did affect neither glial cells nor PV neurons even at high doses of CNO. However, it is possible that the CNO effect may be canceled by suppression of neuronal activity. It would be best to remove the result at high concentrations of CNO or to list them separately. In relation to this issue, the following statement in Discussion seems far from logical conclusion. "Our study also raises concerns regarding microglia and astrocyte responses that can be elicited even without chemogenetic activation, while solely CNO application causes clear changes in glial cells. Collectively, this shows that acute changes in the neuronal activity of a subset of cortical L5 and hippocampal neurons can have an immense global effect on glial cells."

Answers:

'First, the authors showed the laminar distribution of cFos+ cells after CNO application, but did not provide an adequate interpretation in the text. They still describe as if the effect is due to layer 5 projection neurons and specific cell type (see L143, "This increase was layer-specific as a function of dose as..." and the first paragraph in Discussion). Certainly neuronal activation could be originated from layer 5 neurons and a subset of hippocampal neurons, but the activation should immediately spread to cortical and subcortical regions. In fact, the new data clearly showed that cFos+ cells are densely distributed in other layers'.

- We agree with these comments from the referee, and we would not like to imply anything else. We acknowledged that layer 5 has many cortico-cortical and cortico-thalamic projections, and activation of layer 5 not only affects layer 5 but other projection layers as well. Second, we also agree with the reviewer that cFos labelling looks less dense in cortical layer 5 than in other layers, this is simply because cell density in layer 5 is lowest of all cortical layers.
- We added the following sentences to explain all this: lines 140-142: Within the region of Cre+ cell bodies, this increase was dose dependent (Fig 2a) and cFos+ cell density increased across all cortical layers in V1 (Supplementary Fig. 1a, not quantified).

"In particular, it is likely that the effect on glial cell distribution and morphology is attributed to dramatic activation of all layer neurons. A logical interpretation should be described"

- Lines 455-457: Changes in microglial distribution and dynamics might not only be influenced by chemogenetic manipulations of layer 5 neurons but also due to increased neuronal activity of other cortical layers.

'An interpretation of the results regarding PV neurons has been added at the end of the discussion. However, the description is too general to make sense of the result. Moreover, the results of statistical analysis are exaggerated despite the small sample size. Furthermore, as the reviewer #2 pointed out, the high dose of CNO (10 mg/kg) has some effect without chemogenetic activation. This should also be considered with regard to effect on PV neurons. The authors emphasize that chemogenetic silencing did affect neither glial cells nor PV neurons even at high doses of CNO.'

We now state that acute chemogenetic manipulations had a minor effect, considering small sample size and even with high doses of CNO we did not observe any differences in PV+. The changes can be found here:

- Lines 85-90: PV+ neuron density was not altered, yet the density of cells surrounded by perineuronal nets (PNN) decreased in layers 2 (L2), 3 and 5. Moreover, fewer PV+ cells were surrounded by perineuronal nets in cortical layer 6a of retrosplenial area (RSP) but we observed increased PV+ co-localisation with perineuronal nets in L4-5 of the primary somatosensory cortex (S1) after CNO application.

- Lines 377-378: These results suggest that acutely activating and silencing Rbp4^{Cre} neurons does not affect PV+ interneuron densities, with only minor effects on PNNs.
- Lines 480-487: Lastly, we showed that acute chemogenetic manipulations have minor effects on PV+ cells and perineuronal nets (PNNs). It is known that changes in neuronal-activity can modify PV expression and PNNs³³, but acute activation of Rbp4^{Cre} neurons seems to have no effect on PV+ expression yet decreased the density of PNNs in layers 2 (L2), 3 and 5 and increased the proportion of PV+ cells without perineuronal nets in cortical layer 6a. Moreover, DREADD activation leads to increased PV+ co-localisation with perineuronal nets in L4-5 of S1. However, acute silencing of Rbp4^{Cre} neurons did not result in notable differences in PV expression and PNNs.

'In relation to this issue, the following statement in Discussion seems far from logical conclusion. "Our study also raises concerns regarding microglia and astrocyte responses that can be elicited even without chemogenetic activation, while solely CNO application causes clear changes in glial cells. Collectively, this shows that acute changes in the neuronal activity of a subset of cortical L5 and hippocampal neurons can have an immense global effect on glial cells."

- Lines 385-394: We showed that CNO alone has no influence on neuronal activity as measured by immediate early gene expression. However, previous reports describe an effect of CNO on sleep⁴⁴. We demonstrate that acute chemogenetic activation decreases excitatory synaptic density, with silencing having the opposite effect. Additionally, acute neuronal activation rapidly increases microglia and astrocyte reactivity. Conversely, both glial cells exhibit decreased reactivity with acute neuronal silencing. We also show that, surprisingly, microglia and astrocytes rapidly respond to CNO application even without DREADDs present. However, the effect we observed with CNO in the presence of inhibitory DREADD is in a direction opposite to the effect observed with CNO alone. We therefore believe that DREADD-mediated activity itself does influence glial reactivity.

It would be best to remove the result at high concentrations of CNO or to list them separately.

We considered this suggestion made by the reviewer but decided to keep results with the high dose of CNO, since with excitatory DREADDs it is one of the most commonly used concentrations in the literature. However, in response to the reviewer request we now discuss the side effects in more detail.

In addition to reviewers' comments, we performed data analysis comparing Rbp4^{Cre} and Rbp4^{Cre}-hM3Dq mice with saline and 10 mg/kg CNO and we now discuss the interaction between genotypes and treatment conditions.

- Lines 210-236: We further compared changes in microglial density between Rbp4^{Cre} and Rbp4^{Cre}-hM3Dq mice injected with the highest dose of CNO (10 mg/kg). No significant difference in microglial density was observed in V1 laminar distribution,

across V1 and SC when comparing Rbp4^{Cre} and Rbp4^{Cre}-hM3Dq mice injected with saline or CNO (genotypes x treatment conditions: laminar distribution $F_{(3,12)} = 0.9723$, $p = 0.4378$; V1: $F_{(1,12)} = 1.753$, $p = 0.2102$; SC: $F_{(1,14)} = 0.04187$, $p = 0.8408$, Supplementary Fig.3a-c). We observed statistically significant interaction between genotypes and treatment conditions in microglial density in the CA1 region (genotypes x treatment conditions: $F_{(1,14)} = 7.196$, $p = 0.0179$, Supplementary Fig. 3d). In CA3, microglial density was increased in saline Rbp4^{Cre}-hM3Dq compared with saline Rbp4^{Cre} mice but decreased in Rbp4^{Cre}-hM3Dq after 10 mg/kg CNO application compared to Rbp4^{Cre} mice injected with the same concentration of CNO (genotypes x treatment conditions: $F_{(1,14)} = 56.49$, $p < 0.0001$, Supplementary Fig. 3e). Statistically significant interactions between genotype and treatment condition were also observed when assessing the percentage of Iba1+ microglia that were also CD68+. Specifically, we found no statistically significant interaction between genotypes and treatment conditions in CD68 in Iba1 microglia in V1 layer 5 and SC, but it was statistically significant in CA1 and CA3 regions (genotypes x treatment conditions: V1 L5 ($F_{(1,14)} = 0.8998$, $p = 0.3589$; SC: $F_{(1,14)} = 2.611$, $p = 0.1284$; CA1: $F_{(1,14)} = 22.05$, $p = 0.0003$; CA3: $F_{(1,14)} = 43.66$, $p < 0.0001$, Supplementary Fig.3f-i). Moreover, in all four brain regions assessed, the percentage of CD68 in Iba1+ microglia was increased in saline injected Rbp4^{Cre}-hM3Dq mice compared with Rbp4^{Cre}, and in 10 mg/kg CNO injected Rbp4^{Cre}-hM3Dq compared with Rbp4^{Cre} (Supplementary Fig.3f-i). These data suggest that the high dose of CNO has no effect with or without chemogenetic activation in V1 and Rbp4^{Cre} projection region to SC but has an effect both by itself and in combination with chemogenetic activation in the CA3 and CA1 regions. Given the effect of high doses of CNO alone in the Rbp4^{Cre} control mice, we cannot be certain whether the reduction in microglial density observed in CA1 and CA3 in Rbp4^{Cre}-hM3Dq after 10 mg/kg CNO (Fig 4d) is the result of CNO, or chemogenetic activation.

- Lines 290-299: We further investigated changes in GFAP+ density, S100β+ density and the proportion of GFAP+ cells expressing S100β between Rbp4^{Cre} and Rbp4^{Cre}-hM3Dq mice injected with saline and the highest dose of CNO. In V1, we found no statistically significant interaction between genotypes and treatment conditions in laminar distribution of GFAP+ cell or across V1 (genotypes x treatment conditions: laminar $F_{(3,12)} = 0.9723$, $p = 0.4378$; across V1 $F_{(1,11)} = 3.055$, $p = 0.1083$), but we observed significant changes in the laminar distribution of S100β+ and GFAP+ cells expressing S100β, but not across V1 (genotypes x treatment conditions: laminar S100β+: $F_{(3,11)} = 6.909$, $p = 0.007$, across $F_{(1,11)} = 3.417$, $p = 0.0916$; laminar GFAP+ cells expressing S100β $F_{(3,11)} = 5.605$, $p = 0.014$, across $F_{(1,11)} = 2.559$, $p = 0.1379$ Supplementary Fig. 5a,b,f,j).
- Lines 314-330: Furthermore, in SC, we found no statistically significant interaction between genotypes and treatment conditions (genotypes x treatment conditions: GFAP+ $F_{(1,11)} = 0.9638$, $p = 0.3473$; S100β+ $F_{(1,11)} = 3.071$, $p = 0.9957$; GFAP+ cells expressing S100β: $F_{(1,11)} = 1.470$, $p = 0.2508$, Supplementary Fig. 5c,g,k). In CA1, we observed a statistically significant interaction between genotypes and treatment conditions in GFAP+ and S100β+ cell densities and GFAP+ cells expressing S100β (genotypes x treatment conditions: GFAP+ $F_{(1,11)} = 16.74$, $p = 0.0018$; S100β+ $F_{(1,11)} = 8.632$, $p = 0.0135$; GFAP+ cells expressing S100β: $F_{(1,11)} = 17.25$, $p = 0.0016$,

Supplementary Fig. 5d,h,l). GFAP+ and S100 β + cell densities were increased in Rbp4^{Cre}-hM3Dq mice compared to Rbp4^{Cre} mice injected with saline, and the density of GFAP+ cells expressing S100 β was lower in Rbp4^{Cre}-hM3Dq mice compared to Rbp4^{Cre} mice injected with 10 mg/kg CNO (Supplementary Fig. 5d,h,l). Finally, in CA3, we observed a statistically significant interaction between genotypes and treatment conditions (genotypes x treatment conditions: GFAP+ $F_{(1,11)} = 6.654$, $p = 0.0256$; S100 β + $F_{(1,11)} = 13.38$, $p = 0.0038$; GFAP+ cells expressing S100 β : $F_{(1,11)} = 30.60$, $p = 0.0002$, Supplementary Fig. 5e,i,m). GFAP+ density was increased in Rbp4^{Cre}-hM3Dq mice compared with Rbp4^{Cre} injected with saline and CNO. Moreover, following CNO injection, GFAP+ cells expressing S100 β decreased in Rbp4^{Cre}-hM3Dq mice compared to Rbp4^{Cre}.

Reviewer #2 (Remarks to the Author):

First of all, I would like to thank the authors for conducting the requested additional control experiments including these with CNO alone in Rbp4Cre mice. They also answered most questions adequately. A few concerns however arise/remain.

As shown in the new results, the higher dose of CNO (10 mg/kg) by itself has several effects that may affect the readouts from the experiments in the DREADD mice. Despite the fact that the authors clearly describe these unexpected, toxic (?) effects, however, they take little or no account of these results when describing the effects of chemogenetic modulation on astrocytes and microglia with this highest dose.

It is clear that 10 mg/kg CNO has effects on several read-outs, wouldn't it be better NOT to use this highest dose precisely in those experiments with DREADD modulation looking at the same read-outs? In such cases, how can one separate the CNO effects from the DREADD-mediated effects? For example, suppl Fig 3, the effect of 10 mg/kg CNO is very pronounced in Rbp4Cre mice (panel c) and therefore makes it very difficult, if not impossible, to correctly interpret panel a of the same figure. Panel a second graph S100beta, with low doses rather decreased density, only at the high dose an effect, impossible to tell whether this effect arises from CNO alone or from Gq-DREADD activation by CNO in my opinion. In panel b the 10 mg/kg dose was not used, only lower doses.

It is very important to describe the effects of CNO 10 mg/kg in the article and indicate that they are way too high. However, I also think it is important to describe the real, true DREADD mediated effects, so maybe with the low CNO doses but not with 10 mg/kg. Would the authors be willing to critically revise their data and description, and add more nuance here? There are certainly DREADD mediated effects at lower CNO doses, and some read-outs, such as cFos in neurons, are not affected by 10 mg/kg CNO. But in other places an unambiguous interpretation or description is not possible in my opinion, especially on the microglial and astrocytic markers. For example page 3 lines 18-23. Another example is Fig 4 with description on page 7 lines 13-18.

Also in the discussion the observation that microglia and astrocytes respond to (high?) CNO

application is mentioned, but these effects are not further dissected from the real DREADD mediated responses, and therefore make the discussion unclear and sometimes a bit incoherent. E.g. page 11 lines 8-14. On what basis do the authors link the effects of CNO application alone also to neuronal activity changes? E.g. page 12 lines 16-29, the effect of CNO alone on microglia cannot be denied, maybe in the Gi-DREADD experiments the effects are opposite and therefore compensating or antagonising the CNO effect? E.g. end of second paragraph on page 13, maybe the authors could bring in some nuance in the observed effects, not all doses of CNO per se might trigger astrogliosis, as far as I can understand from the data, this holds true for the 10 mg/kg dose, but is this also observed for the lower doses?

Are the authors sure that activation of hM3Dq receptors 'sedated' some animals and observed epileptic convulsions in other animal? It may very well be that animals that appear 'sedated' may have epileptic non-convulsive seizures, i.e. without effects on their limbs? Non-convulsive seizures can result in staring and immobilised posture. Moreover, the statement on page 5 line 2-3 does not seem to be in line with the table 4 animal information; there I see that also 3 Rbp4^{cre}-hM3Dq mice treated with 10 mg/kg CNO exhibited sedation like behaviour???

Answers:

In response to reviewers' comments, we performed data analysis comparing Rbp4^{Cre} and Rbp4^{Cre}-hM3Dq mice with saline and 10 mg/kg CNO and we discussed interaction between genotypes and treatment conditions in more detail. However, we decided not to remove 10 mg/kg CNO data because we feel that it is important that this issue is discussed not just for the higher doses, but for all. This effect might be present not only at 10 mg/kg dose, but all other doses. We kept the 10 mg/kg experiments with the others and now clearly articulate these concerns throughout the text as the referee requested. See lines:

- Lines 210-236: We further compared changes in microglial density between Rbp4^{Cre} and Rbp4^{Cre}-hM3Dq mice injected with the highest dose of CNO (10 mg/kg). No significant difference in microglial density was observed in V1 laminar distribution, across V1 and SC when comparing Rbp4^{Cre} and Rbp4^{Cre}-hM3Dq mice injected with saline or CNO (genotypes x treatment conditions: laminar distribution $F_{(3,12)} = 0.9723$, $p = 0.4378$; V1: $F_{(1,12)} = 1.753$, $p = 0.2102$; SC: $F_{(1,14)} = 0.04187$, $p = 0.8408$, Supplementary Fig.3a-c). We observed statistically significant interaction between genotypes and treatment conditions in microglial density in the CA1 region (genotypes x treatment conditions: $F_{(1,14)} = 7.196$, $p = 0.0179$, Supplementary Fig. 3d). In CA3, microglial density was increased in saline Rbp4^{Cre}-hM3Dq compared with saline Rbp4^{Cre} mice but decreased in Rbp4^{Cre}-hM3Dq after 10 mg/kg CNO application compared to Rbp4^{Cre} mice injected with the same concentration of CNO (genotypes x treatment conditions: $F_{(1,14)} = 56.49$, $p < 0.0001$, Supplementary Fig. 3e). Statistically significant interactions between genotype and treatment condition were also observed when assessing the percentage of Iba1+ microglia that were also CD68+. Specifically, we found no statistically significant interaction between genotypes and treatment conditions in CD68 in Iba1 microglia in V1 layer 5 and SC, but it was statistically significant in CA1 and CA3 regions (genotypes x treatment

conditions: V1 L5 ($F_{(1,14)} = 0.8998$, $p = 0.3589$; SC: $F_{(1,14)} = 2.611$, $p = 0.1284$; CA1: $F_{(1,14)} = 22.05$, $p = 0.0003$; CA3: $F_{(1,14)} = 43.66$, $p < 0.0001$, Supplementary Fig.3f-i). Moreover, in all four brain regions assessed, the percentage of CD68 in Iba1+ microglia was increased in saline injected Rbp4^{Cre}-hM3Dq mice compared with Rbp4^{Cre}, and in 10 mg/kg CNO injected Rbp4^{Cre}-hM3Dq compared with Rbp4^{Cre} (Supplementary Fig.3f-i). These data suggest that the high dose of CNO has no effect with or without chemogenetic activation in V1 and Rbp4^{Cre} projection region to SC but has an effect both by itself and in combination with chemogenetic activation in the CA3 and CA1 regions. Given the effect of high doses of CNO alone in the Rbp4^{Cre} control mice, we cannot be certain whether the reduction in microglial density observed in CA1 and CA3 in Rbp4^{Cre}-hM3Dq after 10 mg/kg CNO (Fig 4d) is the result of CNO, or chemogenetic activation.

- Lines 290-299: We further investigated changes in GFAP+ density, S100 β + density and the proportion of GFAP+ cells expressing S100 β between Rbp4^{Cre} and Rbp4^{Cre}-hM3Dq mice injected with saline and the highest dose of CNO. In V1, we found no statistically significant interaction between genotypes and treatment conditions in laminar distribution of GFAP+ cell or across V1 (genotypes x treatment conditions: laminar $F_{(3,12)} = 0.9723$, $p = 0.4378$; across V1 $F_{(1,11)} = 3.055$, $p = 0.1083$), but we observed significant changes in the laminar distribution of S100 β + and GFAP+ cells expressing S100 β , but not across V1 (genotypes x treatment conditions: laminar S100 β +: $F_{(3,11)} = 6.909$, $p = 0.007$, across $F_{(1,11)} = 3.417$, $p = 0.0916$; laminar GFAP+ cells expressing S100 β $F_{(3,11)} = 5.605$, $p = 0.014$, across $F_{(1,11)} = 2.559$, $p = 0.1379$ Supplementary Fig. 5a,b,f,j).
- Lines 314-330: Furthermore, in SC, we found no statistically significant interaction between genotypes and treatment conditions (genotypes x treatment conditions: GFAP+ $F_{(1,11)} = 0.9638$, $p = 0.3473$; S100 β + $F_{(1,11)} = 3.071$, $p = 0.9957$; GFAP+ cells expressing S100 β : $F_{(1,11)} = 1.470$, $p = 0.2508$, Supplementary Fig. 5c,g,k). In CA1, we observed a statistically significant interaction between genotypes and treatment conditions in GFAP+ and S100 β + cell densities and GFAP+ cells expressing S100 β (genotypes x treatment conditions: GFAP+ $F_{(1,11)} = 16.74$, $p = 0.0018$; S100 β + $F_{(1,11)} = 8.632$, $p = 0.0135$; GFAP+ cells expressing S100 β : $F_{(1,11)} = 17.25$, $p = 0.0016$, Supplementary Fig. 5d,h,l). GFAP+ and S100 β + cell densities were increased in Rbp4^{Cre}-hM3Dq mice compared to Rbp4^{Cre} mice injected with saline, and the density of GFAP+ cells expressing S100 β was lower in Rbp4^{Cre}-hM3Dq mice compared to Rbp4^{Cre} mice injected with 10 mg/kg CNO (Supplementary Fig. 5d,h,l). Finally, in CA3, we observed a statistically significant interaction between genotypes and treatment conditions (genotypes x treatment conditions: GFAP+ $F_{(1,11)} = 6.654$, $p = 0.0256$; S100 β + $F_{(1,11)} = 13.38$, $p = 0.0038$; GFAP+ cells expressing S100 β : $F_{(1,11)} = 30.60$, $p = 0.0002$, Supplementary Fig. 5e,i,m). GFAP+ density was increased in Rbp4^{Cre}-hM3Dq mice compared with Rbp4^{Cre} injected with saline and CNO. Moreover, following CNO injection, GFAP+ cells expressing S100 β decreased in Rbp4^{Cre}-hM3Dq mice compared to Rbp4^{Cre}.

Supplementary figure 5 (lines 1371-1397)

Supplementary Figure 3: Microglial response to CNO application in Rbp4^{Cre} and Rbp4^{Cre}-hM3Dq mice

a, b, No significant differences in microglial density in primary visual cortex in Rbp4^{Cre} and Rbp4^{Cre}-hM3Dq mice after 10 mg/kg CNO or saline injection. Mice: $n = 4$ saline Rbp4^{Cre}, $n = 5$ CNO 10 mg/kg Rbp4^{Cre}, $n = 4$ saline Rbp4^{Cre}-hM3Dq, $n = 3$ CNO 10 mg/kg Rbp4^{Cre}-hM3Dq, with an average of 3 sections per region, a - evaluated using mixed-effects ANOVA via the corrected method of Benjamini and Yekutieli and b - evaluated using two-way ANOVA via the corrected method of Benjamini and Yekutieli. **c**, No significant differences in microglial density in superior colliculus when comparing control Rbp4^{Cre} and Rbp4^{Cre}-hM3Dq mice injected with saline or injected with 10 mg/kg CNO. However, within genotype, microglial density decreased in Rbp4^{Cre} mice after 10 mg/kg CNO injection compared to saline. Similarly, microglial density decreased in Rbp4^{Cre}-hM3Dq mice after 10 mg/kg CNO injection compared with saline. Mice: $n = 4$ saline Rbp4^{Cre}, $n = 5$ CNO 10 mg/kg Rbp4^{Cre}, $n = 6$ saline Rbp4^{Cre}-hM3Dq, $n = 3$ CNO 10 mg/kg Rbp4^{Cre}-hM3Dq, with an average of 3 sections per region, evaluated using two-way ANOVA via the corrected method of Benjamini and Yekutieli. **d**, Microglial density is significantly different between control Rbp4^{Cre} and Rbp4^{Cre}-hM3Dq mice injected with saline in CA1 region, but no significant difference between genotypes injected with 10 mg/kg CNO. Microglial density was significantly decreased in Rbp4^{Cre}-hM3Dq mice after 10 mg/kg CNO application compared with saline. Mice: $n = 4$ saline Rbp4^{Cre}, $n = 5$ CNO 10 mg/kg Rbp4^{Cre}, $n = 6$ saline Rbp4^{Cre}-hM3Dq, $n = 3$ CNO 10 mg/kg Rbp4^{Cre}-hM3Dq, with an average of 3 sections per region, evaluated using two-way ANOVA via the corrected method of Benjamini and Yekutieli. **e**, Microglial density in CA3 is significantly different in control Rbp4^{Cre} mice and Rbp4^{Cre}-hM3Dq mice injected with saline, and also when comparing Rbp4^{Cre} and Rbp4^{Cre}-

hM3Dq mice injected with 10 mg/kg CNO. Microglial density increased in Rbp4^{Cre} mice after 10 mg/kg CNO application compared with saline but decreased in Rbp4^{Cre}-hM3Dq after CNO application compared with saline. Mice: *n* = 4 saline Rbp4^{Cre}, *n* = 5 CNO 10 mg/kg Rbp4^{Cre}, *n* = 6 saline Rbp4^{Cre}-hM3Dq, *n* = 3 CNO 10 mg/kg Rbp4^{Cre}-hM3Dq, with an average of 3 sections per region, evaluated using two-way ANOVA via the corrected method of Benjamini and Yekutieli. **f, g, h, i**, Significant baseline difference in % of Iba1+ microglia that are also CD68+ in primary visual cortex layer 5, superior colliculus, CA1 and CA3 regions between Rbp4^{Cre} and Rbp4^{Cre}-hM3Dq mice injected with saline, and Rbp4^{Cre} and Rbp4^{Cre}-hM3Dq injected with 10 mg/kg CNO. Moreover, injection with 10 mg/kg CNO increased the percentage of Iba1+ microglia that were also CD68+ in all regions tested in both genotypes (Rbp4^{Cre} and Rbp4^{Cre}-hM3Dq). Mice: *n* = 4 saline Rbp4^{Cre}, *n* = 5 CNO 10 mg/kg Rbp4^{Cre}, *n* = 6 saline Rbp4^{Cre}-hM3Dq, *n* = 3 CNO 10 mg/kg Rbp4^{Cre}-hM3Dq, with an average of 3 sections per region, evaluated using two-way ANOVA via the corrected method of Benjamini and Yekutieli. All data are presented as mean ± SEM, false discovery rate of 0.05 adjusted using Benjamini and Yekutieli, * *p* < 0.05, ** *p* < 0.01, *** *p* < 0.001, **** *p* < 0.0001. Detailed statistical information is listed in Supplementary Statistical Data. Source data is provided as a Source Data file.

Supplementary figure 5

Supplementary Figure 5: Changes in astrocytes response to CNO application between Rbp4^{Cre} and Rbp4^{Cre}-hM3Dq mice

a, No significant changes in GFAP+ and the proportion of GFAP+ cells expressing S100β in primary visual cortex between Rbp4^{Cre} and Rbp4^{Cre}-hM3Dq mice after 10 mg/kg CNO application. S100β density increased in Rbp4^{Cre}-hM3Dq after 10 mg/kg CNO application compared with Rbp4^{Cre}-hM3Dq saline and compared to Rbp4^{Cre} mice injected with 10 mg/kg CNO. Mice: $n = 4$ saline Rbp4^{Cre}, $n = 5$ CNO 10 mg/kg Rbp4^{Cre}, $n = 4$ saline Rbp4^{Cre}-hM3Dq, $n = 3$ CNO 10 mg/kg Rbp4^{Cre}-hM3Dq, with an average of 3 sections per region, evaluated using mixed-effects ANOVA via the corrected method of Benjamini and Yekutieli. **b, c, d, e**, Changes in GFAP+ density in primary visual cortex, superior colliculus, CA1 and CA3 regions between: Rbp4^{Cre} and Rbp4^{Cre}-hM3Dq injected with saline; Rbp4^{Cre} and Rbp4^{Cre}-hM3Dq injected with 10 mg/kg CNO; Rbp4^{Cre} injected with saline and 10 mg/kg CNO; Rbp4^{Cre}-hM3Dq injected with saline and 10 mg/kg CNO. Mice: $n = 4$ saline Rbp4^{Cre}, $n = 5$ CNO 10 mg/kg Rbp4^{Cre}, $n = 4$ saline Rbp4^{Cre}-hM3Dq, $n = 3$ CNO 10 mg/kg Rbp4^{Cre}-hM3Dq, with an average of 3 sections per region, evaluated using two-way ANOVA via the corrected method of Benjamini and

Yekutieli. **f, g, h, i**, Changes in S100 β + density in primary visual cortex, superior colliculus, CA1 and CA3 regions between: Rbp4^{Cre} and Rbp4^{Cre}-hM3Dq injected with saline; Rbp4^{Cre} and Rbp4^{Cre}-hM3Dq injected with 10 mg/kg CNO; Rbp4^{Cre} injected with saline and 10 mg/kg CNO; Rbp4^{Cre}-hM3Dq injected with saline and 10 mg/kg CNO. Mice: $n = 4$ saline Rbp4^{Cre}, $n = 5$ CNO 10 mg/kg Rbp4^{Cre}, $n = 4$ saline Rbp4^{Cre}-hM3Dq, $n = 3$ CNO 10 mg/kg Rbp4^{Cre}-hM3Dq, with an average of 3 sections per region, evaluated using two-way ANOVA via the corrected method of Benjamini and Yekutieli. **j, k, l, m**, Changes in the proportion of GFAP+ cells expressing S100 β in primary visual cortex, superior colliculus, CA1 and CA3 regions between: Rbp4^{Cre} and Rbp4^{Cre}-hM3Dq injected with saline; Rbp4^{Cre} and Rbp4^{Cre}-hM3Dq injected with 10 mg/kg CNO; Rbp4^{Cre} injected with saline and 10 mg/kg CNO; Rbp4^{Cre}-hM3Dq injected with saline and 10 mg/kg CNO. Mice: $n = 4$ saline Rbp4^{Cre}, $n = 5$ CNO 10 mg/kg Rbp4^{Cre}, $n = 4$ saline Rbp4^{Cre}-hM3Dq, $n = 3$ CNO 10 mg/kg Rbp4^{Cre}-hM3Dq, with an average of 3 sections per region, evaluated using two-way ANOVA via the corrected method of Benjamini and Yekutieli. All data are presented as mean \pm SEM, false discovery rate of 0.05 adjusted using Benjamini and Yekutieli, * $p < 0.05$, ** $p < 0.01$, *** $p < 0.001$, **** $p < 0.0001$. Detailed statistical information is listed in Supplementary Statistical Data. Source data is provided as a Source Data file.

We agree with Reviewer 2 on the assessment of the behavioural changes observed after the activation of hM3Dq receptors with high doses of CNO. We agree that the 'sedated', immobile state without epileptic convulsions may in fact indicate epileptic non-convulsive seizures, i.e. seizures but without effects on their limbs. We agree that non-convulsive seizures can result in staring and immobilised posture. We now discuss these possibilities in the text, but the resolution would require EEG recordings under these conditions.

- Lines 406-409: With 10 mg/kg CNO we observed a range of adverse effects ranging from death to a sedated, immobile state without movements of limbs. Further EEG recording would be required to study whether this immobilised posture is indicative of non-convulsive seizures.

Reviewer #3 (Remarks to the Author):

The authors have addressed all of my concerns.

The issue of the low sample number ($n = 2$ in many experiments), which according to the authors resulted from a combination of Covid-related problems and from ethical concerns due to adverse side effects of the CNO, has been adequately addressed by showing individual samples on the bar graphs. This allows the reader to take the low sample number into consideration when evaluating the reliability of the results.

Minor points:

In their description of the CNO preparation, the authors now added the following sentence: "CNO was kept at -20°C and defrosted on the day of use and diluted using strike saline." What is strike saline? Was sterile saline meant?

Answer: Yes, we meant sterile saline.

The authors did not highlight the changes made within the manuscript as is customary,

which made it difficult to identify what has been altered with respect to the original manuscript. It would be helpful to include these highlights in future revised manuscripts.

Answer: yes, we agreed and submitted a marked-up version to the editor. We also submitted both versions of the manuscript with these final revisions.

** See the Nature Portfolio author and referees' website at www.nature.com/authors for information about policies, services and author benefits

Communications Biology is committed to improving transparency in authorship. As part of our efforts in this direction, we are now requesting that all authors identified as 'corresponding author' create and link their Open Researcher and Contributor Identifier (ORCID) with their account on the Manuscript Tracking System prior to acceptance. ORCID helps the scientific community achieve unambiguous attribution of all scholarly contributions. You can create and link your ORCID from the home page of the Manuscript Tracking System by clicking on 'Modify my Springer Nature account' and following the instructions in the link below. Please also inform all co-authors that they can add their ORCIDs to their accounts and that they must do so prior to acceptance.

If you experience problems in linking your ORCID, please contact the Platform Support Helpdesk.

This email has been sent through the Springer Nature Tracking System NY-610A-NPG&MTS

Confidentiality Statement:

This e-mail is confidential and subject to copyright. Any unauthorised use or disclosure of its contents is prohibited. If you have received this email in error please notify our Manuscript Tracking System Helpdesk team at <http://platformsupport.nature.com> .

Details of the confidentiality and pre-publicity policy may be found here <http://www.nature.com/authors/policies/confidentiality.html>

Privacy Policy | Update Profile

With kind regards,

Zoltán Molnár MD DPhil
Professor of Developmental Neuroscience
University of Oxford
<https://www.dpag.ox.ac.uk/team/zoltan-molnar>

Auguste Vadisiute MSc DPhil
Junior Research Fellow in Physiology and Medicine
University of Oxford
<https://www.dpag.ox.ac.uk/team/auguste-vadisiute>

REVIEWERS' COMMENTS:

Reviewer #1 (Remarks to the Author):

The authors have made the revision based on the reviewers' comments including my comments. I think that the final decision should be done by the editor

Reviewer #3 (Remarks to the Author):

The authors have adequately addressed my remaining concerns or have at least described the conflicting results with 10 mg/kg CNO in detail.

Line 490 maybe add '10 mg/kg CNO' in sentence "some concerning effects of ..."

Department of Physiology, Anatomy and Genetics
University of Oxford
South Parks Road, Oxford, OX1 3QX, UK

Reception: +44-1865-272168
Fax: +44-1865-272420

Zoltán Molnár MD DPhil
Professor of Developmental Neuroscience
Tutor of Human Anatomy – St John's College

Direct Line: +44-1865-282664
Laboratory: +44-1865-282663
E-mail: Zoltan.Molnar@dpag.ox.ac.uk

Oxford, 13/8/24

Re: Response to referees:

Point by point summary of the revisions implemented in Vadisiute et al., manuscript

Reviewers' comments appear in blue, our answers in black:

Reviewer #1 (Remarks to the Author):

The authors have made the revision based on the reviewers' comments including my comments. I think that the final decision should be done by the editor.

Reviewer #3 (Remarks to the Author):

The authors have adequately addressed my remaining concerns or have at least described the conflicting results with 10 mg/kg CNO in detail.

Line 490 maybe add '10 mg/kg CNO' in sentence "some concerning effects of ...".

We added additional information: line 512: "as well as some concerning effects of 10mg/kg CNO itself."

** See the Nature Portfolio author and referees' website at www.nature.com/authors for information about policies, services and author benefits

Communications Biology is committed to improving transparency in authorship. As part of our efforts in this direction, we are now requesting that all authors identified as 'corresponding author' create and link their Open Researcher and Contributor Identifier (ORCID) with their account on the Manuscript Tracking System prior to acceptance. ORCID helps the scientific community achieve unambiguous attribution of all scholarly contributions. You can create and link your ORCID from the home page of the Manuscript Tracking System by clicking on 'Modify my Springer Nature account' and following the instructions in the link below. Please also inform all co-authors that they can add their ORCIDs to their accounts and that they must do so prior to acceptance.

If you experience problems in linking your ORCID, please contact the Platform Support Helpdesk.

This email has been sent through the Springer Nature Tracking System NY-610A-NPG&MTS

Confidentiality Statement:

This e-mail is confidential and subject to copyright. Any unauthorised use or disclosure of its contents is prohibited. If you have received this email in error please notify our Manuscript Tracking System Helpdesk team at <http://platformsupport.nature.com>.

Details of the confidentiality and pre-publicity policy may be found here <http://www.nature.com/authors/policies/confidentiality.html>

Privacy Policy | Update Profile

With kind regards,

Zoltán Molnár MD DPhil
Professor of Developmental Neuroscience
University of Oxford
<https://www.dpag.ox.ac.uk/team/zoltan-molnar>

Auguste Vadisiute MSc DPhil
Junior Research Fellow in Physiology and Medicine
University of Oxford
<https://www.dpag.ox.ac.uk/team/auguste-vadisiute>